# Parasites trigger epithelial cell crosstalk to drive gut–brain signalling

Kouki K. Touhara[1✉], Jinhao Xu[1], Joel Castro[2,3], Hong-Erh Liang[4], Guochuan Li[5,6], Mariana Brizuela[2,3], Andrea M. Harrington[2,3], Sonia Garcia-Caraballo[2,3], Tracey O'Donnell[2,3], Daniel Neumann[2,3], Nathan D. Rossen[1,7], Fei Deng[5,6], Gudrun Schober[2,3], Yulong Li[5,6], Richard M. Locksley[4,8,9], Stuart M. Brierley[2,3] & David Julius[1✉]

Parasitic infections modulate both immune and sensory responses, but how these systems collaborate to elicit protective behaviours remains incompletely understood. The gut epithelium contains specialized sensory cells that detect pathogens and irritants. These include cholinergic tuft cells, which sense parasites and initiate type 2 immune responses[1–3], as well as serotonergic enterochromaffin (EC) cells, which detect irritants and communicate with afferent nerve fibres to transmit nociceptive signals[4–6]. Here we show that paracrine signalling between these cells constitutes a mechanism for neuro–immune interaction and gut–brain communication. We find that tuft cells use two distinct mechanisms of acetylcholine (ACh) release despite lacking synaptic vesicles and excitable membranes. These include acute release in response to parasite-derived metabolites, followed by constitutive 'leak-like' release, which occurs with type 2 inflammation. Although both mechanisms can activate muscarinic receptors on crypt-residing EC cells, only the sustained mode of ACh release elicits levels of serotonin sufficient to stimulate vagal afferent neurons that suppress food intake. This two-phase paracrine signalling mechanism explains how parasitic infection progresses from an initial asymptomatic phase to symptomatic established disease, in which type 2 immune and sensory signalling pathways within the gut–brain axis collaborate to evoke protective behaviours.

The gastrointestinal tract is equipped with a complex sensory system that detects a range of harmful stimuli and infections, evoking appropriate protective responses. The first line of defence is provided by sensory cells in the epithelial lining of the gut, including EC and tuft cells[7,8]. EC cells are now recognized as polymodal integrators of noxious stimuli in the gut, where they detect ingested irritants, stress-related agents and mechanical stimulation[5,8–11]. Once activated, these cells release serotonin, eliciting visceral hypersensitivity, nausea-like sensations and/or pain through 5-HT$_3$ receptor-expressing primary afferent nerve fibres that densely innervate the gut mucosa[4–6,8]. Tuft cells respond to parasitic worm (helminth) and protist infections[1–3], which also elicit abdominal discomfort, diarrhoea, nausea and malnutrition[12,13], but whether and how tuft and EC cells collaborate to transmit such nocifensive information to the brain remains unknown.

When tuft cells detect helminth and protist infections, they release interleukin 25 (IL-25), initiating a type 2 immune response that leads to a cascade of defensive actions such as tissue remodelling and enhanced mucus secretion to combat and clear the infections[1–3]. Tuft cells also promote fluid secretion, decrease worm fecundity and induce mastocytosis through the release of ACh, providing additional protection against parasitic infections[14,15]. Notably, although tuft cells express *Chat*, a biosynthetic enzyme responsible for the synthesis of ACh, they lack secretory vesicles or other cellular machinery that is generally required for ACh release, such as vesicular ACh and choline transporters[16–19]. This raises questions about the mechanisms and specific physiological conditions that govern ACh release by tuft cells, as well as the downstream targets engaged by this action. Here we address these key questions by delineating a neuro–immune pathway in which intra-epithelial crosstalk between tuft and EC cells activates a neural circuit to suppress food intake during parasitic infections.

## ACh selectively activates crypt EC cells

If tuft and EC cells collaborate to detect noxious stimuli, they should be capable of communicating through some sort of paracrine signalling mechanism. We therefore asked whether EC cells can be activated by the major tuft-cell-derived transmitter, ACh. Using organoids derived from *Tac1^Cre^;Polr2a^GCaMP5g-IRES-tdTomato^* mice, in which the calcium indicator

[1]Department of Physiology, University of California, San Francisco, San Francisco, CA, USA. [2]Visceral Pain Research Group, Hopwood Centre for Neurobiology, South Australian Health and Medical Research Institute (SAHMRI), Adelaide, South Australia, Australia. [3]School of Pharmacy and Biomedical Science, College of Health, Adelaide University, Adelaide, South Australia, Australia. [4]Department of Medicine, University of California, San Francisco, San Francisco, CA, USA. [5]State Key Laboratory of Membrane Biology, New Cornerstone Science Laboratory, School of Life Sciences, Peking University, Beijing, China. [6]PKU–IDG/McGovern Institute for Brain Research, Beijing, China. [7]Tetrad Graduate Program, University of California, San Francisco, San Francisco, CA, USA. [8]Department of Microbiology and Immunology, University of California, San Francisco, San Francisco, CA, USA. [9]Howard Hughes Medical Institute, University of California, San Francisco, San Francisco, CA, USA. ✉e-mail: Koki.Tohara@ucsf.edu; David.Julius@ucsf.edu

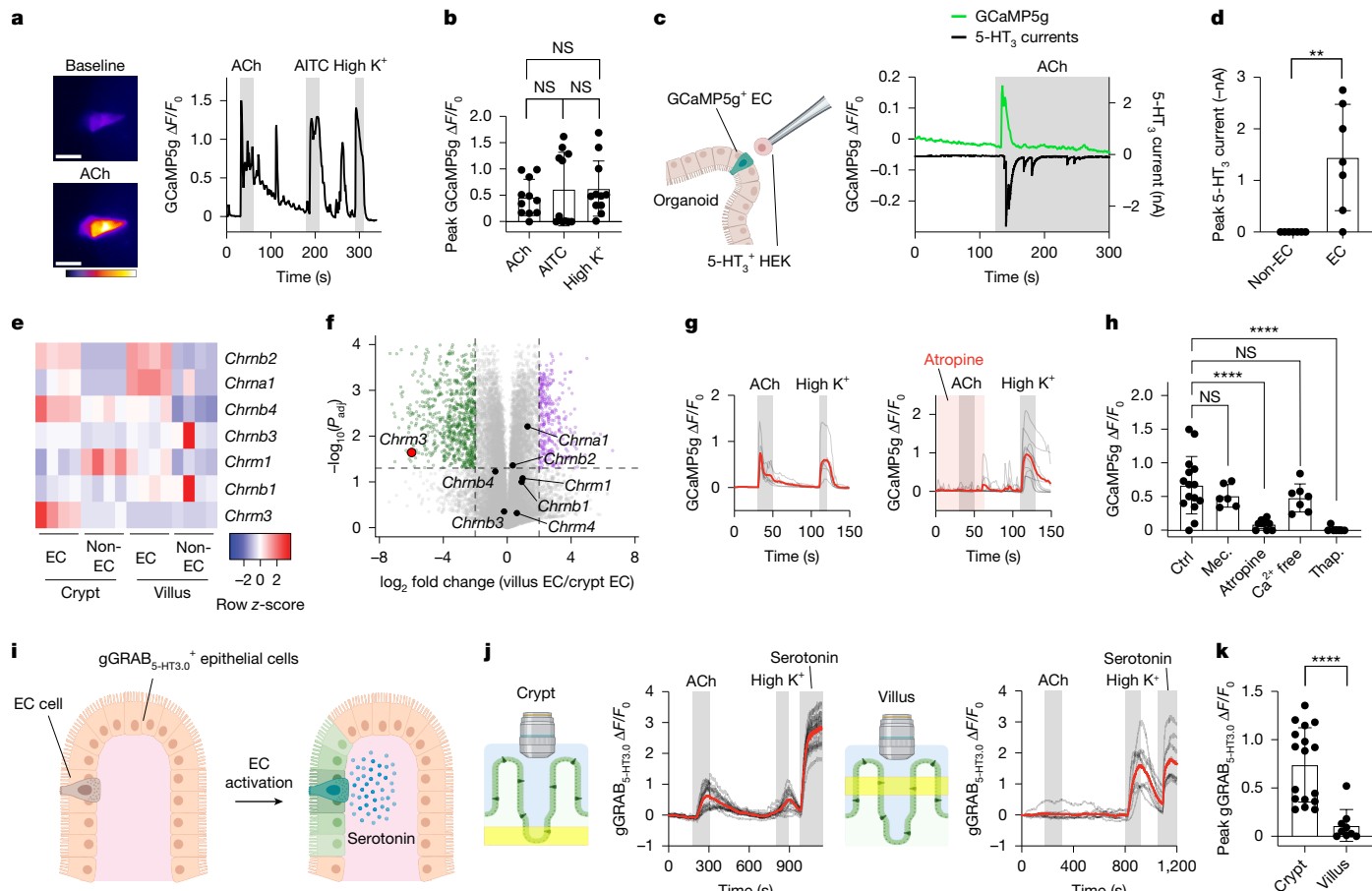

**Fig. 1 | ACh activates crypt EC cells. a**, GCaMP5g signals in EC cells within *Tac1^Cre^;Polr2a^GCaMP5g-IRES-tdTomato^* organoids in response to 10 μM ACh, 100 μM AITC and high K⁺ (70 mM KCl). Scale bars, 10 μm. Colour scale shows relative fluorescence intensity. **b**, Peak GCaMP5g signals from EC cells. Mean ± s.d., Kruskal–Wallis test with Dunn's multiple comparisons; *P* > 0.9999. *n* = 11 cells. **c**, 5-HT₃ currents in HEK293FT cells positioned adjacent to EC cells during stimulation with 10 μM ACh. **d**, Peak 5-HT₃ currents from EC and non-EC cells. Mean ± s.d., two-tailed Welch's *t*-test; *P* = 0.0099. *n* = 7 cells. **e**, *Z*-scores for ACh receptor expression in crypt and villus EC and non-EC cells. **f**, Volcano plot of differential gene expression between crypt and villus EC cells, with ACh receptor genes highlighted. Green and purple indicate crypt- and villus-biased genes, respectively (|log₂ fold change| > 2, adjusted *P* < 0.05). **g**, GCaMP5g response of EC cells to 10 μM ACh with (right) or without (left) 10 μM atropine (a mAChR antagonist). Individual cells in grey, average in red. **h**, Peak GCaMP5g responses of EC cells to 10 μM ACh under various conditions: mecamylamine (Mec., 10 μM), *P* = 0.6197; atropine (10 μM), *P* < 0.0001; Ca²⁺-free, *P* = 0.4070; and thapsigargin pre-treatment (Thap., 4 μM), *P* < 0.0001. Mean ± s.d., ordinary one-way ANOVA with Dunnett's test. *n* = 15, 6, 8, 7 and 7 cells (bars from left to right). **i**, *Vil^Cre^;gGRAB₅-HT3.0* mice enable visualization of serotonin. **j**, ACh-stimulated serotonin release in crypts or villi. EC cells were stimulated with 10 μM ACh and high K⁺ (70 mM KCl). gGRAB₅-HT3.0 was fully activated with 20 μM serotonin. Individual cells in grey, average in red. **k**, Peak gGRAB₅-HT3.0 response. Mean ± s.d., Mann–Whitney test; *P* < 0.0001. *n* = 17 crypts and 9 villi. **\*\****P* < 0.01, **\*\*\*\****P* < 0.0001. NS, not significant. Illustrations in **b,i,j** created in BioRender; Touhara, K. https://BioRender.com/q580au1 (2026).

GCaMP5g and tdTomato are selectively expressed in EC cells[5], we found that ACh robustly stimulated EC cells to a level comparable with responses elicited by the TRPA1 channel agonist allyl isothiocyanate (AITC, commonly known as mustard oil), or after depolarization with high external potassium (Fig. 1a,b). Notably, responses to ACh were also seen in dissociated EC cells, ruling out indirect activation through gap junctions (Extended Data Fig. 1a).

To determine whether ACh promotes the release of serotonin from EC cells, we positioned HEK293FT cells expressing serotonin-gated ion channels (5-HT₃ receptors) next to EC cells[8], enabling simultaneous measurement of calcium signals in EC cells and serotonin-evoked membrane currents in the adjacent biosensor cell (Fig. 1c). Significant inward currents were observed after ACh-evoked stimulation of EC cells, promoting bolus serotonin release at levels (micromolar) sufficient to activate 5-HT₃ receptors[5] (Fig. 1c,d).

To pinpoint the subtype of ACh receptors mediating this response, we analysed bulk-RNA-sequencing data from EC cells isolated from both crypts and villi[11], revealing expression of muscarinic ACh receptor 3

(*Chrm3*) specifically within crypt-residing EC cells (Fig. 1e,f and Extended Data Fig. 1b). This finding was corroborated functionally in that ACh-evoked responses in EC cells were inhibited by atropine, a muscarinic receptor antagonist, but not by mecamylamine, a nicotinic receptor antagonist (Fig. 1g,h and Extended Data Fig. 1c). In addition, depletion of Ca²⁺ from intracellular stores, but not from the extracellular medium, diminished the ACh response (Fig. 1h and Extended Data Fig. 1c), further supporting the involvement of metabotropic muscarinic (rather than ionotropic nicotinic) receptors.

We next asked whether ACh preferentially stimulates EC cells residing in crypts versus villi, as suggested by our RNA-sequencing data (Fig. 1e,f and Extended Data Fig. 1b). We used *Vil^Cre^;gGRAB₅-HT3.0* mice (serotonin sensor mice), in which a genetically encoded fluorescent serotonin sensor, gGRAB₅-HT3.0, is expressed throughout the intestinal epithelium[5,20] (Fig. 1i). We imaged filleted jejunal segments from these mice to monitor the release and propagation of serotonin within the intact crypt–villus architecture (Fig. 1j). ACh elicited serotonin release from EC cells within crypts, but not within villi (Fig. 1j,k and Extended Data

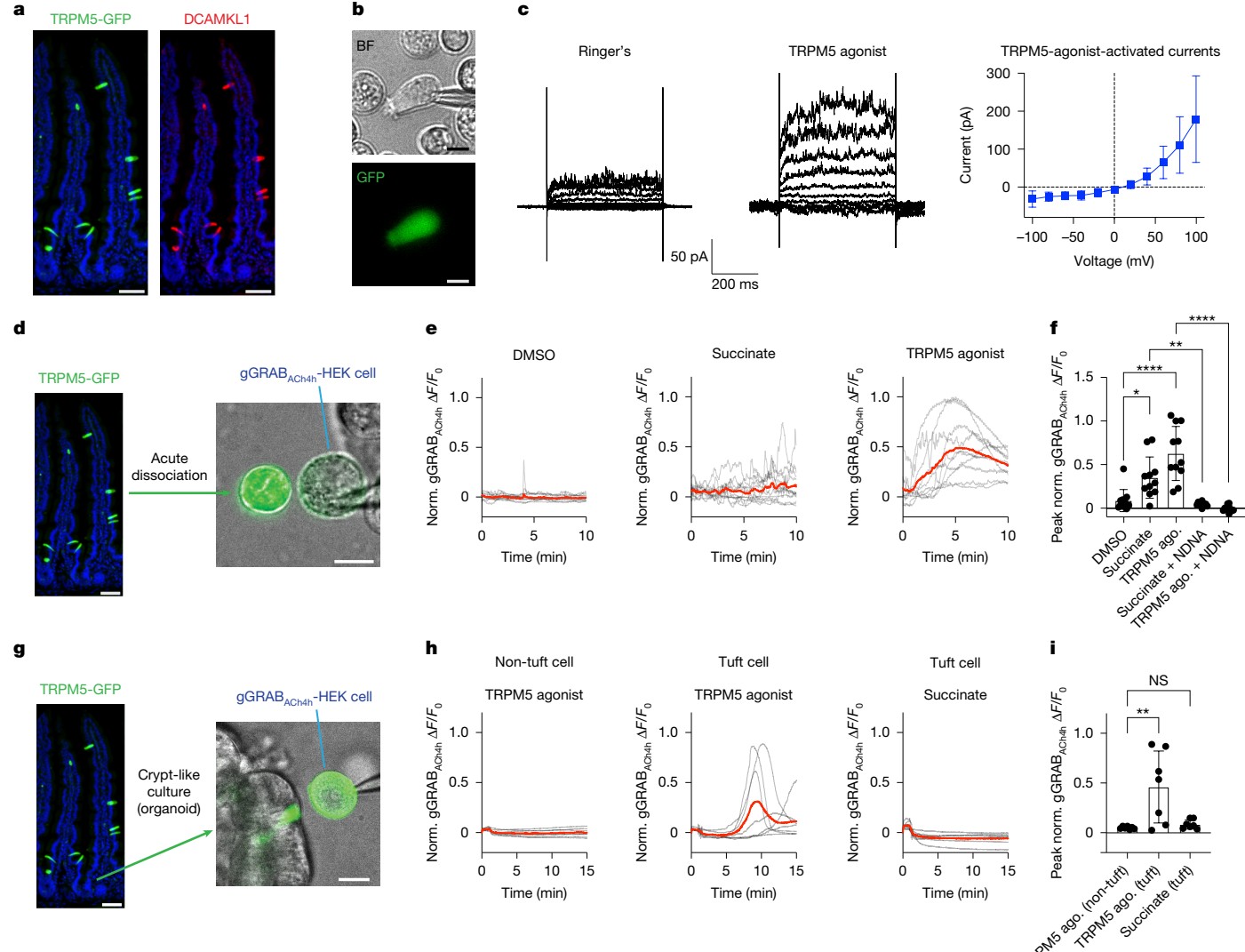

**Fig. 2 | Isolated tuft cells release ACh upon TRPM5 stimulation. a**, TRPM5-GFP mice label tuft cells (red). Representative of at least five independent experiments. **b**, Isolated tuft cells show brush structure. BF, bright-field. **c**, Voltage-clamp recordings from isolated tuft cells. Membrane held at −80 mV; 20-mV steps from −100 mV to 100 mV. TRPM5 channels were activated by 2 μM TRPM5 agonist, and activated currents are plotted (mean ± s.d., *n* = 6 cells). **d**, gGRAB$_{ACh4h}$ biosensor HEK293FT cells were positioned adjacent to isolated tuft cells. **e,f**, Normalized gGRAB$_{ACh4h}$ signals (**e**) and quantification (**f**) from biosensor HEK293FT cells while isolated tuft cells were stimulated by dimethyl sulfoxide (DMSO), 10 mM succinate or 10 μM TRPM5 agonist. The TRPM5 antagonist NDNA (10 μM) was used to block TRPM5. Signals were normalized to full sensor activation. Individual cells in grey, average in red. Mean ± s.d., ordinary one-way ANOVA with Dunnett's comparisons. *n* = 11, 11, 11, 10 and 10 cells (bars from left to right). *P* = 0.0166 (DMSO versus succinate); *P* < 0.0001 (DMSO versus TRPM5 agonist); *P* = 0.0051 (succinate versus succinate + NDNA); *P* < 0.0001 (TRPM5 agonist versus TRPM5 agonist + NDNA). **g**, GRAB$_{ACh4h}$ biosensor HEK293FT cells were positioned adjacent to tuft cells within crypts. **h,i**, Normalized gGRAB$_{ACh4h}$ signals (**h**) and quantification (**i**) from biosensor HEK293FT cells while non-tuft or tuft cells in crypts were stimulated by 10 mM succinate or 10 μM TRPM5 agonist. Signals were normalized to full sensor activation. Individual cells in grey, average in red. Mean ± s.d., ordinary one-way ANOVA with Dunnett's comparisons. *n* = 7 cells. *P* = 0.005 (non-tuft versus tuft with TRPM5 agonist); *P* = 0.9474 (non-tuft stimulated by TRPM5 agonist versus tuft stimulated by succinate). Scale bars, 100 μm (**a** and **d,g** left); 10 μm (**b** and **d,g** right). *\*P* < 0.05, \*\**P* < 0.01, \*\*\*\**P* < 0.0001. NS, not significant.

Fig. 1d,e). Together, our results show that ACh preferentially activates crypt-residing EC cells through muscarinic ACh receptors (mAChRs), leading to a substantial release of serotonin that is capable of activating downstream serotonergic pathways.

## Two temporal phases of ACh release

Having identified crypt EC cells as a bona fide target of ACh, we wondered whether and how they are activated by cholinergic tuft cells within the same epithelial compartment. Whereas canonical cholinergic neurons are excitable and release ACh via synaptic vesicles, our analysis of published single-cell RNA-sequencing datasets[21]

revealed that tuft cells lack synaptic-vesicle machinery, consistent with previous electron microscopic studies[16,17] (Extended Data Fig. 2a). Nevertheless, tuft-derived ACh has been shown to regulate fluid secretion[14,15], raising the question of whether tuft cells release ACh through an unconventional cell-autonomous mechanism, or by interaction with surrounding cells through gap junctions or lateral interdigitating spinules[16,17]. To address this question, we characterized the biophysical properties of acutely dissociated tuft cells obtained from TRPM5-GFP mice, in which intestinal tuft cells are specifically labelled (Fig. 2a).

Whole-cell patch-clamp recordings from acutely dissociated tuft cells showed no evidence of voltage-activated inward currents and only

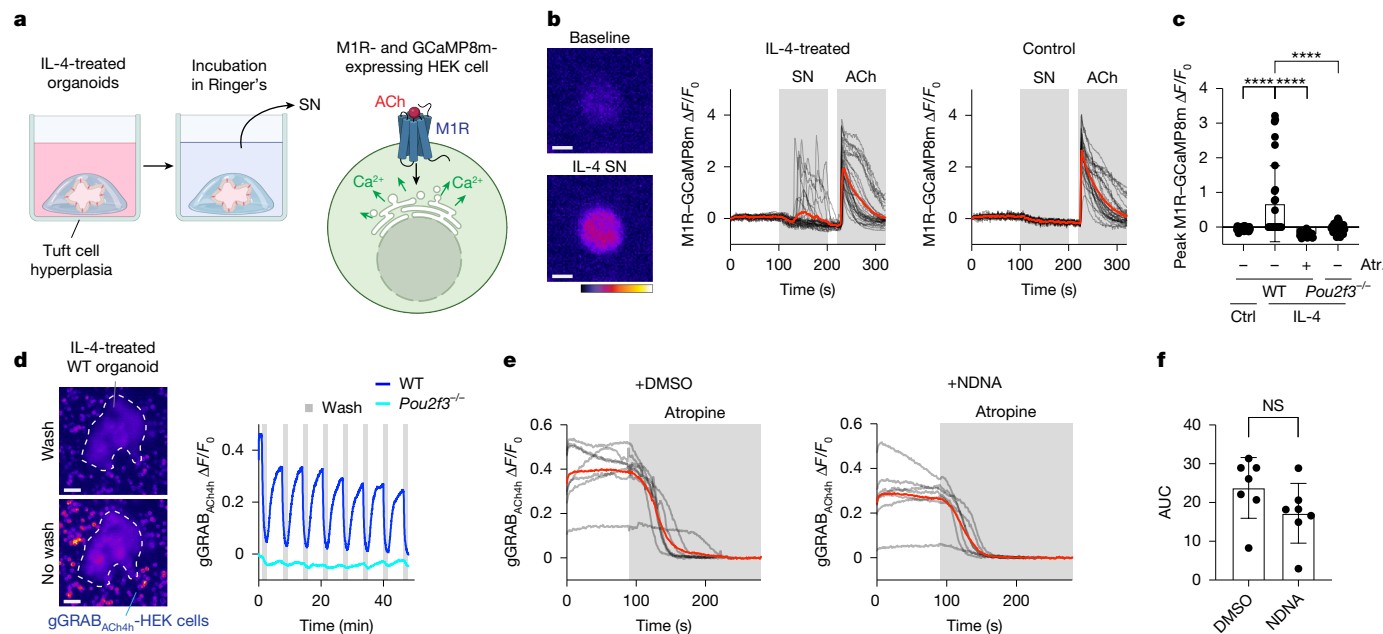

**Fig. 3 | Tuft cells exhibit constitutive ACh release during type 2 inflammation.** **a**, Schematic of the M1R–GCaMP8m biosensor experiments. Tuft cell hyperplasia was induced by exposing organoids to 20 ng ml$^{-1}$ IL-4 for two days. After IL-4 washout, organoids were incubated in 300 µl Ringer's solution for two hours. The resulting supernatant (SN) was perfused onto HEK293FT cells expressing M1R and GCaMP8m. **b**, Supernatants from IL-4-treated wild-type organoids, but not from vehicle-treated controls, stimulated M1R in HEK293FT cells. M1R was fully activated with 10 µM ACh at the end of recordings. Individual cells in grey, average in red. Scale bars, 5 µm. **c**, Comparison of peak M1R–GCaMP signals elicited by the supernatants collected from control and IL-4-treated wild-type (WT) or *Pou2f3$^{-/-}$* organoids. The specificity of the response to M1R-mediated calcium influx was validated by adding 1 µM atropine (Atr.). Mean ± s.d., two-way ANOVA with Tukey's multiple comparisons; *P* < 0.0001 (Ctrl versus IL-4, IL-4 versus IL-4 + atropine and IL-4 versus *Pou2f3$^{-/-}$*). *n* = 30 cells. **d**, IL-4-treated organoids exhibit continuous ACh release. gGRAB$_{ACh4h}$ biosensor HEK293FT cells were plated surrounding the IL-4-treated wild-type (blue) or *Pou2f3$^{-/-}$* (cyan) organoids. ACh was repeatedly washed away with local perfusion. Representative of at least four independent recordings. Scale bars, 50 µm. **e,f**, ACh release was quantified in the presence of DMSO or 10 µM NDNA (a TRPM5 antagonist). gGRAB$_{ACh4h}$ was fully quenched by 10 µM atropine. Individual cells in grey, average in red (**e**); AUC, area under the curve (**f**). Mean ± s.d., two-tailed Welch's *t*-test; *P* = 0.1405. *n* = 7 cells. ****P < 0.0001. NS, not significant. Illustrations in **a** created in BioRender; Touhara, K. https://BioRender.com/us460pk and https://BioRender.com/6u6lqbg (2026).

minimal steady-state outward currents (83 ± 50 pA at 80 mV; *n* = 11 cells) (Fig. 2b,c). This included an absence of voltage-gated calcium currents, as assessed using recording solutions containing high extracellular Ba$^{2+}$ (Extended Data Fig. 2b). On the other hand, tuft cells exhibited robust voltage-dependent outward currents in response to a selective agonist for the TRPM5 ion channel[14], confirming their viability in this preparation (Fig. 2c). These findings align with our transcriptomic analysis showing that tuft cells express TRPM5, but not voltage-gated sodium or calcium channels (Extended Data Fig. 2a). Thus, in contrast to canonical cholinergic cells, intestinal tuft cells lack excitable membranes.

We then asked whether isolated tuft cells can release ACh in response to stimulation. Succinate serves as a protist-derived factor that acts on tuft cells to initiate type 2 immune responses[22–24]. Tuft cells express succinate receptor 1 (*Sucnr1*), which stimulates the release of stored calcium through G$_q$–phospholipase C signalling, thereby activating Ca$^{2+}$-activated TRPM5 channels. To test whether this pathway triggers ACh release, we positioned HEK293FT cells expressing an ACh sensor (gGRAB$_{ACh4h}$) adjacent to dissociated tuft cells, such that ACh release can be assessed by measuring increased gGRAB$_{ACh4h}$ fluorescence[25] (Fig. 2d and Extended Data Fig. 2c). Although we observed minimal spontaneous ACh release, stimulation with either succinate or the TRPM5 agonist triggered robust acute ACh release from single isolated tuft cells, which was blocked by the specific TRPM5 antagonist NDNA[26] (Fig. 2e,f and Extended Data Fig. 2c).

We next validated these findings in intestinal organoids derived from TRPM5-GFP mice, which maintain crypt epithelial architecture and cell–cell interactions (Fig. 2g). Here, we observed ACh release after TRPM5 stimulation, but not in response to succinate (Fig. 2h,i). This finding aligns with observations that succinate receptors (*Sucnr1*)

are expressed by villus-residing but not by crypt-residing tuft cells[27] (Extended Data Fig. 3a,b). Together, these findings show that tuft cells, in isolation or within an epithelial environment, release ACh autonomously and acutely in response to physiological stimuli, despite lacking electrical excitability and conventional synaptic-release machinery.

When parasites colonize the intestine and begin to mature and reproduce, they induce type 2 inflammation and subsequent tuft cell hyperplasia over several days[1–3]. Having established that tuft cells acutely release ACh in response to succinate—which mimics initial parasitic infection—we next asked whether tuft cells continue to release ACh constitutively during type 2 inflammation. To model an ongoing type 2 immune response, we exposed intestinal organoids to recombinant interleukin 4 (IL-4), which induces tuft cell hyperplasia[1–3] (Extended Data Fig. 3c). After treatment with IL-4, organoids were washed and cultured in IL-4-free medium for one day, then incubated in Ringer's solution for two hours. This supernatant fluid was then applied to HEK293FT cells ectopically expressing the muscarinic ACh receptor 1 (M1R) together with the calcium indicator GCaMP8m (Fig. 3a). Activation of M1R by ACh triggers the release of Ca$^{2+}$ from intracellular stores, measurable as elevated GCaMP8m fluorescence. We observed robust activation of M1R by supernatant collected from IL-4-treated organoids, but not from vehicle-treated controls. Furthermore, supernatant from IL-4-treated *Pou2f3$^{-/-}$* organoids, which lack tuft cells, did not activate M1R biosensor cells (Fig. 3b,c, and Extended Data Fig. 3d). These results show that type 2 immune activation drives the release of ACh specifically from tuft cells.

To corroborate these bulk measurements, we positioned HEK293FT cells expressing the gGRAB$_{ACh4h}$ sensor adjacent to single tuft cells

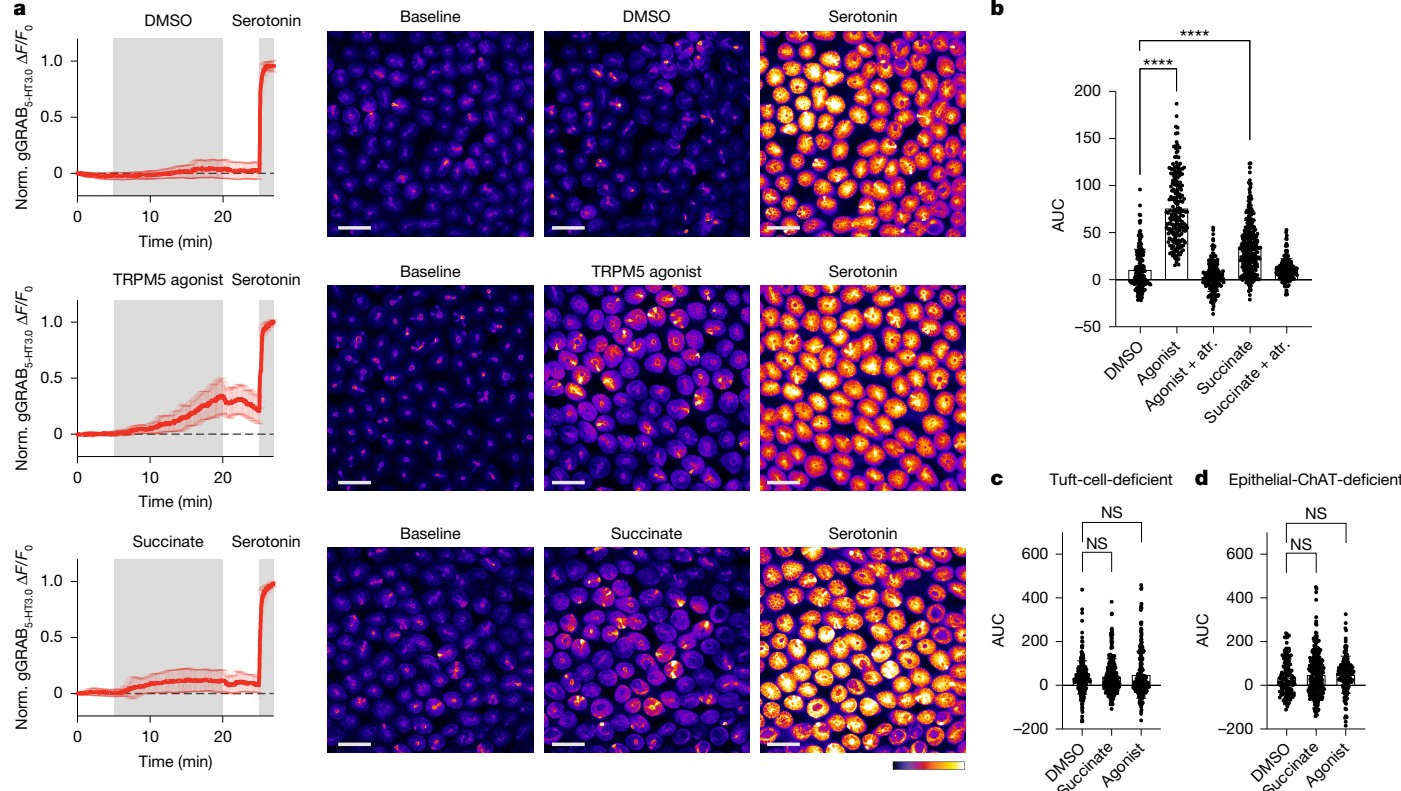

**Fig. 4 | Acute tuft cell stimulation leads to EC cell activation. a**, Serotonin release was monitored in crypts using ex vivo intestine preparations from $Vil^{Cre}$;$gGRAB_{5\text{-}HT3.0}$ mice. Tuft cells were stimulated with DMSO, 10 µM TRPM5 agonist or 10 mM succinate. Average normalized $gGRAB_{5\text{-}HT3.0}$ signals ± s.d. (left) and corresponding fluorescence images at baseline, during stimulation and after serotonin application (right). Scale bars, 100 µm. **b**, Quantification of AUC for normalized $gGRAB_{5\text{-}HT3.0}$ signals. Communication between tuft and EC cells was blocked by 10 µM atropine (atr). Mean ± s.d., Kruskal–Wallis test with Dunn's multiple comparison; $P < 0.0001$. $n = 167, 196, 194, 271$ and $173$ crypts (bars from left to right). **c**, Serotonin release in tuft-cell-deficient mice ($Vil^{Cre}$;$gGRAB_{5\text{-}HT3.0}$;$Pou2f3^{-/-}$) stimulated with DMSO, 10 mM succinate or 10 µM TRPM5 agonist. Mean ± s.d., Kruskal–Wallis test with Dunn's multiple comparisons; $P > 0.9999$. $n = 247, 245$ and $227$ crypts (bars from left to right). **d**, Serotonin release in epithelial-ChAT-deficient mice ($Vil^{Cre}$;$gGRAB_{5\text{-}HT3.0}$; $Chat^{flox/flox}$) stimulated with DMSO, 10 mM succinate or 10 µM TRPM5 agonist, showing that tuft-cell-derived ACh is required for EC cell activation. Mean ± s.d., Kruskal–Wallis test with Dunn's multiple comparison; $P > 0.9999$. $n = 120, 270$ and $136$ crypts (bars from left to right). ****$P < 0.0001$. NS, not significant.

within TRPM5-GFP organoids, where we also observed pronounced release of ACh after IL-4 treatment (Extended Data Fig. 3e,f). To assess the duration of ACh release, we surrounded IL-4-treated organoids with $gGRAB_{ACh4h}$ biosensor cells and monitored ACh release over an extended period with periodic washouts (Fig. 3d). Notably, we detected sustained ACh release from IL-4-treated organoids for more than 45 min, which was not blocked by the TRPM5 antagonist (Fig. 3e,f). Together, these results reveal the existence of a prolonged 'leak-like' release mode that is temporally and mechanistically distinct from the acute TRPM5-dependent stimulus-evoked mode.

## Tuft-cell-derived ACh activates EC cells

Having established that tuft cells release ACh through both acute and constitutive mechanisms, we asked whether these distinct modes are capable of activating crypt EC cells within live intestines. For acute release, we used either the TRPM5 agonist or succinate to stimulate tuft cells in intestinal segments isolated from serotonin sensor-expressing mice ($Vil^{Cre}$;$gGRAB_{5\text{-}HT3.0}$). Both compounds elicited serotonin release from crypts, which was diminished by mAChR blockade (Fig. 4a,b and Extended Data Fig. 4a). To confirm the involvement of tuft cells, we generated serotonin sensor mice lacking either tuft cells ($Vil^{Cre}$;$gGRAB_{5\text{-}HT3.0}$; $Pou2f3^{-/-}$) or choline acetyltransferase (ChAT) in intestinal tuft cells ($Vil^{Cre}$;$gGRAB_{5\text{-}HT3.0}$;$Chat^{flox/flox}$). In these mice, neither agonist elicited serotonin release from crypt EC cells, confirming that tuft-cell-derived

ACh mediates this paracrine signalling pathway (Fig. 4c,d and Extended Data Fig. 4b).

To ask whether tuft cells also activate EC cells during ongoing type 2 inflammation, we performed intraperitoneal injection of IL-25 to replicate the cascade of type 2 responses downstream of tuft cell activation[1–3]. After four consecutive days of administering IL-25 to serotonin sensor mice, we observed tuft cell hyperplasia in the small intestine (Fig. 5a). We isolated jejunal sections and performed serotonin sensor imaging in crypts. After monitoring basal serotonin secretion for 8 min, we applied a saturating dose of serotonin for normalization, followed by the high-affinity $gGRAB_{5\text{-}HT3.0}$ sensor antagonist RS 23597-190 to quench the signal and establish baseline (Fig. 5b). IL-25-injected mice showed markedly increased basal serotonin levels compared with saline-injected controls (Fig. 5b,c and Extended Data Fig. 5a). The increased serotonin levels were significantly reduced by acute atropine treatment, and were absent in preparations from mice lacking tuft cells or ChAT in intestinal tuft cells (Fig. 5b,c and Extended Data Fig. 5a–c). These results show that tuft-cell-derived ACh activates muscarinic receptors on crypt EC cells to promote sustained serotonin release during type 2 inflammation.

To examine this pathway in a pathophysiological setting, we infected serotonin sensor mice with *Nippostrongylus brasiliensis* (*Nb*), a rodent model of human hookworm infection[28], and measured baseline serotonin levels after nine days, when tuft cell hyperplasia peaks[1–3] (Fig. 5d). *Nb* infection increased crypt serotonin levels, which were diminished in tuft-cell-deficient and ChAT-deficient cohorts,

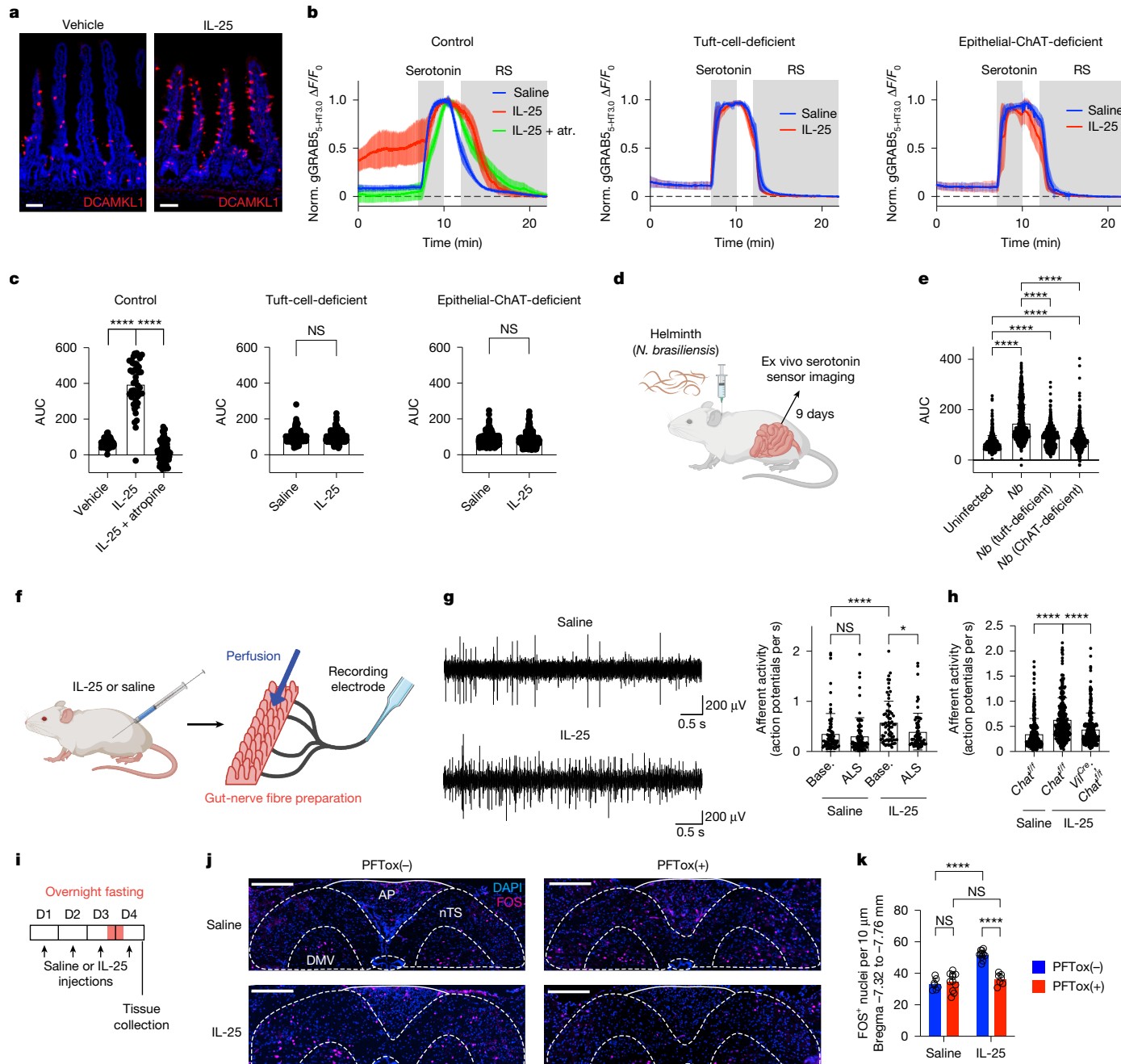

**Fig. 5 | Type 2 inflammation activates the tuft cell–EC cell–vagal axis.**
**a**, IL-25-induced tuft cell hyperplasia. **b**, Normalized gGRAB$_{5\text{-HT3.0}}$ signals ± s.d. in $Vil^{Cre}$;gGRAB$_{5\text{-HT3.0}}$ (left), $Vil^{Cre}$;gGRAB$_{5\text{-HT3.0}}$;$Pou2f3^{-/-}$ (tuft-cell-deficient; middle) and $Vil^{Cre}$;gGRAB$_{5\text{-HT3.0}}$;$Chat^{flox/flox}$ (epithelial-ChAT-deficient; right) mice receiving IL-25 or saline (four days). Wild-type samples were pre-incubated in 10 µM atropine. gGRAB$_{5\text{-HT3.0}}$ was activated with 20 µM serotonin and quenched with 50 µM RS 23597-190 (RS). **c**, AUC at baseline. Mean ± s.d. Left, $Vil^{Cre}$; gGRAB$_{5\text{-HT3.0}}$ mice. Brown–Forsythe and Welch ANOVA tests; $P < 0.0001$. $n = 100$, 45 and 78 crypts (bars from left to right). Middle, $Vil^{Cre}$;gGRAB$_{5\text{-HT3.0}}$;$Pou2f3^{-/-}$ (tuft-cell-deficient) mice. Mann–Whitney test; $P = 0.9860$. $n = 242$ (saline) and 356 (IL-25) crypts. Right, $Vil^{Cre}$;gGRAB$_{5\text{-HT3.0}}$;$Chat^{flox/flox}$ (epithelial-ChAT-deficient) mice. Mann–Whitney test; $P = 0.0939$. $n = 288$ (saline) and 299 (IL-25) crypts. **d**, Schematic of $Nb$ infection. **e**, AUC at baseline in the indicated cohorts. Mean ± s.d., Kruskal–Wallis test (Dunn's multiple comparison); $P < 0.0001$. $n = 495$, 710, 556 and 543 crypts (bars from left to right). **f**, Schematic. **g**, Jejunal

afferent recordings from saline- or IL-25-injected mice. Black traces: total single units with or without 10 µM alosetron (ALS). Mean ± s.d., Kruskal–Wallis test (Dunn's multiple comparisons); $P > 0.9999$ (baseline: saline versus ALS: saline), $P < 0.0001$ (baseline: saline versus baseline: IL-25), $P = 0.0208$ (baseline: IL-25 versus ALS: IL-25). $n = 78$ (saline) and 68 (IL-25) fibres. **h**, Epithelial ChAT deletion reduced IL-25-induced nerve activity. Mean ± s.d., Kruskal–Wallis test (Dunn's multiple comparisons); $P < 0.0001$. $n = 266$, 250 and 289 fibres (bars from left to right). **i**–**k**, PFTox mice received IL-25 or saline (four days). **i**, Schematic. **j**, FOS$^+$ neurons (magenta) in medial nTS. **k**, Quantification of FOS$^+$ nuclei. Mean ± s.d., two-way ANOVA (Tukey's test). $P < 0.0001$ (PFTox(−): saline versus PFTox(−): IL-25 and PFTox(−): IL-25 versus PFTox(+): IL-25). $P = 0.8052$ (PFTox(−): saline versus PFTox(+): saline). $P = 0.9989$ (PFTox(+): saline versus PFTox(+): IL-25). $n = 7$, 11, 11 and 7 mice (bars from left to right). Scale bars, 100 µm (**a**,**j**). *$P < 0.05$, ****$P < 0.0001$. NS, not significant. Illustrations in **d**,**f** created in BioRender; Touhara, K. https://BioRender.com/so6yfih (2026).

confirming our findings from IL-25 experiments (Fig. 5e and Extended Data Fig. 5d,e).

## Constitutive ACh drives the EC cell–vagal axis

We next investigated the physiological consequences of tuft–EC cell activation. Most extrinsic mucosal fibres innervating the small intestine are of vagal origin and are activated by serotonin through 5-HT$_3$ receptors[5]. Because ACh has also been reported to directly activate vagal afferents[29], we sought to determine whether tuft cell-derived ACh acts mainly through the EC cell–serotonin pathway or via direct stimulation of mucosal vagal afferents. Using calcium imaging to characterize transmitter sensitivity of traced and dissociated mucosal vagal neurons (Extended Data Fig. 6a), we observed that 68% responded to mCPBG (a selective 5-HT$_3$ receptor agonist), whereas only 28% responded to ACh (Extended Data Fig. 6b). This direct ACh activation is likely to involve both muscarinic and nicotinic receptors, because responses were reduced by antagonists to each subtype (Extended Data Fig. 6c). Moreover, transcripts for both muscarinic and nicotinic receptors were expressed by these neurons (Extended Data Fig. 6d). To directly assess the sensitivities of mucosal nerve fibres, we performed jejunal afferent recordings using ex vivo gut-nerve fibre preparations from mice expressing channelrhodopsin 2 (ChR2) in Na$_V$1.8$^+$ fibres (Extended Data Fig. 6e), enabling us to test whether a given stimulus sensitizes nerve endings by enhancing their response to light stimulation. ACh increased the basal firing frequency and lowered the activation threshold, effects that were mostly abolished by alosetron, a 5-HT$_3$ receptor antagonist that attenuates EC cell–nerve communication (Extended Data Fig. 6f–h). Together, these findings show that the tuft–EC axis mediates most ACh effects on mucosal afferents, rather than through direct ACh activation.

We next asked whether the tuft–EC axis activates mucosal afferents during acute tuft cell stimulation versus type 2 inflammation. Notably, acute stimulation of tuft cells with succinate did not increase nerve fibre activity in ex vivo gut-nerve preparations (Extended Data Fig. 6i). Similarly, the TRPM5 agonist had no effect on basal nerve firing, although it did enhance optogenetic sensitivity (Extended Data Fig. 6j), consistent with our previous observation that the TRPM5 agonist evoked greater release of serotonin from EC cells than did succinate (Fig. 4a). By contrast, IL-25-induced type 2 inflammation significantly increased the basal compound action potential frequency, which was diminished by alosetron (Fig. 5f,g) and reduced in preparations from mice lacking ChAT in intestinal tuft cells (Fig. 5h). These results reveal that functional communication between the tuft–EC axis and mucosal afferents occurs most robustly during type 2 inflammation, compared with acute stimulation.

Vagal afferents that innervate the small intestine project predominantly to the nucleus of the solitary tract (nTS) in the brainstem[30]. In line with this, after IL-25-induced type 2 inflammation (Fig. 5i), FOS expression increased in the nTS, but not in the area postrema (AP) or dorsal motor nucleus (DMV) (Fig. 5j,k and Extended Data Fig. 7a). This increase was abrogated in PFTox(+) mice, in which expression of tetanus toxin in EC cells (Tac1$^{Cre}$;ePet$^{Flp}$;PFTox) blocks transmitter release[4] (Fig. 5j,k), confirming that type 2 inflammation activates nTS-projecting vagal afferents through EC cell-mediated signalling. Of note, activated nTS neurons showed minimal overlap with Tac1-expressing neurons, which mediate nausea-like responses to bacterial toxins[6], or with other established populations in the nTS (expressing Dbh, Npy, Prlh, Th, Cck or Nr4a2)[31–34] (Extended Data Fig. 7b,c). Together, these findings reveal that the tuft–EC axis activates a unique subset of nTS neurons during type 2 inflammation, distinct from canonical nausea circuits, whereas acute tuft cell stimulation does not engage this pathway.

## The tuft cell–EC cell–vagal axis suppresses feeding

To determine the behavioural consequences of tuft–EC activation, we measured food intake as a relevant physiological response that is suppressed by global activation of EC cell–nociceptive circuits[21,35]. We first tested acute effects by orally administering the TRPM5 agonist and monitoring food intake over three hours (Extended Data Fig. 8a). The TRPM5 agonist did not alter food intake compared with saline controls (Extended Data Fig. 8b), nor did it affect locomotor, grooming, orienting or rearing behaviours that would indicate visceral malaise[36] (Extended Data Fig. 8c,d). These behavioural findings align with our nerve fibre recordings showing that acute tuft–EC activation minimally stimulates vagal afferents (Extended Data Fig. 6i,j).

We next examined food intake during type 2 inflammation, when the tuft–EC axis activates vagal afferents most effectively. PFTox(+) and PFTox(−) control mice exhibited normal baseline food intake before treatment with IL-25 (Fig. 6a). However, type 2 inflammation induced a marked decrease in food intake that was both delayed and attenuated in PFTox(+) mice, compared with PFTox(−) mice (Fig. 6a). These findings show that EC cell signalling mediates feeding suppression during type 2 immune responses. To examine whether reduced food intake depends on tuft-cell-derived ACh in a more physiological setting, we infected mice with Nb and monitored food intake over 11 days. Wild-type mice reduced their food intake at the peak of tuft cell hyperplasia (days 7–9), whereas tuft-cell-deficient (Pou2f3$^{−/−}$) and ChAT-deficient (Vil$^{Cre}$;Chat$^{flox/flox}$) mice maintained normal feeding (Fig. 6b,c), confirming that tuft-cell-derived ACh mediates the reduction in food intake during type 2 immune activation. Notably, all mice showed reduced food intake on day 1 after infection, regardless of genotype, probably owing to systemic effects associated with the migration of Nb larvae through the bloodstream to the lungs, or to bacterial contamination from the rat faecal culture used for collecting Nb larvae[28] (Fig. 6b,c). We also collected the brainstem on day 11 after infection and quantified FOS$^+$ neurons in the nTS. Consistent with our previous results, we observed fewer FOS$^+$ neurons in Pou2f3$^{−/−}$ and VilCre;Chat$^{flox/flox}$ mice than in controls (Extended Data Fig. 9a,b). However, the difference was less pronounced than what we observed in PFTox mice (Fig. 5j,k), probably because tissues were collected after peak type 2 immune responses. In any case, our results revealed that the tuft–EC axis sends signals to the nTS through vagal afferents, leading to reduced food intake during the type 2 inflammatory response.

We also asked whether spontaneous behaviours are reliable measures of symptoms induced by type 2 inflammation (Extended Data Fig. 10a,b). Grooming behaviour increased in female mice and was diminished in PFTox(+), ChAT-deficient (Vil$^{Cre}$;Chat$^{flox/flox}$) or tuft-cell-deficient (Pou2f3$^{−/−}$) mice (Extended Data Fig. 10a,b). However, this phenotype was observed only in females, which might reflect greater sensitivity of the EC cell–mucosal afferent circuit[4,37] and/or more robust amplification of type 2 immune response in female versus male mice[38]. Other behaviours, including locomotion and rearing, were not consistently affected across genotypes, and abdominal mechanical sensitivity and nesting behaviour did not change with type 2 inflammation (Extended Data Fig. 10c,d). Thus, whether reduced food intake reflects nausea, abdominal discomfort or other aversive states is unclear.

## Discussion

Gastrointestinal symptoms during parasite infection have been vaguely attributed to tissue damage and alterations in the gut microbiota[39,40]. Although previous studies have identified aspects of neuro-immune communication between enteric neurons and type 2 innate lymphoid (ILC2) cells[41–43], these circuits have been implicated in the modulation of type 2 immune responses and control of worm burden, rather than initiation of protective or aversive behaviours. Our investigation of paracrine communication between tuft and EC cells now reveals a direct link between sensory and immune systems that alters food intake through the gut–brain axis. Furthermore, we propose that the spatio-temporal dynamics of this interaction can account for the increased severity of symptoms as parasitic infections progress from initial colonization to

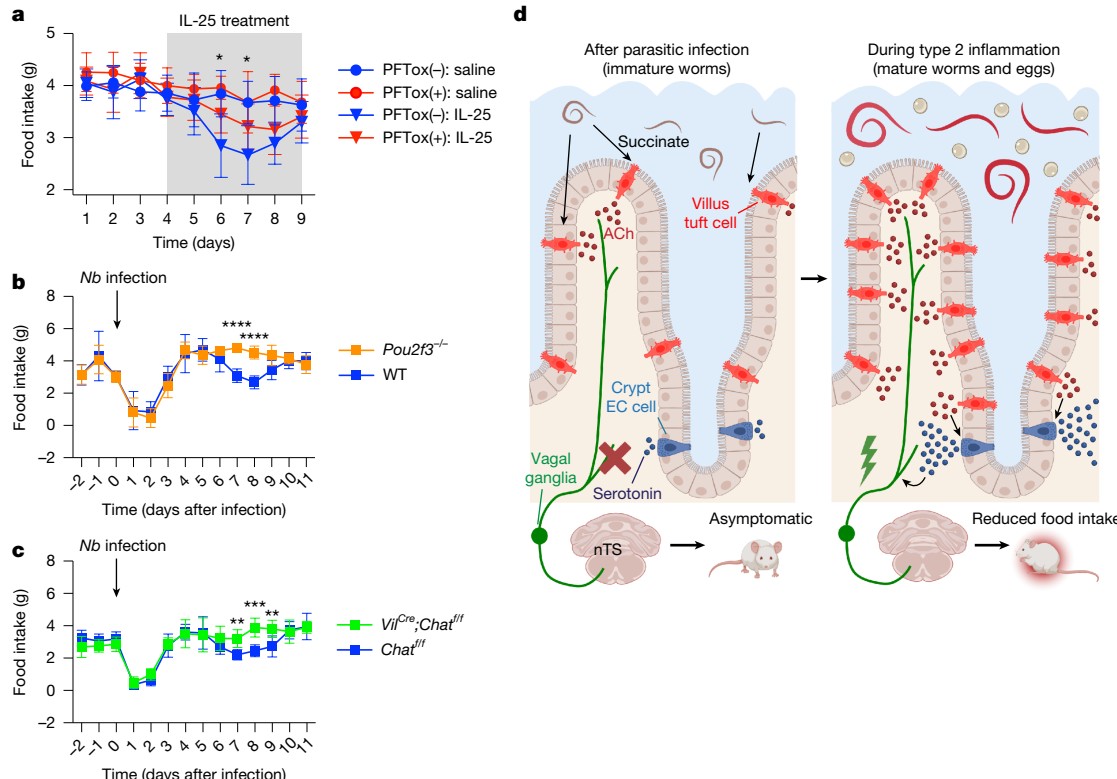

**Fig. 6 | The tuft cell–EC cell–vagal axis suppresses food intake. a**, Changes in daily food intake in PFTox(+) and PFTox(−) control mice with type 2 immune activation. After three-day baseline measurements, mice received IL-25 or saline daily for six days. Mean + s.d., two-way ANOVA with repeated measures (Holm–Šídák post hoc); $P = 0.0165$ and $0.0380$ (PFTox(−): IL-25 versus PFTox(+): IL-25) on days 6 and 7, respectively. $n = 11$ (PFTox(−): saline), 10 (PFTox(+): saline), 13 (PFTox(−): IL-25) and 15 (PFTox(+): IL-25) mice. **b,c**, Changes in food intake in $Pou2f3^{-/-}$ (**b**) and $Vil^{Cre};Chat^{flox/flox}$ (**c**) mice with $Nb$ infection. Mean ± s.d., two-way ANOVA with Holm–Šídák's multiple comparisons; $P < 0.0001$ on days 7 and 8 (WT versus $Pou2f3^{-/-}$), $P = 0.0026, 0.0002$ and $0.0091$ on days 7, 8 and 9, respectively ($Chat^{flox/flox}$ versus $Vil^{Cre};Chat^{flox/flox}$). $n = 10$ mice. **d**, Left, at first,

parasite-derived succinate stimulates transient ACh release from villus tuft cells. This limited ACh minimally engages crypt EC cells, resulting in levels of serotonin release that are insufficient to activate vagal neurons, and thus keeping hosts asymptomatic. Right, during type 2 inflammation, tuft cell hyperplasia and prolonged ACh release occur throughout the intestinal epithelium. This sustained ACh signalling robustly activates crypt EC cells, triggering substantial serotonin release that stimulates vagal afferents. The resulting vagal activation signals to the nTS, ultimately suppressing food intake. $^*P < 0.05$. $^{**}P < 0.01$, $^{***}P < 0.001$, $^{****}P < 0.0001$. NS, not significant. Illustration in **d** created in BioRender; Touhara, K. https://BioRender.com/6b8p2yr (2026).

established type 2 inflammation[44] (Fig. 6d). This neuro-immune cascade involves two consecutive paracrine interactions within the elaborate architecture of the gut: first, cholinergic signalling between tuft and EC cells; and second, serotonergic transmission between EC cells and sensory nerve fibres (with a potential component also involving direct cholinergic activation of primary afferents). In this two-step cascade, transmitter release must achieve a level that is sufficient to trigger the next step. After colonization, cholinergic communication is at first limited by three factors: sparse tuft cells (quantity); spatial separation between villus tuft and crypt EC cells (distance); and the transient nature of the signal (time). This results in minimal activation of EC cells and sensory nerves (Fig. 6d, left). If parasites are expelled early—for instance, through ACh-stimulated fluid secretion[14,15]—the infection will resolve without engaging nociceptive neurons. However, as parasites mature and begin egg-laying, these constraints (quantity, distance and time) will be overcome through an established type 2 immune response. This amplifies the detection–effector circuit sufficiently to exceed the threshold required for activation of EC cells and downstream nociceptive sensory circuits, ultimately suppressing feeding behaviour (Fig. 6d, right).

Parasite infections elicit a range of nocifensive symptoms, including abdominal discomfort and nausea[12,13]. Consistent with these clinical observations, we found that type 2 immune activation significantly decreased food intake in a tuft-cell- and ACh-dependent manner, and that this effect was delayed and attenuated by inhibiting the release of serotonin from EC cells. Together with nTS involvement,

these results are likely to reflect gastrointestinal discomfort or diminished appetite. However, precise behavioural assessment of visceral sensations—particularly from the small intestine—remains challenging in rodents, and even in people[45,46]. We also note that our study does not exclude the contribution of other pathways to overall visceral symptoms during parasitic infections, such as ILC2–enteric neuron signalling[41–43] and direct activation of cortical neurons by IL-25 (ref. 47). However, our findings expand researchers' basic understanding of how mucosal sensory circuits modulate physiological responses to inflammation and infection, and provide mechanistic insights that could be used in the development of therapeutic or diagnostic tools to treat or assess parasite-associated gastrointestinal symptoms. In this regard, understanding how intestinal tuft cells release ACh in the absence of synaptic-vesicle machinery and electrical excitability might reveal further unique aspects of this pathway that could be targeted to modulate sensory responses to parasitic infections. However, this will depend not only on the exact mechanism that underlies this unusual secretory process, but also on the extent to which it is used by tuft cells in other visceral organs, such as the respiratory tract and gall bladder[48–50].

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

## Methods

### Mice

All experimental procedures were conducted in accordance with guidelines approved by the Institutional Animal Care Committees at the University of California, San Francisco (UCSF) and the South Australian Health and Medical Research Institute (SAHMRI) and aligned with the NIH and NHMRC Guide for the Care and Use of Laboratory Animals, respectively. Mice of both sexes, aged 8–16 weeks, were used, and given ad libitum access to standard lab chow and sterile water. They were housed in a controlled environment under a 12-h light–dark cycle. TRPM5-GFP mice were used to visualize tuft cells (gift from R. F. Margolskee). $Pou2f3^{-/-}$ mice (Jackson Laboratory, strain 037040) were used to knock out tuft cells. For conditional knockout of $Chat$ in intestinal tuft cells, $Vil^{Cre}$ mice (Jackson Laboratory, strain 021504) were crossed to $Chat^{flox}$ mice (gift from J. Chan). For gGRAB$_{5-HT3.0}$ sensor imaging, $Vil^{Cre}$ mice were crossed to the $Rosa26^{gGRAB-5-HT3.0-P2A-jRGECO1a}$ reporter line. GCaMP imaging in organoids used $Tac1^{Cre}$ mice (Jackson Laboratory, strain 021877) crossed with $Polr2a^{GCaMP5g-IRES-tdTomato}$ mice (gift from L. Jan, Jackson Laboratory, strain 024477). To inhibit serotonin release from EC cells, we expressed the tetanus toxin in EC cells using the Cre- and Flp-dependent tetanus toxin light-chain reporter mouse, $RC::PFTox$ (gift from S. Dymecki). Specifically, $ePet^{Flp}$ hemizygous/$RC::PFTox$ homozygous mice were crossed to $Tac1^{Cre}$ homozygous mice to produce $Tac1^{Cre};ePet^{Flp};PFTox$ mice. For nerve fibre recordings, $Scn10a^{Cre}$ mice (gift from W. Imlach, Jackson Laboratory, strain 036564) were crossed to the ChR2 reporter line ($Rosa26^{lsl-ChR2}$; Jackson Laboratory, strain 012569). Genetically modified mice or control mice (littermates or age-matched mice) were randomly selected for all behavioural experiments.

### Crypt cell isolation and organoid culture

Adult male $Tac1^{Cre};Polr2a^{GCaMP5g-IRES-tdTomato}$, TRPM5-GFP and $Pou2f3^{-/-}$ mice were used to generate intestinal organoids as previously reported[51]. Approximately 8-cm pieces of the ileum were used to establish TRPM5-GFP organoids. For $Tac1^{Cre};Polr2a^{GCaMP5g-IRES-tdTomato}$ organoids, the upper jejunum was used to avoid ectopic expression of $Tac1^{Cre}$ in the lower intestine. Organoids were maintained and passaged every six days in organoid growth medium (advanced Dulbecco's modified Eagle's medium (DMEM)/F12 supplemented with penicillin–streptomycin, 10 mM HEPES, GlutaMAX, B27 (Thermo Fisher Scientific), 1 mM $N$-acetylcysteine (Sigma), 50 ng ml$^{-1}$ mouse recombinant epidermal growth factor (Thermo Fisher Scientific), R-spondin 1 (10% final volume) and 100 ng ml$^{-1}$ mouse Noggin (Peprotech)).

### Cell lines

The R-spondin-1-expressing HEK293FT (ATCC) cells were maintained in DMEM, 20% fetal calf serum (FCS), 1% penicillin–streptomycin and 125 µg ml$^{-1}$ zeocin (Thermo Fisher Scientific) at 37 °C, 5% $CO_2$. Zeocin was removed after production of R-spondin 1 conditioned medium. HEK293T cells (ATCC) were grown in DMEM, 10% FCS and 1% penicillin–streptomycin at 37 °C, 5% $CO_2$ and transfected using Lipofectamine 3000 (Thermo Fisher Scientific) according to the manufacturer's protocol. For biosensor experiments, 200 ng pDisplay-gGRAB$_{ACh4h}$-IRES-mCherryCAAX or 200 ng pcDNA3-hM1R-P2A-GCaMP8m was transfected to HEK293FT cells in 24-well plates. For the 5-HT$_3$ biosensor experiment, 200 ng pcDNA3-5-HT$_{3A}$ and 20 ng pcDNA3-mApple were co-transfected to HEK293FT cells in 24-well plates.

### Induction of tuft cell hyperplasia

Tuft cell hyperplasia in organoids was induced by exposing organoids to 20 ng ml$^{-1}$ IL-4 (R&D Systems) in the growth medium for two days (days 3–5), followed by one day of growth in IL-4-free medium. The biosensor experiments were then performed on day 6. To induce tuft cell hyperplasia in mice, the mice received intraperitoneal injections of 500 ng of IL-25 (R&D Systems) on days 0, 1, 2 and 3. Tissues were subsequently collected for imaging on day 5.

### GCaMP imaging using intestinal organoids

Five days after passage, $Tac1^{Cre};Polr2a^{GCaMP5g-IRES-tdTomato}$ organoids were removed from Matrigel (Corning) and mechanically broken up with a 200-µl pipette. The organoid fragments were seeded onto Cell-Tak (Corning)-coated coverslips and placed in a recording chamber containing Ringer's solution (140 mM NaCl, 5 mM KCl, 2 mM CaCl$_2$, 2 mM MgCl$_2$, 10 mM D-glucose and 10 mM HEPES-Na (pH 7.4)). EC cells were identified by tdTomato expression. GCaMP imaging was performed with an upright microscope equipped with a Grasshopper 3 (FLIR) camera and a Lambda LS light source (Sutter). Organoids were maintained under a constant laminar flow of Ringer's solution applied by a pressure-driven microperfusion system (SmartSquirt, Automate Scientific). All pharmacological reagents were delivered by local perfusion. Acquired images were analysed with Fiji. Regions of interest (ROIs) were drawn around individual EC cells, and $\Delta F/F_0$ was calculated.

### M1R–GCaMP8m biosensor experiments

Tuft cell hyperplasia was induced in organoids as described above. In each well of 24-well plates, the organoid medium was replaced with 1 ml of Ringer's solution and incubated for 10 min to remove residual growth medium. Subsequently, the Ringer's solution was replaced with 300 µl fresh Ringer's solution and incubated for two hours, after which the supernatant was collected for imaging. For the preparation of M1R–GCaMP8m biosensor cells, HEK293FT cells transiently transfected with pcDNA3-M1R-P2A-GCaMP8m were dissociated with trypsin and plated onto 5-mm glass coverslips. After one hour, coverslips were moved to µ-Slide III 3D Perfusion chamber slides (ibidi). Imaging was performed using an upright microscope equipped with a Grasshopper 3 camera (FLIR) and a Lambda LS light source (Sutter Instrument). The entire area of each biosensor cell was used for the calculation of $\Delta F/F_0$ values. The organoid supernatant was manually applied with a 200-µl pipette. All images were analysed using Fiji software v.2.14 (NIH).

### gGRAB$_{ACh4h}$ biosensor experiments

HEK293FT cells transiently transfected with pDisplay-gGRAB$_{ACh4h}$-IRES-mCherryCAAX were dissociated with trypsin and washed once with Ringer's. The dissociated cells were plated on top of intestinal organoids. For biosensor experiments with isolated tuft cells and TRPM5-GFP organoids, individual HEK293FT cells were carefully lifted from coverslips and positioned 5 µm from a tuft cell using a glass pipette. In other gGRAB$_{ACh4h}$ biosensor experiments, signals were measured from biosensor cells surrounding the organoids. Imaging was performed using an upright microscope equipped with a Grasshopper 3 camera (FLIR) and a Lambda LS light source (Sutter Instrument). The entire area of each biosensor cell was used for the calculation of $\Delta F/F_0$ values. The bath solution was static to prevent the washout of endogenously released ACh, and 10 µM TRPM5 agonist 39 (ProbeChem), 10 µM NDNA (ProbeChem) and 10 mM succinate were applied manually with a 1,000-µl pipette. All images were analysed using Fiji.

### 5-HT$_3$ biosensor experiments

HEK293FT cells transiently transfected with pcDNA3-5-HT$_{3A}$ and pcDNA3-mApple were dissociated with trypsin and washed once with Ringer's. The dissociated cells were plated on top of intestinal organoids. Transfected cells were identified based on the mApple expression, and whole-cell configuration was achieved. While the membrane potential was held at −80 mV, individual HEK293FT cells were carefully lifted from coverslips and positioned 5 µm from an EC cell using a glass pipette. GCaMP imaging of organoid EC cells was performed using an inverted microscope equipped with a Grasshopper 3 camera (FLIR)

and a Lambda LS light source (Sutter Instrument). The bath solution was static to prevent the washout of endogenously released serotonin, and pharmacological agents were applied manually with a 1,000-µl pipette. All images were analysed using Fiji.

## Patch-clamp recordings

Recording was performed with pClamp software v.10.7 (Molecular Devices) using a Multiclamp 700A amplifier (Molecular Devices) connected to a Digidata 1550B digitizer (Molecular Devices). Patch electrodes (2–4 MΩ) were pulled from borosilicate capillaries (BF-150-110-10, Sutter Instrument). The intracellular solution consists of 140 mM K-aspartate, 13.5 mM NaCl, 1.6 mM $MgCl_2$, 0.09 mM EGTA and 9 mM HEPES-K (pH 7.35). Recordings were performed in Ringer's solution. For measuring voltage-gated calcium currents, the following solutions were used: (external) 140 mM NMDG, 10 mM $BaCl_2$, 10 mM HEPES and 12.5 mM glucose (pH 7.4); (internal) 140 mM $CsMeSO_4$, 2 mM $MgCl_2$, 0.1 mM $CaCl_2$, 10 mM HEPES and 5 mM EGTA (pH 7.4). TRPM5 agonist 39 (2 µM; ProbeChem) and 10 mM succinate reagents were delivered locally by gravity-driven microperfusion (Valvebank, Automate Scientific).

## Ex vivo serotonin sensor imaging

An approximately 1 cm piece of the jejunum was isolated from an 8–16-week-old $Vil^{Cre}$;$Rosa26^{gGRAB-5-HT3.0-P2A-jRGECO1a}$ mouse. For recordings from $Vil^{Cre}$;$Rosa26^{gGRAB-5-HT3.0-P2A-jRGECO1a}$;$Pou2f3^{-/-}$ or $Vil^{Cre}$; $Rosa26^{gGRAB-5-HT3.0-P2A-jRGECO1a}$;$Chat^{f/f}$ mice, genetically modified mice or control mice (littermates or age-matched mice) were randomly selected. The tissue was immediately filleted open along the mesenteric border, and the smooth-muscle layer was manually removed for tissue stability. Tissue segments were pinned down to a Sylgard-coated recording chamber, and imaged from the smooth-muscle side to observe crypts and from the luminal side to observe villi. Imaging was performed with a Leica SP8 confocal microscope with an HC APO L 20×/1,00 W objective and LAS X software (Leica Microsystems). The tissue was bath-perfused with bubbled room-temperature Krebs buffer (118 mM NaCl, 4.7 mM KCl, 2 mM $MgCl_2$, 2 mM $CaCl_2$, 1.2 mM $KH_2PO_4$, 25 mM $NaHCO_3$ and 10 mM D-glucose) at a rate of around 1 ml per min. All pharmacological reagents were diluted in Krebs buffer and bath-perfused with simultaneous manual application. For measuring baseline serotonin levels, the $gGRAB_{5-HT3.0}$ sensor was fully activated by bath-applied 20 µM serotonin, followed by the complete quenching of $gGRAB_{5-HT3.0}$ with 50 µM RS 23597-190 (Tocris), a $5-HT_4$ receptor antagonist. For baseline serotonin measurements after $Nb$ infection, mice were infected subcutaneously with 500 third-stage $Nb$ larvae (L3) and euthanized on day 9 for imaging. Acquired images were analysed with Fiji. ROIs were drawn around individual crypts or villi and $\Delta F/F_0$ was calculated and normalized to serotonin-activated maximum signals. AUC was calculated as $\sum$(Normalized $gGRAB_{5-HT3.0}$ $\Delta F/F_0$) for the duration of 3 min during baseline.

## Dissociation of tuft cells

The ileum from TRPM5-GFP mice was cut into approximately 3-cm segments, incubated in 10 ml of cold Dulbecco's phosphate-buffered saline (DPBS) with 2 mM EDTA on ice for 12 min, then transferred to 6 ml of warm DPBS with 2 mM EDTA and incubated at 37 °C for 8 min. To dissociate the epithelial layer, vigorous shaking was applied for 30–60 s. The dissociated epithelium was centrifuged and washed with DPBS containing 10% fetal bovine serum (FBS). The washed epithelium was digested in 10 ml digestion buffer (Hanks' balanced salt solution (HBSS) with 0.2 mg ml$^{-1}$ dispase II (Sigma) and 0.2 mg ml$^{-1}$ DNaseI (Sigma)) at 37 °C for 4 min, with vigorous shaking at 2-min intervals. The cells were then washed once with HBSS containing 10% FBS and 0.2 mg ml$^{-1}$ DNaseI, filtered through 70-µm and 40-µm strainers and resuspended in cold Ringer's solution supplemented with 5 µM Y-27632 (Sigma). Cells were kept on ice and used within two hours.

## Dissociation of EC cells

For GCaMP imaging, EC cells were isolated from the upper half of the small intestine of 8–16-week-old $Tac1^{Cre}$;$Polr2a^{GCaMP5g-IRES-tdTomato}$ mice. To isolate crypt and villus epithelial cells, intestinal tissue was cut into approximately 3-cm segments and filleted, and the villus epithelium was dissociated by gentle scraping with a glass coverslip. The remaining tissue was subsequently incubated in 10 ml of cold DPBS with 30 mM EDTA and 1.5 mM dithiothreitol (DTT) on ice for 20 min. The tissue was transferred to 6 ml of pre-warmed DPBS with 30 mM EDTA and incubated at 37 °C for 8 min, followed by 30–60 s of vigorous shaking to dissociate the crypt epithelium. The dissociated crypt and villi fractions were washed with 10 ml DPBS with 10% FBS and digested in 10 ml HBSS with 0.6 mg ml$^{-1}$ dispase II [Sigma] and 0.2 mg ml$^{-1}$ DNaseI (Sigma)) at 37 °C for 12 min, with vigorous shaking at 2-min intervals. The cells were then washed with 10 ml HBSS with 10% FBS and 0.2 mg ml$^{-1}$ DNaseI. For GCaMP imaging, dissociated crypt cells were filtered through 70-µm and 40-µm strainers, resuspended in DMEM supplemented with 10% FBS, B27 and 5 µM Y-27632 (Sigma) and plated onto glass coverslips precoated with 2.5% Matrigel solution. Three days after dissociation, healthy EC cells were identified by tdTomato fluorescence and characteristic polygonal or cone-shaped morphology.

## Brainstem tissue collection

Mice received an intraperitoneal injection of 500 ng IL-25 or saline every 24 h for a duration of 4 days. To minimize background FOS signals in the nTS region, mice were fasted overnight from day 3. On day 4, four hours after receiving the final injection, mice were euthanized for tissue collection. For helminth infection experiments, mice were euthanized for tissue collection 11 days after infection. Whole-body perfusion was performed for each mouse with 30 ml ice-cold DPBS (Thermo Fisher Scientific) followed by 20 ml ice-cold 4% paraformaldehyde (PFA). The whole brain and brainstem were then collected and post-fixed overnight at 4 °C in 4% PFA. After fixation, brainstems were dissected and washed thoroughly in PBS, followed by 30% sucrose dehydration at 4 °C overnight. The brainstems were then cryo-embedded in OCT (Sakura) for cryosections. Ten-micrometre sections were collected from Bregma −6.92 mm to −8.00 mm to cover the whole nTS region that receives extensive projection from the gut-innervating vagal afferents[30].

## Histology and immunostaining

Immunofluorescence in the intestine and brainstem was performed using 10-µm cryosections. For staining organoids, whole organoids were fixed in 4% PFA for 30 min at room temperature. Immunofluorescence was then performed with whole organoids or 7-µm cryosections. Blocking was performed with 5% w/v bovine serum albumin (BSA; Sigma), 5% normal serum corresponding to secondary antibody species and 0.3% Triton-X in PBS at room temperature for 30 min. Primary antibodies were incubated overnight at 4 °C at the indicated dilutions. The antibodies used were against DCAMKL1 (1:250, Abcam), GFP (1:300, Abcam) and FOS (1:300, Synaptic Systems). Secondary antibodies from Invitrogen (Alexa Fluor 647 goat anti-rabbit, Alexa Fluor 647 goat anti-rat, Alexa Fluor 568 goat anti-rabbit, Alexa Fluor 568 donkey anti-rat, Alexa Fluor 488 goat anti-chicken and Alexa Fluor 488 donkey anti-chicken) were incubated at a 1:500 dilution for two hours at room temperature at a 1:500 dilution. All the imaging was performed at the UCSF Center for Advanced Light Microscopy. Confocal images were captured on an inverted Nikon Ti microscope run using Micro Manager 2.0 Gamma[52], equipped with a Zyla 4.2 CMOS camera (Andor), piezo XYZ stage (ASI), CSU-W1 spinning disk with Borealis upgrade (Yokogowa/Andor), Spectra-X (Lumencor) and ILE 4 line Laser Launch (405/488/561/640 nm; Andor). Images were taken using a Plan Apo λ 20×/0.75 and Plan Apo VC 60×/1.4 Oil using lasers 405, 488, 561

and 647 nm and emission filters 447/60, 525/50, 607/36 and 645/65 for DAPI, GFP, RFP and Cy5, respectively. Large, stitched images were taken using the same set-up if necessary. Maximum-intensity projections were generated in Fiji.

## In situ hybridization

Cryosections (5 or 10 μm) were prepared as described above. RNA fluorescence in situ hybridization (RNA-FISH) was performed using a RNAscope Multiplex Fluorescent Detection Kit v.2 (Advanced Cell Diagnostics) according to the manufacturer's protocol. The following probes were used in this study: Mm-Dclk1-C2 (476631-C2), Mm-Chat (408731), Mm-Sucnr1 (437721), Mm-Fos-C2 (316921-C2), Mm-Npy (313321), Mm-Dbh (407851) and Mm-Nr4a2 (423351). Z-stack images were taken with a Nikon CSU-W1 spinning disk confocal microscope as described above (UCSF Center for Advanced Light Microscopy). Maximum-intensity projections were generated using Fiji (v.2.14).

## Brainstem FOS quantification

We selected the medial part of gut-innervating vagal afferents that project to the nTS (−7.32 mm to −7.76 mm) region for the quantification of activated neurons, because we found few FOS$^+$ signals in the rostral (Bregma −6.92 mm to −7.32 mm) or caudal (Bregma −7.76 mm to −8.00 mm) parts. Each data point represents one mouse, and the numbers of FOS$^+$ neurons were averaged across multiple slides per individual. At least 15 (PFTox study) or 8 (colocalization study and helminth study) slides from each individual were analysed. For colocalization analysis, we performed IL-25 injections on $Tac1^{cre}Rosa26^{GCaMPSg-IRES-tdTomato}$ mice and stained GFP to visualize $Tac1^+$ neurons. In IL-25–injected wild-type mice, we performed RNAscope co-staining of $Fos$ with $Dbh$, $Npy$ or $Nr4a2$ to visualize $Dbh^+$, $Npy^+$ and $Nr4a2^+$ neurons, respectively. CCK8 staining was used to visualize $Cck^+$ neurons. Comparable numbers of males and females were used in each study, with no sex differences detected.

## Food intake assay

For food intake measurements with IL-25 injections, 8–12-week-old mice were acclimated to metabolic cages (41700UB, Animalab) for 11 days. Genetically modified mice or control mice (littermates or age-matched mice) were randomly selected. A standard laboratory powder diet (5053, PicoLab Rodent Diet) was provided. Mice were adjusted to the cage environment for three days. After this adjustment period, body weight and food and water intake were measured every 24 h. After 3 days of baseline measurements, mice received an intraperitoneal injection of either 500 ng IL-25 or saline every 24 h for a duration of 6 days.

To measure food intake after acute tuft cell stimulation, 8–12-week-old mice were singly housed and acclimated to FED3.1 automated feeding devices (OEPS-7510, Open Ephys) for 2 days before experimentation. The FED3.1 system continuously monitored and recorded individual feeding events, providing precise measurements of food consumption. After the acclimation period, a two-day experimental protocol was implemented using a within-subject design. On day 1, mice received oral gavage administration of 200 μl saline as a vehicle control at 18:00, immediately before the onset of the dark cycle. Food intake was continuously monitored overnight for 12 h using the FED3.1 system. On the following day (day 2), the same mice received oral gavage administration of 200 μl of TRPM5 agonist 39 (100 μM dissolved in saline) at 18:00 to stimulate tuft cell activation. Food intake was again monitored overnight for 12 h after treatment.

For food intake measurements during $Nb$ infections, 10–13-week-old mice were singly housed and acclimated to controlled feeding devices (929102, Research Products International) for 2 days before infection. The feeding system was loaded with 20 mg precision pellets (F0071, BioServ) to enable accurate quantification of food consumption throughout the infection period. Infectious third-stage $Nb$ larvae (L3) were raised and maintained as previously described[53]. On day 0, mice were infected subcutaneously with 500 $Nb$ L3. Food intake was continuously monitored from the day of infection (day 0) to day 11 after infection using the controlled feeding devices. Daily food consumption was recorded as total pellet intake (g).

## Ex vivo nerve fibre recordings of mucosal sensory afferents innervating the small intestine

Jejunal recordings were performed in 12–18-week-old male and female wild-type and $Vil^{Cre};Chat^{flox/flox}$ mice, as previously described[5]. In brief, mice were euthanized by $CO_2$ inhalation, and a small section of the jejunum (around 2 cm) was removed along with the attached neurovascular bundle. Jejunum segments were opened longitudinally and pinned flat, mucosal side up, in a specialized organ bath consisting of two adjacent compartments. The jejunal compartment was perfused with Krebs solution (117.9 mM NaCl, 4.7 mM KCl, 25 mM $NaHCO_3$, 1.3 mM $NaH_2PO_4$, 1.2 mM $MgSO_4$, 2.5 mM $CaCl_2$ and 11.1 mM D-glucose), bubbled with carbogen (95% $O_2$, 5% $CO_2$) at a temperature of 34 °C. Krebs solution also contained 1 μM nifedipine (to suppress smooth-muscle activity) and 3 μM indomethacin (to suppress potential inhibitory actions of endogenous prostaglandins). The end of the neurovascular bundle, which contains the vagus nerve, was extended from the tissue compartment into the paraffin-oil-filled recording compartment and laid onto a mirror. The whole nerve bundle supplying the jejunal segment was then located within the neurovascular bundle, carefully cleaned away and placed on a platinum recording electrode. Action potentials generated within the jejunum travelled along the nerve fibres, through to the recording electrode and into a differential amplifier. These signals were filtered, sampled (20 kHz) using a 1401 interface (CED) run by Spike 2 software (v.5.18) and stored on a PC for off-line analysis.

## Optogenetic stimulation of jejunal mucosal afferents

$Scn10a^{Cre};Rosa26^{lsl-ChR2}$ mice, which express ChR2 in sensory afferents but not in EC cells, were used[4,5]. This transgenic line enabled us to optogenetically activate mucosal afferents in an EC-independent and mechanically independent manner, as previously described. Using the jejunal ex vivo afferent preparation described above, we recorded the action potentials generated by stimulating an approximately 3-mm$^2$ section of the jejunum with continuous light (470 nm) at increasing intensities (0.08–7 mW; 2 s exposure each intensity and a 10-s interval between exposures). Light was delivered using a High Power Fiber-Coupled LED Light Source (model BLS-FCS-0470-10) and Multimode Fiber Patchcords (numerical aperture: 0.39 NA; core size: 400 μm; FPC-0400-39-025MA-BP, Mightex). After this graded illumination protocol, we perfused (by gravity) the section of the jejunum receiving the light stimuli with 10 μM ACh, succinate (10 mM) or TRPM5 agonist 39 (10 μM). These agonists were perfused alone or after 10 min perfusion with (and still in the presence of) the 5-HT$_3$ antagonist alosetron (10 μM). Five minutes after perfusion of the compounds, we repeated the graded illumination protocol described above. Action potentials generated by light stimuli were analysed off-line using the Spike 2 wavemark function and discriminated as single units on the basis of a distinguishable waveform, amplitude and duration (CED). Data are expressed as either: (1) afferent activity induced by individual light stimuli (action potentials per s) or the compounds tested; or (2) light intensity threshold for action potential activation (mW mm$^{-2}$). Data were analysed to determine whether they were normally distributed using Shapiro–Wilk tests, with subsequent analysis using two-way ANOVA with two-tailed Wilcoxon matched-pairs rank test, or Kruskal–Wallis test with Dunn's multiple comparison (for more than two groups). For comparisons between two groups, two-tailed paired $t$-tests were used. $P < 0.05$ was considered statistically significant throughout.

## Behavioural assays

To induce tuft cell hyperplasia, wild-type, PFTox(+), *Pou2f3*[−/−] and *Vil*[cre];*Chat*[flox/flox] mice were injected intraperitoneally daily on four consecutive days with either sterile saline (vehicle) or 500 ng IL-25. In the morning on day 5, mice were transferred from their individually ventilated cages (IVCs) to a temperature-controlled test room (22 ± 1 °C) and allowed to acclimatize for at least 15 min before testing. Assessment of locomotor activity and spontaneous behaviour in mice was evaluated using a behavioural spectrometer (Behavior Sequencer, Behavioral Instruments and BiObserve). This consisted of a 40 × 40-cm square arena enclosed at a height of 45 cm, with an aluminium floor on vibration sensors and a ceiling centre-mounted camera. In addition, the spectrometer is equipped with a row of 32 infrared transmitters and receiver pairs embedded in the walls at a height of 6.5 cm and halogen strip lights illuminating the inside of the behavioural box.

For testing, two main groups of mice were studied: (1) mice orally gavaged with vehicle (200 μl saline) or the TRPM5 agonist (TRPM5 agonist 39, 200 μl bolus at 100 μM); or (2) mice intraperitoneally administered vehicle or IL-25. Mice were individually placed in the centre of the behavioural spectrometer and their behaviours were filmed, tracked and analysed by computerized video tracking software (Viewer, BiObserve) for a total duration of 20 min. Total distance travelled in the open field (cm), time spent in the central area (20 × 20 cm) (s), wall distance (cm) and time spent (s) in 23 different behavioural patterns, including grooming, orienting, rearing and forms of locomotion, were evaluated. All recording sessions were performed between 09:30 and 12:30 each day in an order counterbalanced for the experimental group (maximum of six mice per day). Experimenters were blinded for all behavioural tests.

## Abdominal electronic von Frey Hair testing

Mechanical allodynia was assessed in vivo using electronic Von Frey hairs (eVFH) in C57BL/6 (Jackson Laboratory) mice that were intraperitoneally injected with vehicle or IL-25. Abdominal withdrawal thresholds caused by gradually applied mechanical pressure to the lower abdomen were measured as previously described[54]. Before eVFH testing, on days 2 and 3 of intraperitoneal IL-25 and vehicle treatment, mice were habituated to the eVFH testing enclosure (clear plexiglas observation chamber, around 230 × 240 × 146 mm, divided into 6 compartments; BSBIOPVF, Panlab), mounted on an elevated wire mesh stand, for around 30 min each day. Habituation was designed to reduce the natural explorative behaviour animals display when they are introduced to a new environment. Therefore, on the eVFH testing day, mice will explore less, which facilitates probing the abdominal area with the eVFH filament.

On day 5, mice were individually placed into the eVFH testing enclosure mounted on the raised grid stand and allowed to acclimatize for a minimum of 15 min. Once mice were still and quiet, abdominal withdrawal thresholds were tested using a portable force transducer equipped with a spring tip (BSBIOEVF4S, Panlab) recording device with corresponding software (BIOCIS Force Ramp software; for automatic recording of results on a PC via an RS232/USB port), and a remote foot switch. The eVFH spring tip, mounted on the handheld force transducer, was gently lifted from below to the lower abdomen of the mouse, and force was gradually applied until the abdomen was withdrawn. The maximum force applied (in grams; with a resolution of 0.1 g) that elicited the abdominal withdrawal was recorded as the individual withdrawal threshold value. Abdominal withdrawal thresholds were measured in the same mouse five times, and the results were expressed for each mouse as a mean abdominal withdrawal threshold. All measurements were performed by the same investigator to ensure consistency. At the end of testing, mice were placed in their home cage and returned to the IVC rack in the testing room to recover for the subsequent assessment of nest building, which was performed as an overnight test (as described below). Experimenters were blinded for all tests.

## Assessment of nest building

Nesting behaviour is a natural activity in rodents, which is important for providing shelter and heat conservation. This animal instinct can be used to assess wellbeing, because animals in discomfort will be less inclined to build nests or maintain them[55,56]. Nesting behaviour was assessed in vivo in mice treated intraperitoneally with either vehicle or IL-25 for five consecutive days to induce tuft cell hyperplasia. In brief, mice were kept in their home cage (individually housed from previous testing) with their original wood chip bedding and food, but all environmental enrichment items (tunnels, tissues and existing nesting material) were removed. A fresh, pre-weighed nestlet, consisting of an approximately 5-mm-thick pressed cotton square (50 mm × 50 mm, Able Scientific, Australia) weighing about 2.5–3 g, was placed in the cage and mice were left undisturbed overnight from 16:00 until 09:00 the following day. In the morning of day 6, the nest quality was assessed manually and scored according to a five-point nest-rating scale protocol[55]. Score 1: nestlet not noticeably touched (more than 90% intact); score 2: nestlet partially torn (50–90% remaining intact); score 3: nestlet mostly shredded, with material broadly spread in cage without clear nest area (less than 50–10% remaining intact); score 4: nestlet more than 90% torn and a nest area is present in a corner of the cage (nest is flat with less than 50% of the circumference built up higher than the mouse's body); score 5: more than 90% of the nestlet is torn and the nest has a clear crater with high fluffy walls over more than 50% of the circumference. If the score fell between two scores (for example, scores 3 and 4), the score was split, and a mean value was noted (for example, 3.5). In addition to evaluating the quality of the nest, any unshredded nestlet material left (intact nestlet pieces weighing more than 0.1 g) after a bout of nesting was weighed, and the percentage of nestlet material shredded (% nestlet shredded) was determined for each mouse. Experimenters were blinded for all tests.

## Retrograde tracing of mucosal afferents from the small intestine

Adult male C57BL/6 mice (Jackson Labs) aged 14–19 weeks were used for vagal ganglia collection. Mice were housed in groups (two to five mice per cage) in a specific and opportunistic pathogen-free facility, fed a Jackson lab diet, provided with environmental enrichment (shelter, nesting material and so on) and had normal immune status. Retrograde tracing using cholera toxin subunit B (CTB, 0.5%) directly conjugated to Alexa Fluor AF594 (for calcium imaging) or Alexa Fluor AF488 (for PCR with reverse transcription; RT–PCR) (C234777 and C22841, Invitrogen, Thermo Fisher Scientific) was performed from the lumen of the proximal small intestine. A small aseptic abdominal incision was made in mice anaesthetized with isoflurane (2–4% in 0.6 l min⁻¹ oxygen). The proximal small intestine was located, and 5-μl injections were made through the intestinal wall into the lumen at three to four sites covering 5 cm of the proximal small intestine. The tracer was expelled completely before the withdrawal of the needle back through the intestine wall. Injections were made with a 30-gauge needle (HAMC7803-07, point style: 4 (10–12); Hamilton Company, Bio-Strategy) attached to a Hamilton 5-μl syringe (HAMC7634-01 5 μl 700 series RN syringe; Hamilton Company, Bio-Strategy). The intestinal walls were gently rubbed together using cotton tip applicators to distribute the tracer throughout the lumen and the abdominal incision was sutured closed. Analgesic (buprenorphine, 0.1 mg per kg) and antibiotic (ampicillin, 50 mg per kg) were administered via subcutaneous injection as mice regained consciousness. Mice were then housed individually and closely monitored for four days before vagal ganglia collection for downstream cell picking for single-cell RT–PCR or Ca²⁺ imaging studies.

## Cell culture for Ca²⁺ imaging and single-cell RT–PCR of small-intestine mucosal-traced vagal neurons

After mucosal tracing, vagal ganglia were isolated from three male C57Bl/6J mice and enzymatically dissociated as previously described[5]. Ganglia were placed in 3 ml of magnesium- and calcium-free HBSS

(Gibco) with 12 mg of collagenase II (Gibco, Thermo Fisher Scientific) and 14 mg of dispase (Gibco) for 30 min at 37 °C. The enzyme solution was then replaced with 3 ml HBSS containing 12 mg collagenase II for an additional 10 min at 37 °C. Ganglia were washed twice with HBSS and then triturated through a fire-polished Pasteur pipette until a single-cell suspension was achieved. The cell suspension was centrifuged at $50g$ for one minute and the cell pellet was resuspended in DMEM (25 mM glucose, 1 mM pyruvate; Gibco) containing 10% FCS (Invitrogen), 1× Gluta-MAX (Gibco), 1× MEM non-essential amino acids (Gibco), 1× penicillin–streptomycin (Invitrogen) and 100 ng ml$^{-1}$ NGF (Merck). Neurons were spot-plated on coverslips coated with poly-D-lysine (Merck, 100 μg ml$^{-1}$) and laminin (Merck, 18 μg ml$^{-1}$) and maintained at 37 °C in 5% $CO_2$.

## Single-cell RT–PCR of mucosal-traced vagal ganglion neurons

CTB-traced neurons were manually picked using a micromanipulator under a fluorescent microscope. Cells were under a continuous slow flow of RNA/DNase-free PBS to reduce potential contamination. After picking a traced cell, the glass capillary was broken into a tube containing 9 μl lysis buffer with 1 μl DNaseI (Single Cell-to-CT qRT-PCR kit; Thermo Fisher Scientific) and further processed according to the manufacturer's instructions. A bath control was taken and analysed from every coverslip along with the other samples. PCR was performed according to the manufacturer's instructions using TaqMan Gene Expression master mix (Thermo Fisher Scientific) for 40 cycles. A target was defined to be present when a typical amplification curve was produced at a cycle threshold (ct) of fewer than 32 cycles. Predesigned Taqman probes were purchased from Thermo Fisher Scientific (*Chrna1*; Mm00431629_m1, *Chrna2*; Mm00460630_m1, *Chrna3*; Mm00520145_m1, *Chrna4*; Mm00516561_m1, *Chrna5*; Mm00616329_m1, *Chrna6*; Mm00517529_m1, *Chrna7*; Mm01312230_m1, *Chrna9*; Mm01221611_m1, *Chrna10*; Mm01274155_m1, *Chrnb1*; Mm00680412_m1, *Chrnb2*; Mm00515323_m1, *Chrnb3*; Mm00532602_m1, *Chrnb4*; Mm00804952_m1, *Chrm1*; Mm01231010_m1, *Chrm2*; Mm01167087_m1, *Chrm3*; Mm01338410_m1, *Chrm4*; Mm00432514_s1, *Chrm5*; Mm01701883_s1, *Tubb3*; Mm00727586_s1, *Gfap*; Mm01253033_m1). Non-template controls and a no-RT control were included. Cells with no *Tubb3* (four cells) or positive *Gfap* (one cell) expression were excluded from analysis. A total of 48 cells were picked and analysed with 5 cells excluded.

## Calcium imaging of small-intestine mucosa-innervating vagal ganglia neurons

Retrogradely traced vagal ganglion neurons were isolated from adult mice as previously described[5]. In brief, four days after retrograde tracing, mice were euthanized by $CO_2$ inhalation, and vagal ganglia were surgically removed and digested as described above. Neurons were spot-plated onto coverslips coated with poly-D-lysine (800 mg ml$^{-1}$) and laminin (20 mg ml$^{-1}$) and maintained at 37 °C in 5% $CO_2$. After 24 h in culture, neurons were loaded with 2.5 μM Fura-2-AM (Thermo Fisher Scientific) and 0.02% (v/v) pluronic acid for 30 min at room temperature in Ringer's solution (NaCl 140 mM, KCl 5 mM, $CaCl_2$ 1.25 mM, $MgCl_2$ 1 mM, glucose 10 mM and HEPES 10 mM, pH 7.4). After a brief wash, coverslips were transferred to a recording chamber filled with Ringer's solution at room temperature (around 22 °C). Retrogradely traced vagal ganglion neurons were identified by the presence of the AF488 tracer and viability was verified by responses to 40 mM KCl. Fura-2-AM fluorescence was measured at 340-nm and 380-nm excitation, and 530-nm emission was measured using an Olympus IX71 microscope in conjunction with a Sutter Lambda 10-3 wavelength switcher and the Chroma filter set no. 49011 (ET480/40x (Ex), T510lpxrxt (BS), ET535/50m (Em)). Fluorescence images were obtained every 5 s, using a 4× objective with a monochrome CCD camera (Retiga ELECTRO). Images were taken at baseline and after administration of *m*-chlorophenylbiguanide hydrochloride (mCPBG, 10 μM, Tocris), acetylcholine chloride (Sigma Merck) (10 μM) and KCl (40 mM). Fluorescence signals from neuronal cell bodies were extracted using MetaFluor software (Molecular Devices).

ROIs were manually drawn around neuronal cell bodies and their fluorescence traces were extracted as the 340/380 ratio.

## Analysis of single-cell and single-nucleus RNA-sequencing datasets

For mouse gut epithelial cells, we downloaded processed single-cell RNA-sequencing data (GSE224223)[21] from the Gene Expression Omnibus (GEO) database, for which both tuft and enteroendocrine cell subtypes are well annotated. For the mouse nTS cell atlas, we downloaded processed single-nucleus RNA-sequencing data from wild-type mice (GSE200003)[34] from the GEO, where each population is well annotated. All expression profiling was performed using modified DimPlot or VlnPlot functions from Seurat 5.0.1. (Satija Laboratory, https://satijalab.org/seurat/) and ggplot2 on R v.4.4.1.

## Statistical analysis

Data were analysed with Prism (GraphPad). $n$ represents the number of mice, cells, crypts, villi, nerve fibres or independent experiments, as specified in the legends. Data were considered significant if $P < 0.05$. Statistical parameters are described in the figure legends. All significance tests were justified considering the experimental design, and we assumed normal distribution and variance, as is common for similar experiments. Sample sizes were chosen on the basis of the number of independent experiments required for statistical significance and technical feasibility.

## Reporting summary

Further information on research design is available in the Nature Portfolio Reporting Summary linked to this article.

## Data availability

All data generated or analysed during this study are included in the manuscript and its accompanying data. Reagents used are available upon request. Source data are provided with this paper.

## Code availability

Custom codes were not generated in this study.

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

**Acknowledgements** We thank J. Poblete for technical support; J. Bayrer, H. Ingraham and all members of the D.J. laboratory for discussions; D. Pearce and E. Takagi for sharing metabolic cages; J. Do for assistance with the excision of the mouse brainstem; D. F. Margolskee and L. Jan for sharing TRPM5-GFP mice; and J. Chan for sharing *Chat*^flox/flox mice. We appreciate the support from D. Larsen, K. Herrington and S Y. Kim in the UCSF Center for Advanced Light Microscopy (supported by Shared Instrumentation grant 1S10OD017993-01A1). Illustrations in Figs. 1c,i,j, 3a, 5d,f and 6d and Extended Data Figs. 6a,e and 8a,c were created using BioRender (https://biorender.com). This work was supported by NIH grants NS105038, NS113869 and DK135714 to D.J.; BRAIN Initiative grants 1U01NS113358 and 1U01NS120824 to Y.L.; AI026918 and HL107202 to R.M.L.; National Natural Science Foundation of China grant 31925017 to Y.L.; the New Cornerstone Science Foundation through the New Cornerstone Investigator Program and the Xplorer Prize to Y.L.; the Howard Hughes Medical Institute to R.M.L.; National Health and Medical Research Council (NHMRC) of Australia Investigator Leadership grant APP2008727 to S.M.B.; and NHMRC Ideas grant APP2029332 to J.C. K.K.T. was supported by a Damon Runyon Cancer Research Foundation Fellowship (DRG-2387-30) J.X. was supported by Larry L. Hillblom Fellowship grant 2024-A-044-FEL.

**Author contributions** Research design: K.K.T., Y.L., R.M.L., S.M.B. and D.J. Writing: K.K.T. and D.J., with input from all authors. Development of gGRAB_ACh4h: G.L. Development of gGRAB_5-HT3.0: F.D.

RNA-sequencing analysis: N.D.R. FOS staining and single-cell RNA-sequencing analysis: J.X. Metabolic assay: J.X. and K.K.T. Helminth infections and food intake measurements: H.-E.L. Nerve fibre recordings: J.C. Behavioural tests: J.C., M.B., T.O., D.N. and G.S. Mucosal tracing, calcium imaging and RT–PCR: A.M.H., S.G.-C. and M.B. All other experiments and analysis: K.K.T. Supervision of trainees: Y.L., R.M.L., S.M.B. and D.J. Project administration: K.K.T., Y.L., R.M.L., S.M.B. and D.J.

**Competing interests** The authors declare no competing interests.

**Additional information**
**Correspondence and requests for materials** should be addressed to Kouki K. Touhara or David Julius.

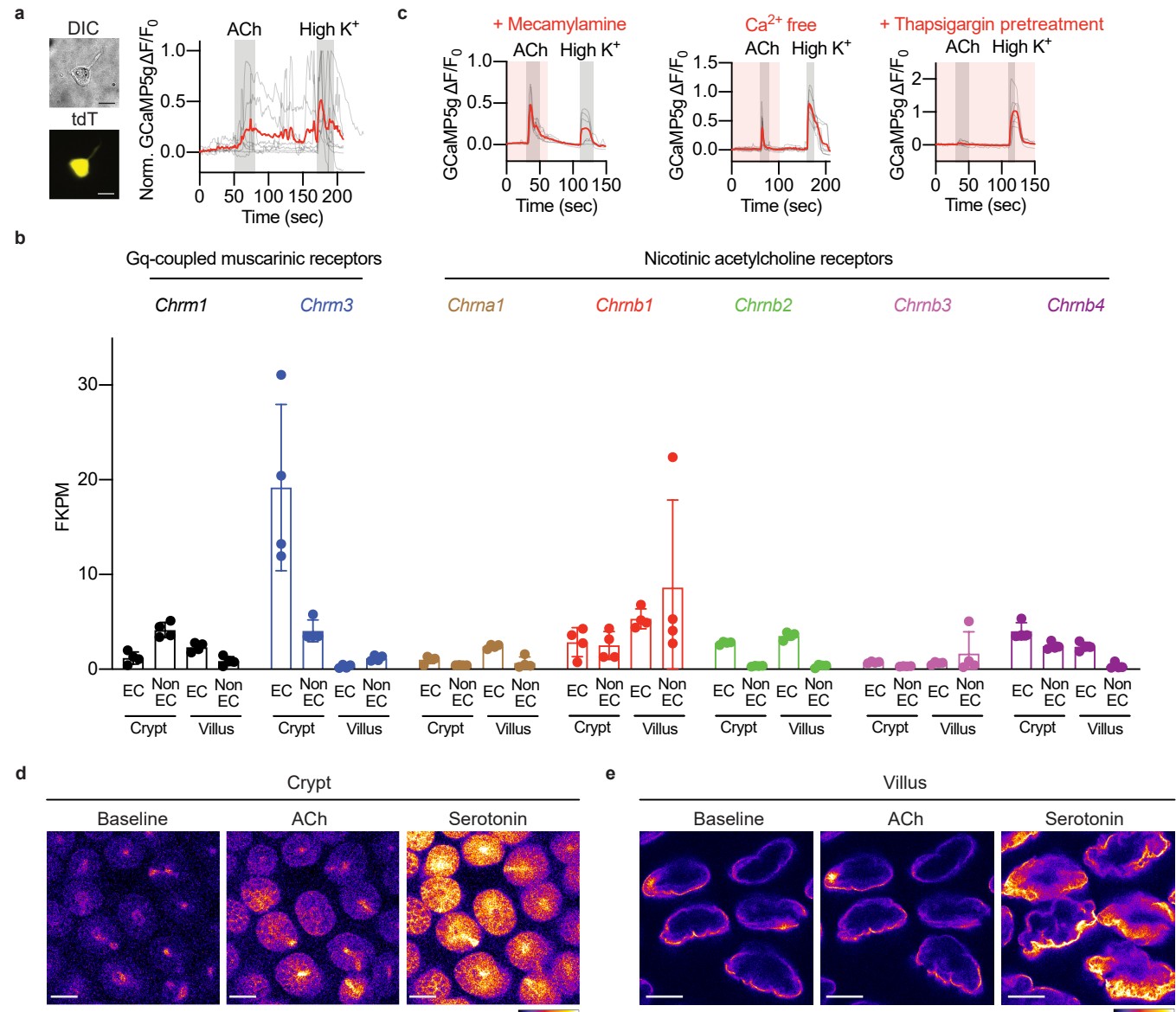

**Extended Data Fig. 1 | ACh activates crypt EC cells through mAChR. a,** Isolated EC cells respond to ACh. Crypt EC cells were dissociated from *Tac1^Cre^; Polr2a^GCaMP5g-IRES-tdTomato^* mice. Single tdTomato-expressing EC cell showing typical polygonal morphology (left). Single EC cells were stimulated with 10 μM ACh and 70 mM KCl (High K$^+$) as indicated. Individual EC cells in grey, average in red. $n = 7$ cells. **b,** mRNA expression profile of detected ACh receptor genes in crypt and villus EC and non-EC cells. Mean ± s.d., $n = 4$ samples. **c,** GCaMP5g response

of EC cells to 10 μM ACh, including conditions with 10 μM mecamylamine, absence of extracellular calcium and after a 20-min pre-treatment with 4 μM thapsigargin. Individual EC cells in grey, average in red. **d,e,** Examples of gGRAB$_{5-HT3.0}$ sensor imaging in crypts (**d**) and villi (**e**). Serotonin release was stimulated with 10 μM ACh. The gGRAB$_{5-HT3.0}$ sensor was fully activated with 20 μM serotonin at the end of recordings. Scale bars, 50 μm in **d** and 100 μm in **e**.

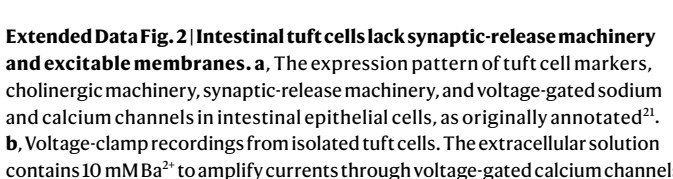

**Extended Data Fig. 2 | Intestinal tuft cells lack synaptic-release machinery and excitable membranes. a**, The expression pattern of tuft cell markers, cholinergic machinery, synaptic-release machinery, and voltage-gated sodium and calcium channels in intestinal epithelial cells, as originally annotated[21]. **b**, Voltage-clamp recordings from isolated tuft cells. The extracellular solution contains 10 mM Ba$^{2+}$ to amplify currents through voltage-gated calcium channels.

The membrane was held at −80 mV, and 20-mV steps were applied from −100 mV to 100 mV (Mean ± s.d., $n$ = 6 cells). **c**, An example of gGRAB$_{ACh4h}$ biosensor recordings. gGRAB$_{ACh4h}$-expressing HEK293FT cells were positioned adjacent to isolated tuft cells. Tuft cells release ACh in response to 10 mM succinate and 10 µM TRPM5 agonist. gGRAB$_{ACh4h}$ was fully activated with 20 µM ACh at the end of recordings for normalization. Scale bars, 10 µm.

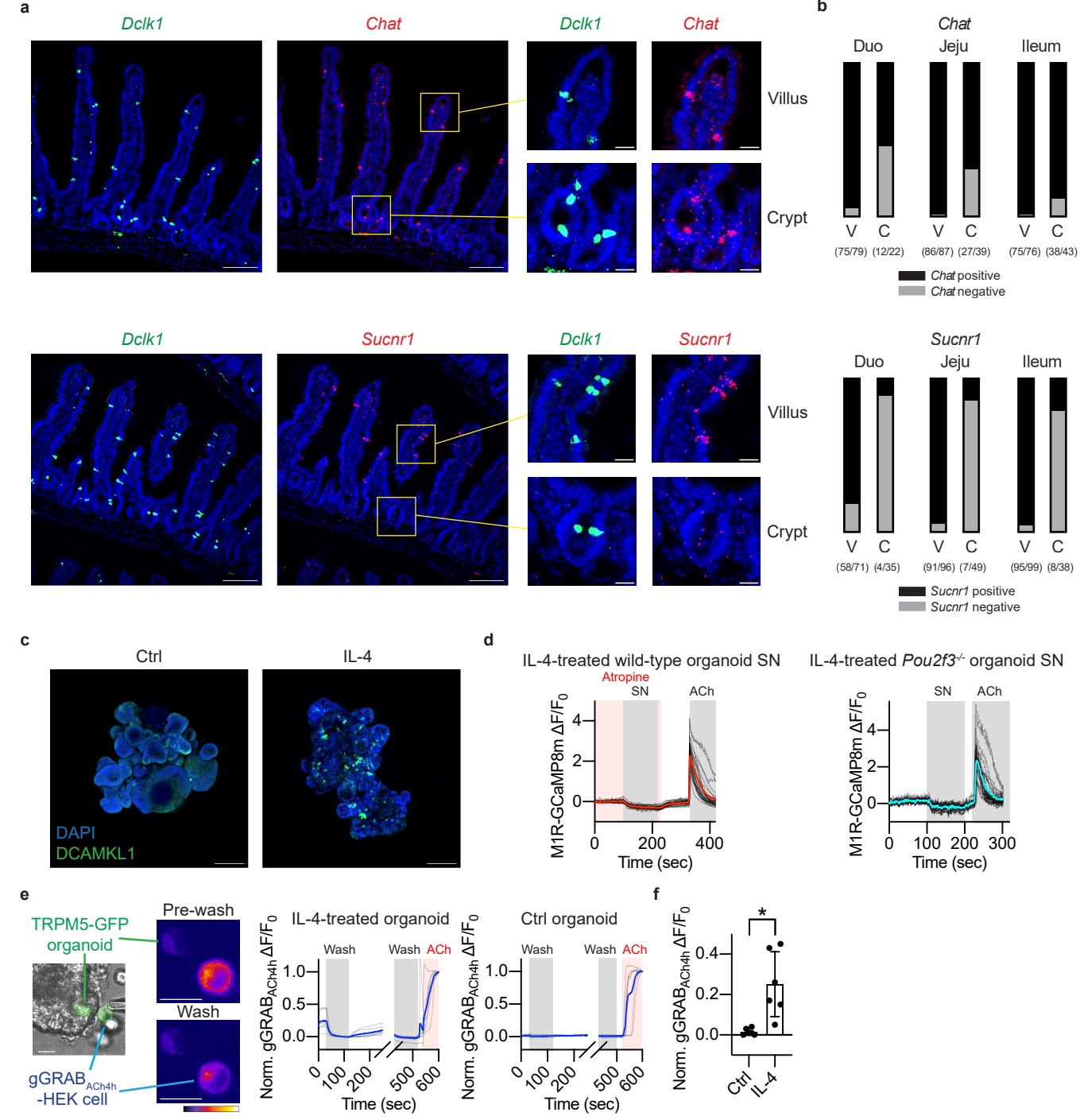

**Extended Data Fig. 3 | Differential expression of succinate receptors in tuft cells along the crypt–villus axis. a**, In situ hybridization of *Dclk1* (green: tuft cell marker), *Chat* (red in top panels), and *Sucnr1* (red in bottom panels: succinate receptor) in the small intestine. Scale bars, 100 μm (left) and 20 μm (right). **b**, *Sucnr1* expression is predominantly limited to villus tuft cells. Bars indicate the ratio of *Chat* (top) or *Sucnr1* (bottom)-positive cells/*Dclk1*-positive cells. $n = 3$ mice. **c**, Representative images of control and IL-4-treated organoids. Organoids were treated with either vehicle or 20 ng ml$^{-1}$ IL-4 for 2 days. Tuft cells were identified using anti-DCAMKL1 antibody. Scale bars, 100 μm. **d**, Supernatants from IL-4-treated wild-type organoids supplemented with 10 μM atropine or *Pou2f3*$^{-/-}$ organoids did not stimulate M1Rs in HEK293FT cells. M1R was fully activated with 10 μM ACh at the end of each recording. Individual M1R-GCaMP8m biosensor cells in grey, average in red or cyan. **e**, Representative gGRAB$_{ACh4h}$ biosensor experiment with TRPM5-GFP organoids. In each experiment, a HEK293FT cell expressing the gGRAB$_{ACh4h}$ sensor was positioned 5 μm away from a GFP-expressing organoid tuft cell. ACh was washed away with a local perfusion as indicated. gGRAB$_{ACh4h}$ was fully activated with 20 μM ACh at the end of each recording for normalization. Individual gGRAG$_{ACh4h}$ biosensor cells in grey, average in blue. Scale bars, 15 μm. **f**, Comparison of the normalized peak gGRAB$_{ACh4h}$ signals was conducted during the first 30 s of recordings obtained from the gGRAB$_{ACh4h}$-expressing HEK293FT cells adjacent to GFP(+) tuft cells in control and IL-4-treated TRPM5-GFP organoids. Mean ± s.d., two-tailed Welch's t-test. $P = 0.0151$. *$P < 0.05$. $n = 6$ cells.

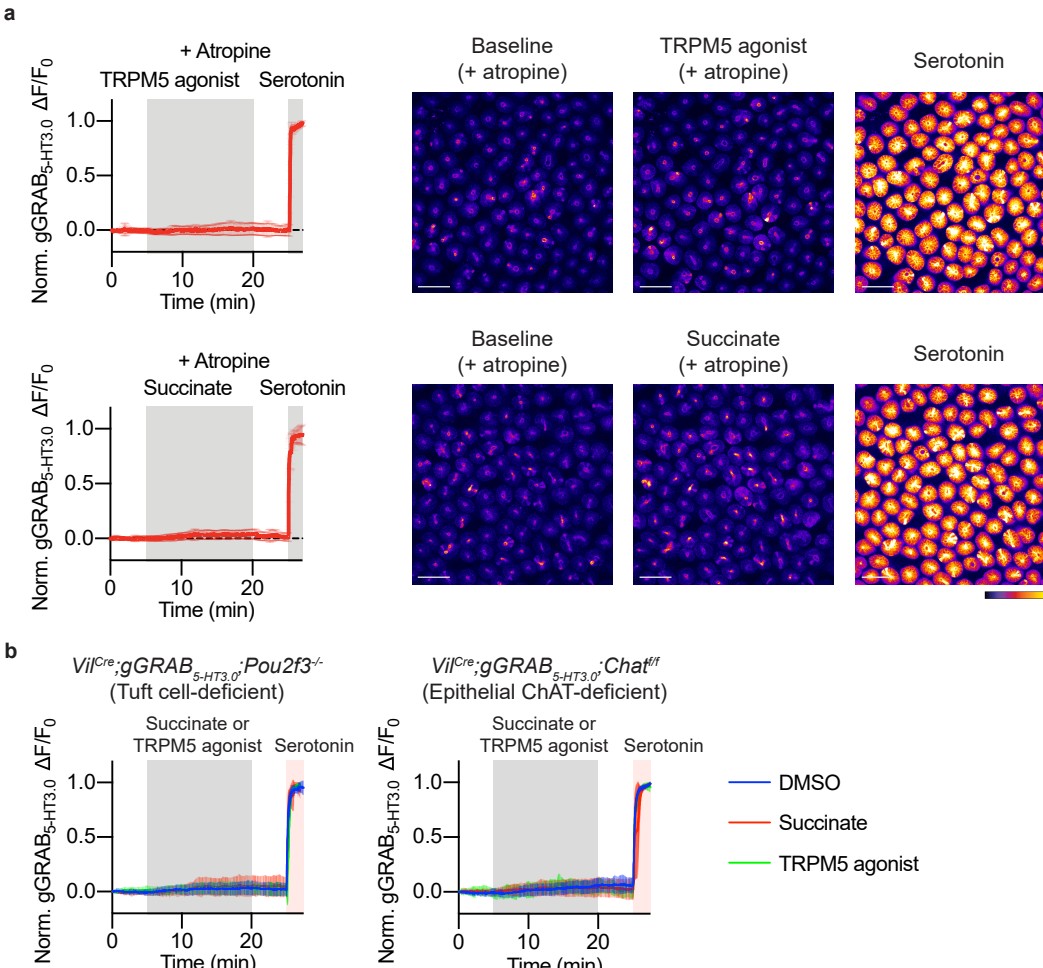

**Extended Data Fig. 4 | Tuft-cell-derived ACh mediates TRPM5-agonist- and succinate-induced serotonin release in crypts. a**, Serotonin release was monitored within crypts using ex vivo intestine preparations from $Vil^{Cre};gGRAB_{5-HT3.0}$ mice. Tuft cells were stimulated with 10 µM TRPM5 agonist or 10 mM succinate in the presence of 10 µM atropine. Average normalized $gGRAB_{5-HT3.0}$ signals ± s.d. (left) and corresponding fluorescence images at baseline, during stimulation, and after serotonin application (right). Scale bars, 100 µm. **b**, Serotonin release was monitored within crypts using ex vivo intestine preparations from tuft cell-deficient (left: $Vil^{Cre};gGRAB_{5-HT3.0};Pou2f3^{-/-}$) and epithelial-ChAT-deficient mice (right: $Vil^{Cre};gGRAB_{5-HT3.0};Chat^{flox/flox}$). Tuft cells were stimulated with 10 µM TRPM5 agonist or 10 mM succinate. Averaged normalized $gGRAB_{5-HT3.0}$ signals ± s.d. are shown.

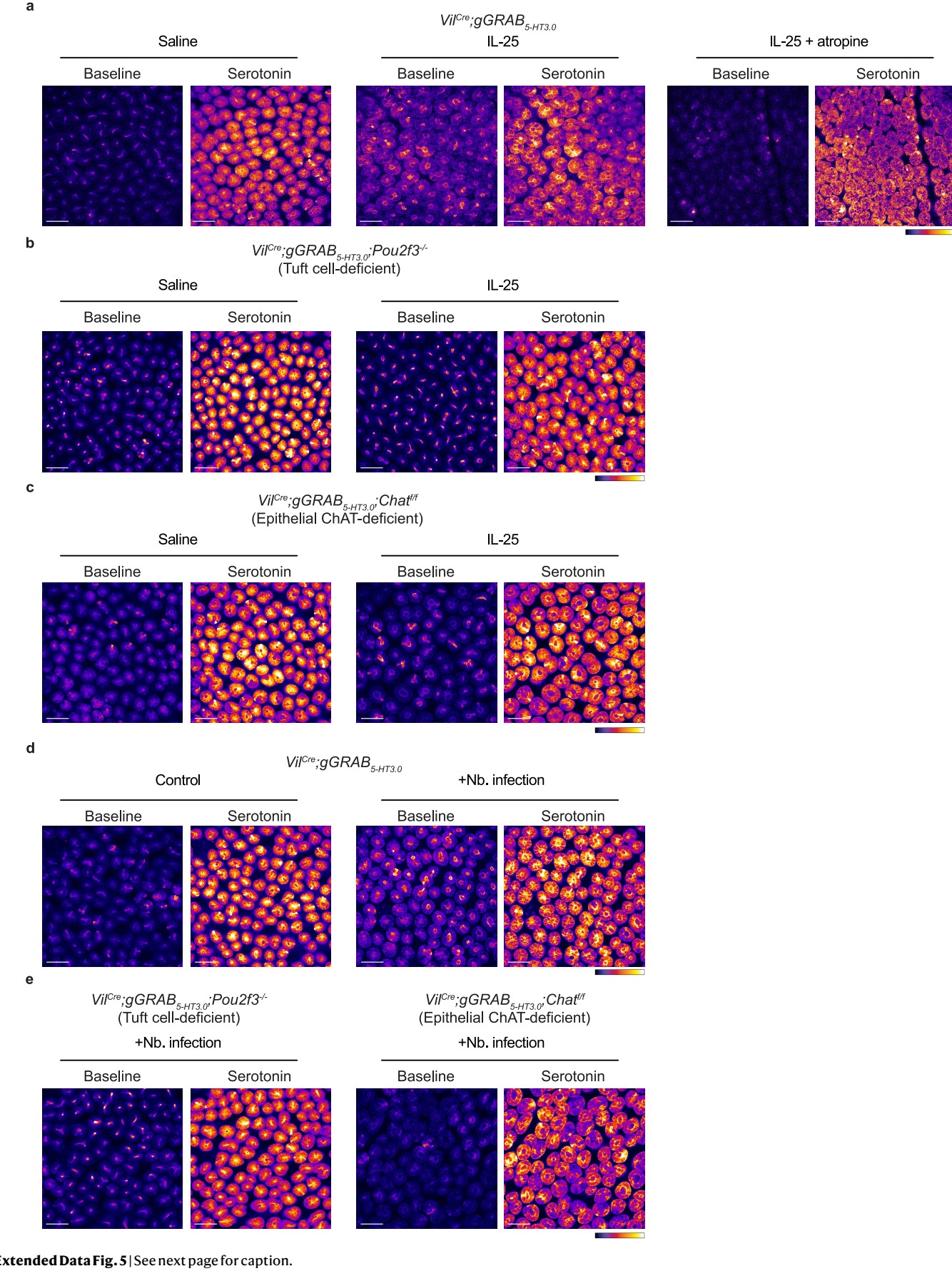

**Extended Data Fig. 5** | See next page for caption.

**Extended Data Fig. 5 | Tuft-cell-derived ACh activates crypt EC cells during type 2 inflammation. a**, Representative fluorescence images of crypts from $Vil^{Cre}$;$gGRAB_{5-HT3.0}$ mice treated with saline, IL-25, or IL-25 + atropine. Images show baseline and serotonin-activated states. **b**, Crypts from tuft-cell-deficient mice ($Vil^{Cre}$;$gGRAB_{5-HT3.0}$;$Pou2f3^{-/-}$) treated with saline or IL-25, showing baseline and serotonin-activated states. **c**, Crypts from epithelial-ChAT-deficient mice ($Vil^{Cre}$;$gGRAB_{5-HT3.0}$;$Chat^{flox/flox}$) treated with saline or IL-25, showing baseline and serotonin-activated states. **d**, Crypts from control and $Nb$-infected $Vil^{Cre}$;$gGRAB_{5-HT3.0}$ mice, showing baseline and serotonin-activated states. **e**, Crypts from $Nb$-infected tuft cell-deficient ($Vil^{Cre}$;$gGRAB_{5-HT3.0}$;$Pou2f3^{-/-}$) and epithelial-ChAT-deficient mice ($Vil^{Cre}$;$gGRAB_{5-HT3.0}$;$Chat^{flox/flox}$), showing baseline and serotonin-activated states. Scale bars, 100 μm.

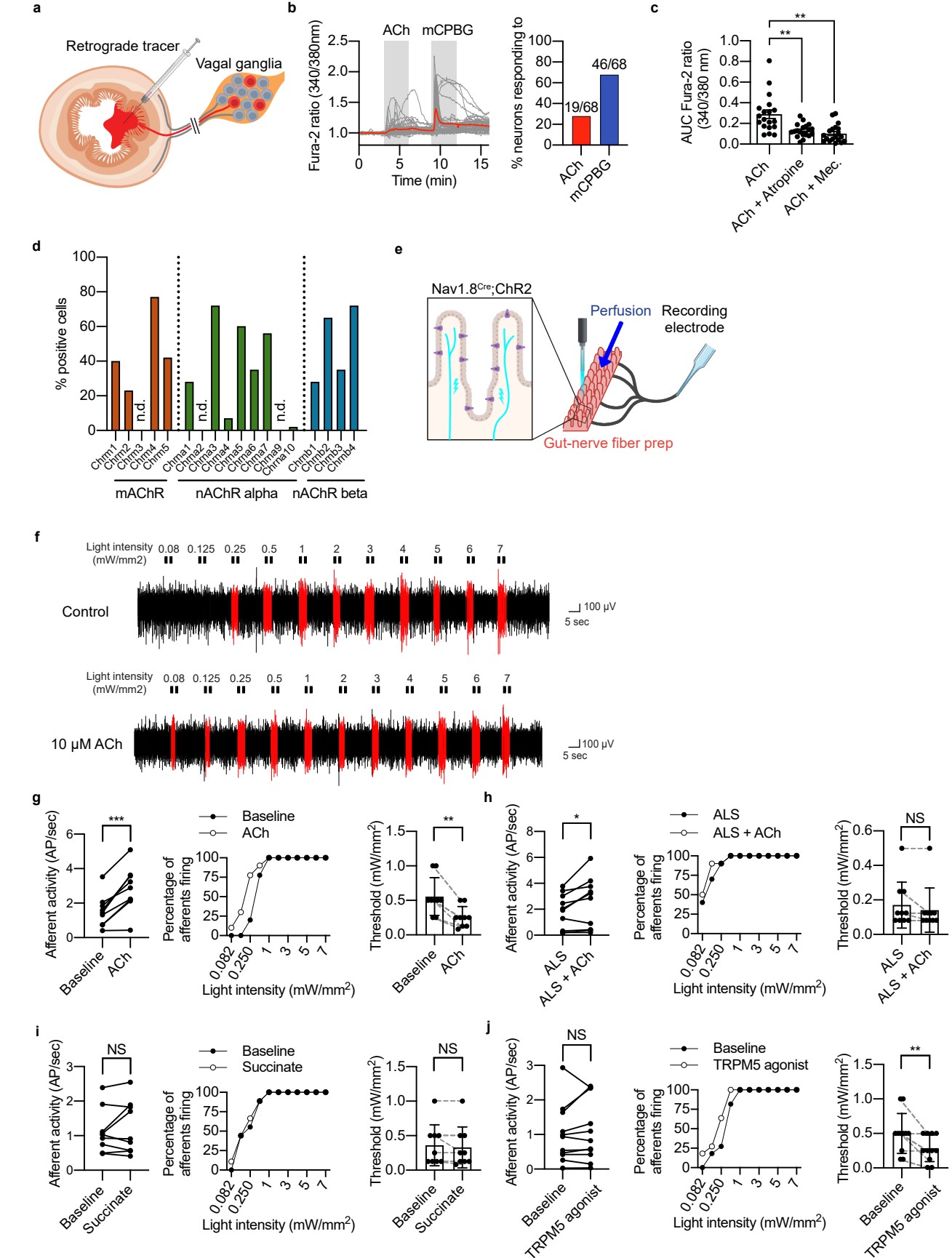

**Extended Data Fig. 6** | See next page for caption.

**Extended Data Fig. 6 | ACh stimulates mucosal vagal afferents through the tuft cell–EC cell pathway. a**, A retrograde tracer was injected into the small-intestine lumen to specifically trace mucosal afferents while excluding fibres innervating the muscle layer. **b**, (Left) Representative calcium imaging traces of dissociated small-intestine mucosa-innervating vagal neurons. 10 μM ACh and 10 μM mCPBG (5-HT$_3$ agonist) were applied as indicated. Individual cells in grey, average in red. (Right) The percentage of small-intestine mucosa-innervating vagal neurons responding to the applied agonists. $n$ = 68 neurons isolated from $n$ = 6 mice. **c**, Calcium imaging of traced mucosal vagal neurons showing responses to ACh (10 μM) alone, ACh with atropine (10 μM) and ACh with mecamylamine (10 μM). Data are presented as area under the curve (AUC) Fura-2 ratio (340/380 nm). Mean ± s.d., one-way ANOVA with repeated measures (Holm-Šídák post hoc); $P$ = 0.0066 (ACh vs. ACh + atropine), $P$ = 0.0036 (ACh vs. ACh + mecamylamine). $n$ = 19 isolated from $n$ = 4 mice. **d**, Single-cell RT–PCR analysis of retrogradely traced small-intestine mucosa-innervating vagal neurons. A target was defined to be present when a typical amplification curve was produced.

$n$ = 43 neurons isolated from $n$ = 4 mice. **e**, Schematic of ex vivo 'flat sheet' recordings from afferents innervating the jejunum of Na$_V$1.8-Cre;ChR2 (*Scn10a$^{Cre}$;Rosa26$^{lsl-ChR2}$*) mice. **f**, Representative jejunal afferent recordings with varying intensities of optogenetic stimulation of Na$_V$1.8+ nerve fibres in the absence and presence of 10 μM ACh. Red signals represent light-evoked action potentials. **g–j**, (left) The number of action potentials/sec (AP/sec) is compared before and after application of (**g**) 10 μM ACh, (**h**) 10 μM ACh + 10 μM alosetron, (**i**) 10 mM succinate, and (**j**) 10 μM TRPM5 agonist 39. Two-tailed paired *t*-test; $P$ = 0.0003, 0.026, 0.3422 and 0.4072, respectively. $n$ = 9, 10, 9 and 11 fibres. (middle) The percentage of afferents responding at indicated light intensities. (right) Activity thresholds of jejunal afferents before and after application of drugs described above. Mean ± s.d., two-tailed Wilcoxon matched-pairs rank test; $P$ = 0.0039, 0.25, 0.5, and 0.0078, respectively. $n$ = 9, 10, 9 and 11 fibres. *$P$ < 0.05. **$P$ < 0.01, ***$P$ < 0.001. NS: not significant. Illustrations in **a**,**e** created in BioRender; Touhara, K. https://BioRender.com/so6yfih (2026).

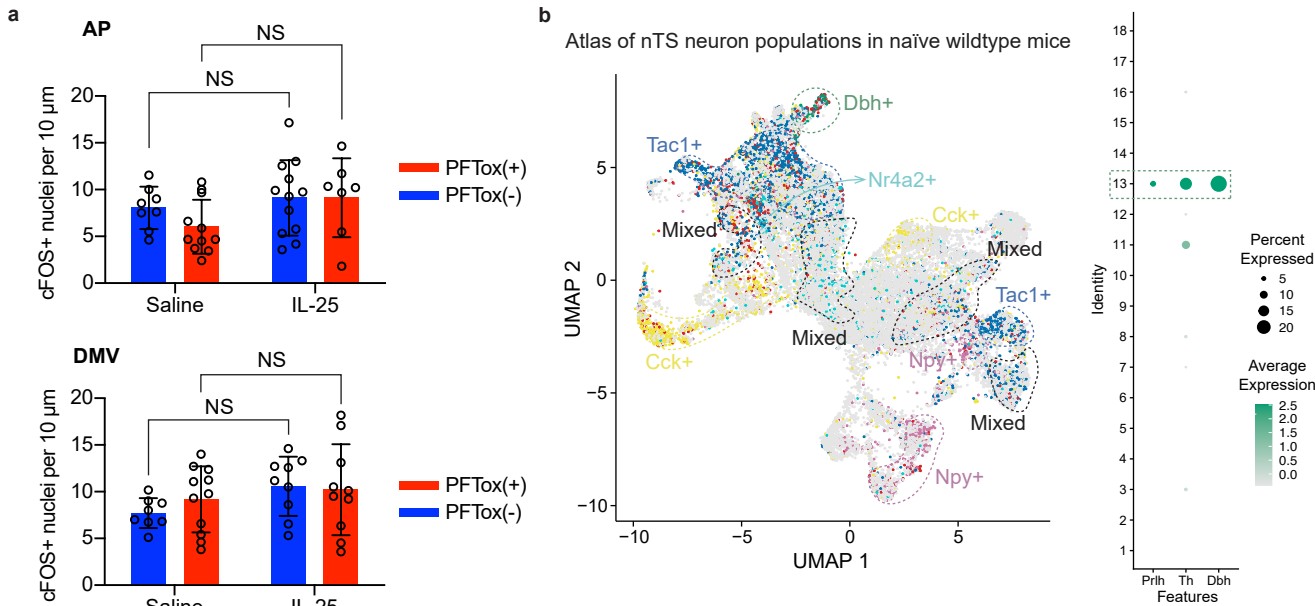

**a**

**AP**

NS

NS

cFOS+ nuclei per 10 µm

Saline    IL-25

■ PFTox(+)
■ PFTox(−)

**DMV**

NS

NS

cFOS+ nuclei per 10 µm

Saline    IL-25

■ PFTox(+)
■ PFTox(−)

**b** Atlas of nTS neuron populations in naïve wildtype mice

**c**

Quantification of cFOS+ neurons in medial nTS during type 2 immune response

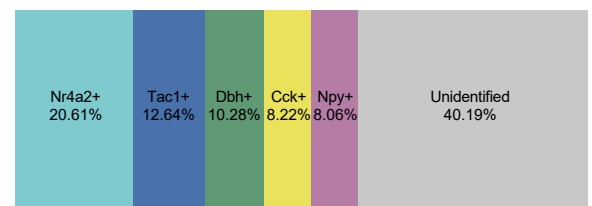

| Nr4a2+ 20.61% | Tac1+ 12.64% | Dbh+ 10.28% | Cck+ 8.22% | Npy+ 8.06% | Unidentified 40.19% |

**Extended Data Fig. 7 | Type 2 inflammation increases cFOS expression in the nTS but not in the AP and DMV. a**, No significant differences in the number of cFOS+ neurons were observed in either the AP (top) or DMV (bottom) between treatment conditions (saline-injected vs. IL-25-injected) or genetic groups (PFTox(+) vs. PFTox(−)). Analyses were performed on the same cohort of mice shown in Fig. 5j,k. Mean ± s.d., two-way ANOVA with multiple comparisons (Tukey's test). AP: $P = 0.9027$ (PFTox(−): saline vs. PFTox(−): IL-25). $P = 0.2669$ (PFTox(+): saline vs. PFTox(+): IL-25). $n = 8, 11, 12$ and 7 mice (bars from left to right). DMV: $P = 0.3595$ (PFTox(−): saline vs. PFTox(−): IL-25). $P = 0.9096$ (PFTox(+): saline vs. PFTox(+): IL-25). $n = 8, 11, 9$ and 10 mice (bars from left to right). AP, area postrema; DMV, dorsal motor nucleus. **b**, (left) Uniform manifold approximation and projection (UMAP) visualization of nTS neuronal populations from naïve

mice, using the published atlas dataset GSE200003. Marker-defined populations include *Dbh* (green), *Tac1* (blue), *Cck* (yellow), *Npy* (pink) and *Nr4a2* (turquoise; commonly co-expressed with Tac1). Clusters containing mixed neuronal identities are outlined in black ("Mixed"), and cells co-expressing multiple markers appear in red. Together, these markers encompass ~30% of nTS neurons in wild-type mice. (Right) Dot plot showing transcriptional overlap among *Prlh*, *Th* and *Dbh*, with the remaining Th+ neurons in cluster 11 covered by *Cck* (marker for cluster 11). **c**, Summary of cFOS+ nTS neuron composition during the type 2 immune response mediated by the tuft–EC–vagal axis described in this study. Shown are the quantifications of colocalization between cFOS and key neuronal marker genes. Quantification procedures are detailed in the Methods. NS: not significant.

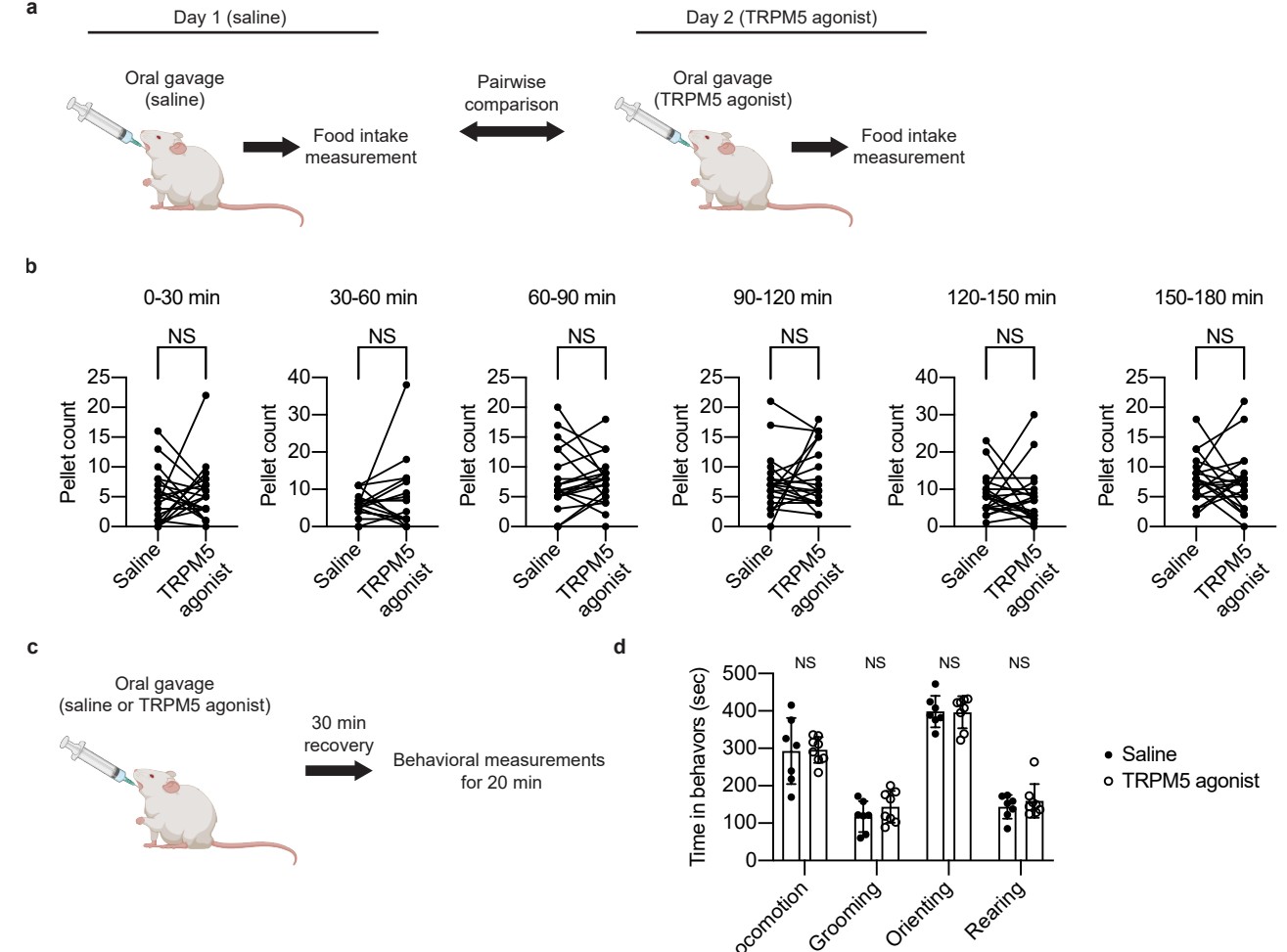

**Extended Data Fig. 8 | Acute stimulation of tuft cells does not affect food intake. a**, Schematic of food intake assay. Mice received oral gavage of 200 μL saline (day 1) or 100 μM TRPM5 agonist (day 2), followed by food intake measurement. **b**, Food intake (pellet counts) in 30-min intervals after oral gavage of saline or TRPM5 agonist. Two-tailed paired t-test; *P* = 0.7001 (0–30 min), 0.3951 (30–60 min), 0.9627 (60–90 min), 0.4999 (90–120 min), 0.7030 (120–150 min), and 0.6658 (150–180 min). *n* = 20 mice. NS: not significant. **c**, Schematic of behavioural measurements with acute TRPM5

agonist exposure. Mice received oral gavage of 200 μL saline or 100 μM TRPM5 agonist. Behavioural measurements were performed for 20 min after a 30-min recovery. **d**, Time spent in spontaneous behaviours after administration of the TRPM5 agonist. Mean ± s.d., mixed-effects analysis with Tukey's test; *P* = 0.8993, 0.2451, 0.9255 and 0.4750. *n* = 7 (saline) and 8 (IL-25) mice. NS: not significant. Illustrations in **a**,**c** created in BioRender; Touhara, K. https://BioRender.com/so6yfih (2026).

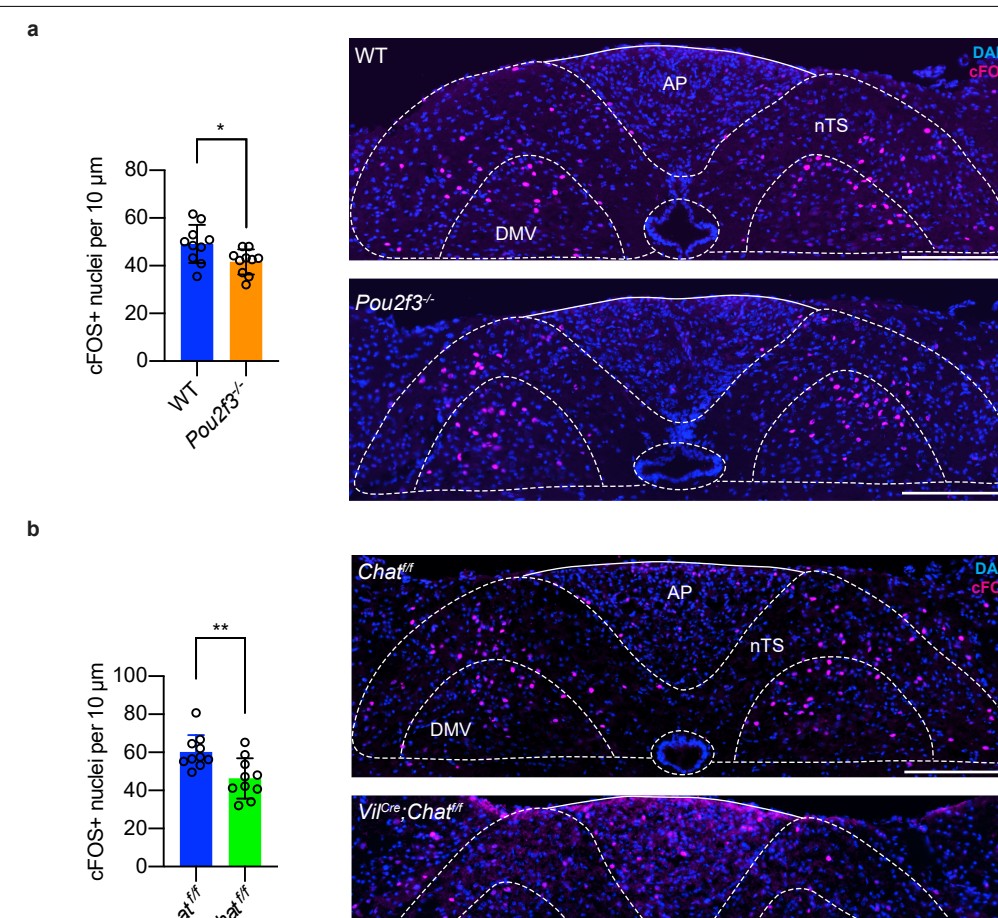

**Extended Data Fig. 9 | Helminth infection increases cFOS signals in the nTS.**
**a**, Representative brainstem sections at -Bregma −7.56 mm showing cFOS+ neurons (magenta) in the nTS after helminth-induced type 2 immune activation, which are reduced in *Pou2f3*^−/− (Tuft-cell−deficient) mice. *P* = 0.0244. **b**, Similar to **a**, showing the comparison between *Chat*^flox/flox (Ctrl) and *Vil*^Cre;*Chat*^flox/flox (Chat cKO) mice. *P* = 0.0089. >8 slides analysed per mouse from Bregma −7.32 to −7.76 mm. Comparable numbers of males and females were used in each group, with no sex difference detected. *n* = 10 for each group. AP, area postrema; DMV, dorsal motor nucleus. Scale bars, 200 µm. Mean ± s.d., two-tailed Mann–Whitney test. *P < 0.05. **P < 0.01.

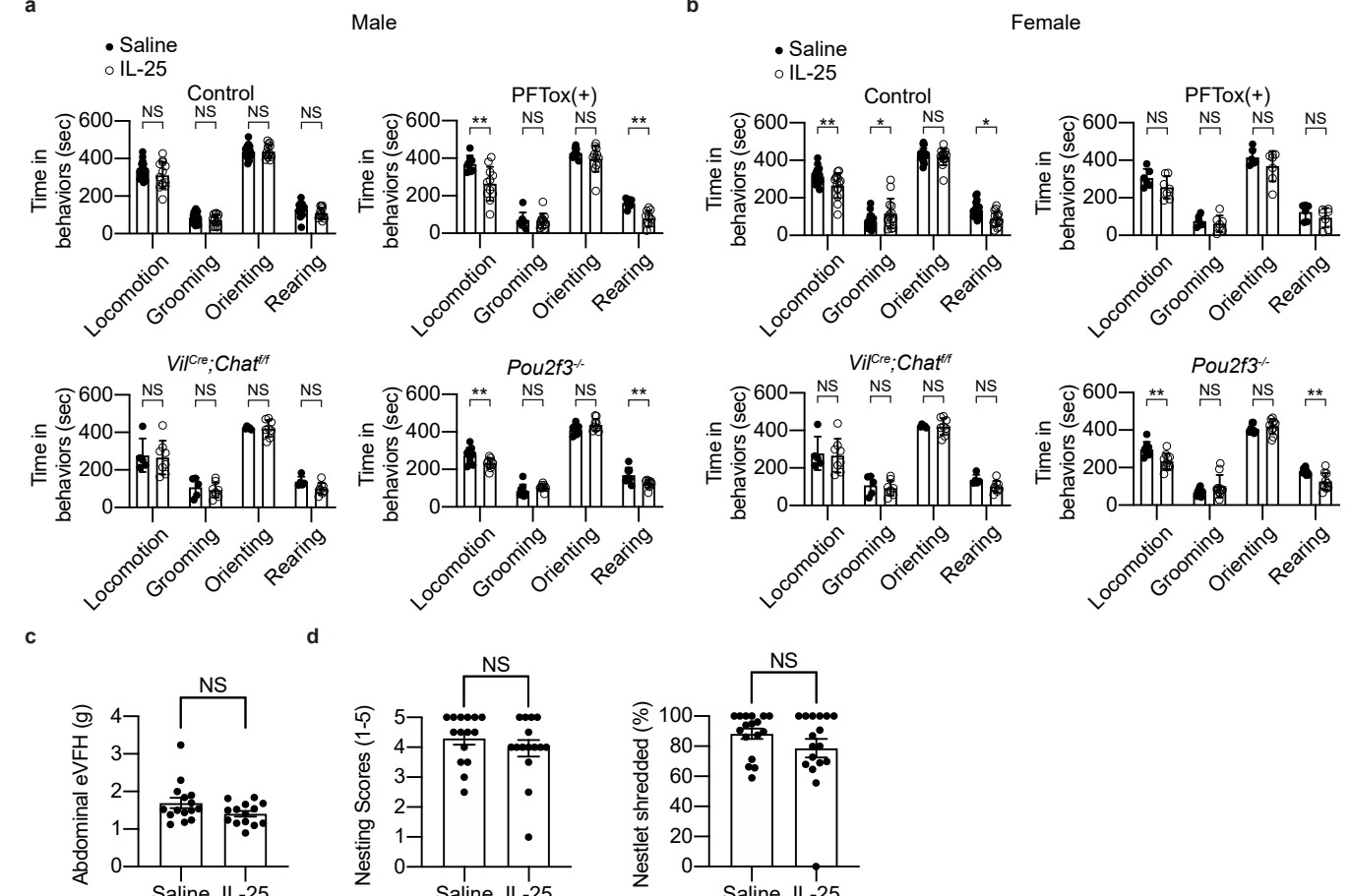

**Extended Data Fig. 10 | Changes in spontaneous behaviours during type 2 inflammation. a**,**b**,Time spent in spontaneous behaviours after saline or IL-25 treatment in male (**a**) and female (**b**) mice. Male: Mean ± s.d., mixed-effects analysis with Tukey's test; *P* = 0.0739, 0.2893, 0.9081, and 0.1554 (*n* = 19 saline-injected and 13 IL-25-injected mice: control). *P* = 0.001, 0.9056, 0.3035 and 0.0077 (*n* = 8 saline-injected and 11 IL-25-injected mice: PFTox(+)). *P* = 0.87, 0.4808, 0.9702 and 0.1686 (*n* = 5 saline-injected and 8 IL-25-injected mice: *Vil^Cre^;Chat^flox/flox^*). *P* = 0.0064, 0.1878, 0.0789 and 0.0028 (*n* = 9 mice: *Pou2f3^−/−^*). Female: Mean ± s.d., mixed-effects analysis with Tukey's test; *P* = 0.0069, 0.0172, 0.4878 and 0.009 (*n* = 18 saline-injected and 14 IL-25-injected mice: control). *P* = 0.09, 0.6024, 0.1114 and 0.2838 (*n* = 6 saline-injected and

7 IL-25-injected mice: PFTox(+)). *P* = 0.066, 0.5552, 0.6864 and 0.1881 (*n* = 8 saline-injected and 9 IL-25-injected mice: *Vil^cre^;Chat^flox/flox^*). *P* = 0.0002, 0.0822, 0.4672 and 0.0011 (*n* = 8 saline-injected and 10 IL-25-injected mice: *Pou2f3^−/−^*). **c**, Electronic von Frey hair (eVFH) thresholds in C57/BL6 mice to abdominal probing after saline or IL-25 treatment; Mean ± s.d., Mann–Whitney test; *P* = 0.1421 (*n* = 15: saline, *n* = 14: IL-25). **d**, (left) Nesting scores in C57/BL6 mice following saline or IL-25 treatment; Mean ± s.d., Mann–Whitney test; *P* = 0.3468 (*n* = 15: saline, *n* = 15: IL-25). (right) Percentage of shredded nestlet after saline or IL-25 treatment in C57/BL6. Mean ± s.d., Mann–Whitney test; *P* = 0.3776 (*n* = 15: saline, *n* = 15: IL-25). *P < 0.05, **P < 0.01. NS: not significant.

# Reporting Summary

## Statistics

For all statistical analyses, confirm that the following items are present in the figure legend, table legend, main text, or Methods section.

| n/a | Confirmed | |
|---|---|---|
| ☐ | ☒ | The exact sample size (*n*) for each experimental group/condition, given as a discrete number and unit of measurement |
| ☐ | ☒ | A statement on whether measurements were taken from distinct samples or whether the same sample was measured repeatedly |
| ☐ | ☒ | The statistical test(s) used AND whether they are one- or two-sided<br>*Only common tests should be described solely by name; describe more complex techniques in the Methods section.* |
| ☐ | ☒ | A description of all covariates tested |
| ☐ | ☒ | A description of any assumptions or corrections, such as tests of normality and adjustment for multiple comparisons |
| ☐ | ☒ | A full description of the statistical parameters including central tendency (e.g. means) or other basic estimates (e.g. regression coefficient) AND variation (e.g. standard deviation) or associated estimates of uncertainty (e.g. confidence intervals) |
| ☐ | ☒ | For null hypothesis testing, the test statistic (e.g. *F*, *t*, *r*) with confidence intervals, effect sizes, degrees of freedom and *P* value noted<br>*Give P values as exact values whenever suitable.* |
| ☒ | ☐ | For Bayesian analysis, information on the choice of priors and Markov chain Monte Carlo settings |
| ☒ | ☐ | For hierarchical and complex designs, identification of the appropriate level for tests and full reporting of outcomes |
| ☒ | ☐ | Estimates of effect sizes (e.g. Cohen's *d*, Pearson's *r*), indicating how they were calculated |

*Our web collection on statistics for biologists contains articles on many of the points above.*

## Software and code

Policy information about availability of computer code

| Data collection | Immunofluorescence images of tissue sections and cells were captured on a Nikon CSU-W1 spinning disk confocal microscope run by the Micro-Manager software (v2.0). Serotonin and acetylcholine sensor images were acquired through a Leica SP8 confocal microscope run by the LAS X software (v3.5.5.19976) or an upright microscope equipped with a Grasshopper 3 (FLIR) camera run by the Micro-Manager software (v2.0). Patch-clamp recordings were made by using a Digidata 1550B digitizer (Molecular Devices) connected to the pClamp software (v10.7). Nerve fiber recordings were made by using a 1401 interface (CED, Cambridge, UK) run by Spike2 software (V.5.18.). Optogenetic activation of nerve fibers was delivered using a High Power Fiber-Coupled LED Light Source (model BLS-FCS-0470-10) and Multimode Fiber Patchcords (Numerical aperture: 0.39 NA, Core size: 400 μm. Catalog # FPC-0400-39-025MA-BP, Mightex, Pleasanton, CA 94566, US). Single cell RT-PCR was performed on an Applied Biosystems® 7500 Real-Time PCR System. Calcium imaging of dissociated nodose neurons were made by using a Olympus IX71 microscope equipped with a CCD camera (Retiga ELECTRO) in conjunction with a Sutter Lambda 10-3 wavelength switcher and the Chroma filter set no. 49011 (ET480/40x (Ex), T510lpxrxt (BS), ET535/50m (Em)). Behaviour was measured using a Behavioural spectrometer (Behavior Sequencer, Behavioral Instruments, NJ and BiObserve, DE). Abdominal electronic Von Frey Hair test was performed in the clear plexiglas observation chamber (BSBIOPVF, Panlab, Spain). |
|---|---|

| Data analysis | We used the Fiji software (NIH, v2.14.) to generate maximal intensity projections and generate GCaMP and serotonin/acetylcholine sensor deltaF/F images and traces. The Clampfit software (v11.2.2.17, Molecular Devices) was used to analyze patch-clamp recording data. Immunofluorescence images were analyzed with Leica LAS Lite 4.0 (Leica Microsystems) or Fiji software v2.14 (NIH). Statistical analyses were done by using the Prism software (GraphPad, v8.4.3). Nerve fiber recordings were analyzed using Spike2 software (V.5.18). Single cell RT-PCR curves were analysed using 7500 Software v.2.06 from Life Technologies. Calcium imaging of dissociated neurons were analyzed using Metafluor software (Molecular Devices, V.7.8.0.0). Behaviour measured using a Behavioural spectrometer was analyzed by a computerized video tracking software (Viewer3, BiObserve, DE). Von Frey Hair data was analyzed with BIOCIS Force Ramp Software. Published single-cell RNA sequencing datasets were analyzed with Seurat 5.0.1. on R 4.4.1. |
|---|---|

For manuscripts utilizing custom algorithms or software that are central to the research but not yet described in published literature, software must be made available to editors and reviewers. We strongly encourage code deposition in a community repository (e.g. GitHub). See the Nature Portfolio guidelines for submitting code & software for further information.

# Data

Policy information about availability of data

All manuscripts must include a data availability statement. This statement should provide the following information, where applicable:
- Accession codes, unique identifiers, or web links for publicly available datasets
- A description of any restrictions on data availability
- For clinical datasets or third party data, please ensure that the statement adheres to our policy

All data generated or analyzed during this study are included in the manuscript.

# Research involving human participants, their data, or biological material

Policy information about studies with human participants or human data. See also policy information about sex, gender (identity/presentation), and sexual orientation and race, ethnicity and racism.

| Reporting on sex and gender | N/A |
|---|---|
| Reporting on race, ethnicity, or other socially relevant groupings | N/A |
| Population characteristics | N/A |
| Recruitment | N/A |
| Ethics oversight | N/A |

Note that full information on the approval of the study protocol must also be provided in the manuscript.

# Field-specific reporting

Please select the one below that is the best fit for your research. If you are not sure, read the appropriate sections before making your selection.

☒ Life sciences    ☐ Behavioural & social sciences    ☐ Ecological, evolutionary & environmental sciences

For a reference copy of the document with all sections, see nature.com/documents/nr-reporting-summary-flat.pdf

# Life sciences study design

All studies must disclose on these points even when the disclosure is negative.

| Sample size | For statistical comparisons, sample size was selected based on power calculations performed with reference to previous or present experiments carried out in our laboratory and in the field. For patch-clamp recording experiments, we collected data from 6 cells. For Ca2+ imaging experiments, we collected data from 6-15 cells. For biosensor experiments, we collected data from 6-21 organoids. For ex vivo serotonin sensor imaging comparing crypts vs. villi, we collected data from 9-17 samples in 2 mice. For other ex vivo serotonin sensor imaging, we collected images from 136-710 crypts from 3-5 animals. For cFOS staining, we collected data from 4-11 brainstem slices from 4-11 animals. For food intake measurements with IL-25 injected mice, we collected data from 10-15 mice (a mixture of males and females). For food intake measurements from helminth-infected mice, we collected data from 10 mice (a mixture of males and females). For nerve fiber recordings, we collected data from 990 nerve fibers recorded from 41 mice (a mixture of males and females). For RT-PCR, we collected data from 43 cells isolated from 3 mice. For calcium imaging we collected data from 68 cells from 6 male mice. We performed behavioral measurements for two studies. 1) Oral gavage of vehicle/Agonist 39 from 15 C57/BL6 mice. 2) I.P administration of vehicle/IL-25 from 64 WT mice, 32PFTox+ mice, 36 Pou2f3-/- mice and 30 Vilcre;Chatflox/flox mice (a mixture of males and females). Von Frey Hair measurements were performed with 29 WT mice. Nesting behavior was measured from 30 WT mice. |
|---|---|
| Data exclusions | For Ca2+ imaging and acetylcholine/serotonin sensor experiments, cells and tissues that showed substantial movements during imaging or |

| Data exclusions | abnormal high K+ response were not analyzed. For patch-clamp recording, we selected recordings that were made at an access resistance <15 MΩ and showed no drifting or excessive noise, which are common criteria for whole-cell patch-clamp recordings. For single cell RT-PCR, 5 cells were excluded, 4 cells that were not positive for Tubb3 and 1 cell that was positive for GFAP expression. For calcium imaging, only cells that displaced robust responses to high KCl were included in the analysis. |
|---|---|
| Replication | For patch-clamp recording, ex vivo serotonin sensor imaging, serotonin and acetylcholine biosensor experiments, nerve fiber recordings and Ca2+ imaging experiments, we did not repeat the same stimulation on the same cell or tissue slice. Instead, we repeated the experiments on multiple cells or tissues from multiple animals and pooled the data for statistical analysis. All experiments involving multiple cohorts were routinely assessed on different days. Replication was successful on biosensor imaging, calcium imaging, patch-clamp recordings, afferent recordings, in situ hybridization, and immunohistological experiments. Behavioral tests were conducted on multiple groups of littermates or age-matched mice at different times depending on the availability of animals, and data from all groups were analyzed collectively. |
| Randomization | Genetically modified mice or control animals (littermates or age-matched mice) were randomly selected for histological, Ca2+ imaging, serotonin sensor imaging, and patch-clamp recording experiments. |
| Blinding | Experimenter was blinded for all behavioral tests. Where possible, experimenter was blinded from the genotype or drug treatment information when performing quantifications, including event counting, and intensity measurements. |

# Reporting for specific materials, systems and methods

We require information from authors about some types of materials, experimental systems and methods used in many studies. Here, indicate whether each material, system or method listed is relevant to your study. If you are not sure if a list item applies to your research, read the appropriate section before selecting a response.

## Materials & experimental systems

| n/a | Involved in the study |
|---|---|
| ☐ | ☒ Antibodies |
| ☐ | ☒ Eukaryotic cell lines |
| ☒ | ☐ Palaeontology and archaeology |
| ☐ | ☒ Animals and other organisms |
| ☒ | ☐ Clinical data |
| ☒ | ☐ Dual use research of concern |
| ☒ | ☐ Plants |

## Methods

| n/a | Involved in the study |
|---|---|
| ☒ | ☐ ChIP-seq |
| ☒ | ☐ Flow cytometry |
| ☒ | ☐ MRI-based neuroimaging |

## Antibodies

| Antibodies used | Target and Conjugate, Host, Dilution, Manufacturer, Catalog #, RRID

Primary antibodies:
GFP, Chicken, 1:500, Abcam, ab13970, AB_300798
DCAMKL1, Rabbit, 1:250, Abcam, ab37994, AB_873538
cFOS, Rat, 1:300, Synaptic System, 226008, AB_2891278
CCK-8, Guinea pig, 1:500, Synaptic System,, 438004, AB_2814938

Secondary antibodies:
Rabbit IgG-Alexa Fluor 647, Goat, 1:500, Thermo Fisher Scientific, A-21244, AB_2535812
Rat IgG-Alexa Fluor 647, Goat, 1:500, Thermo Fisher Scientific, A-21247, AB_141778
Rabbit IgG-Alexa Fluor 568, Goat, 1:500, Thermo Fisher Scientific, A-11036, AB_10563566
Rabbit IgG-Alexa Fluor 488, Goat, 1:500, Thermo Fisher Scientific, A-11034, AB_2576217
Chicken IgY-Alexa Fluor 488, Goat, 1:500, Thermo Fisher Scientific, A-11039, AB_2534096
Chicken IgY-Alexa Fluor 488, Donkey, 1:500, Thermo Fisher Scientific, A78948, AB_2921070
Rat IgY-Alexa Fluor 568, Goat, 1:500, Thermo Fisher Scientific, A-11077, AB_141874
Rat IgY-Alexa Fluor 568, Donkey, 1:500, Thermo Fisher Scientific, A78946, AB_2910653
Guinea pig IgG-Alexa Fluor 488, Donkey, 1:500, Jackson ImmunoResearch, AB_2340472 |
|---|---|
| Validation | GFP, Chicken, 1:500, Abcam, ab13970, AB_300798
Manufacturer's validation information: This antibody is suitable for WB, ICC/IF.
Selected citations: PMID: 34463618, PMID: 34292151

DCAMKL1, Rabbit, 1:250, Abcam, ab37994, AB_873538
Manufacturer's validation information: This antibody is suitable for ICC, WB, and IP.
Selected citations: PMID: 33649045, PMID: 35314700

cFOS, Rat, 1:300, Synaptic System, 226008, AB_2891278
Manufacturer's validation information: This antibody is suitable for WB, ICC, IHC, IHC-P, iDISCO, and Clarity. |

Selected citations: PMID: 38368612, PMID: 38245542

CCK-8, Guinea pig, 1:500, Synaptic System,, 438004, AB_2814938
Manufacturer's validation information: This antibody is suitable for ICC, IHC, and IHC-P.
Selected citations: PMID: 39738072, PMID: 35550065

## Eukaryotic cell lines

Policy information about cell lines and Sex and Gender in Research

| Cell line source(s) | HEK293FT (Thermo Fisher Scientific, R70007)<br>This line is derived from the 293F Cell Line (originally obtained from Robert Horlick at Pharmacopeia) and stably expresses the SV40 large T antigen from the pCMVSPORT6TAg.neo plasmid.<br><br>R-spondin 1 expressing HEK293T (Sigma, SCC111)<br>This line is derived from the 293T cell line and stably expresses RSPO1, a protein used to establish 3D intestinal organoids. |
| --- | --- |
| Authentication | No authentication information could be found from the vendor's website |
| Mycoplasma contamination | No information about Mycoplasma contamination test could be found from the vendor's website |
| Commonly misidentified lines<br>(See ICLAC register) | N/A |

## Animals and other research organisms

Policy information about studies involving animals; ARRIVE guidelines recommended for reporting animal research, and Sex and Gender in Research

| Laboratory animals | We used mice of both sexes between the age of 8-16 weeks. Mice were raised under regular diurnal (12:12) light-dark cycles at a temperature of 68-79 degrees F and a humidity of 30-70% with ad libitum access to food and water. Strains/genotypes used include:<br><br>Villin-Cre mice (MGI:2448639) from Jackson Laboratory<br>Tac1-IRES-Cre mice (MGI:5484668) from Jackson Laboratory<br>ePet1-Flp line (MGI:3795206) is a gift from Dr. Susan Dymecki.<br>RC::PFTox line (MGI:4412286) is a gift from Dr. Susan Dymecki.<br>Polr2aGCaMP5G-tdTomato mice (MGI: 5560331) from Jackson Laboratory<br>TRPM5-EGFP line is a gift from Dr. Robert Margolskee.<br>Ai32(RCL-ChR2(H134R)/EYFP) (MGI:5013789) from Jackson Laboratory.<br>Nav1.8-Cre was gifted from Dr. Wendy Imlach, Monash University, Australia. Jackson Laboratory, Strain no. 036564.<br>gGRAB5HT3.0-P2A-jRGECO1a mice were generated in Peking University (PMID: 39939779).<br>C57BL/6 mice bred at SAHMRI, aquired from Jackson Laboratory<br>Pou2f3-/- (Jackson Laboratory, Strain no. 037040).<br>Chatflox mice (gift from Dr. Jonah Chan). |
| --- | --- |
| Wild animals | The study did not involve wild animals. |
| Reporting on sex | We used mice of both sexes and pooled the data for analyses. |
| Field-collected samples | The study did not involve samples collected from the field. |
| Ethics oversight | All animal experiments done in UCSF were conducted in accordance with protocol AN192533 approved by the Institutional Animal Care and Use Committee, University of California – San Francisco. All animal experiments done in South Australian Health and Medical Research Institute (SAHMRI) were approved and performed in accordance with the guideline of the Animal Ethics Committees of SAHMRI. |

Note that full information on the approval of the study protocol must also be provided in the manuscript.

## Plants

Seed stocks

*Report on the source of all seed stocks or other plant material used. If applicable, state the seed stock centre and catalogue number. If plant specimens were collected from the field, describe the collection location, date and sampling procedures.*

Novel plant genotypes

*Describe the methods by which all novel plant genotypes were produced. This includes those generated by transgenic approaches, gene editing, chemical/radiation-based mutagenesis and hybridization. For transgenic lines, describe the transformation method, the number of independent lines analyzed and the generation upon which experiments were performed. For gene-edited lines, describe the editor used, the endogenous sequence targeted for editing, the targeting guide RNA sequence (if applicable) and how the editor was applied.*

Authentication

*Describe any authentication procedures for each seed stock used or novel genotype generated. Describe any experiments used to assess the effect of a mutation and, where applicable, how potential secondary effects (e.g. second site T-DNA insertions, mosiacism, off-target gene editing) were examined.*

