## [Peer Review file · Nature]

Parasites Trigger Epithelial Cell Crosstalk to Drive Gut-Brain Signaling

Corresponding Author: Professor David Julius

Version 0:

Reviewer comments:

Referee #1

(Remarks to the Author)

Whether Tuft cells communicate with gut-innervating sensory neurons has remained unclear. It has often been the subject of speculation, particularly given the expression of many neuronal-associated genes in Tuft cells, but it has never been proven, unlike for enteroendocrine cells. Touhara et al. now propose a very clever solution to this problem, suggesting that Tuft cells communicate with enterochromaffin cells via acetylcholine and thereby indirectly signal to vagal afferents. This phenomenon involves the intriguing possibility of non-canonical release of the neurotransmitter through gasdermin C pores. The authors suggest that this cross-talk between Tuft cells, enterochromaffin cells, and vagal neurons mediates nocifensive responses to parasitic infections.

If sufficiently proven, this discovery would represent a very interesting new host defense mechanism. However, in its current form, the manuscript describes this mechanism very preliminarily, and almost all elements of the proposed pathway remain insufficiently proven. Below is a list of those elements which require further proof to begin to support the claims made in the manuscript, organized by the three steps of the proposed pathway.

Step 1: Tuft cell activation and acetylcholine release via gasdermin C pores

1. In IL-4 treated organoids across the various different experimental conditions used in this manuscript, what are the absolute concentrations of acetylcholine in the supernatant and how do they compare to what the authors use as their positive control in these experiments? This is important to rule out that the authors' conclusions are confounded by a threshold phenomenon in the response of the biosensor, particularly since different acetylcholine receptors are used for the different biosensor systems.
2. In the organoid IL-4 "washout" experiment shown in Fig. 1F, have the authors verified that IL-4 is indeed removed? Alternatively, can the sustained release of acetylcholine be explained by residual stimulation of the organoids after each wash?
3. Why was IL-4 used for Tuft cell activation in vitro and IL-25 used in vivo? Does IL-4 administration in vivo not lead to serotonin release by enterochromaffin cells? Is the effect of IL-25 on serotonin release dependent on IL-4 in vivo?
4. As the authors demonstrate in Fig. S4B, gasdermin C is not expressed by Tuft cells, which makes the model of acetylcholine release through gasdermin pores unlikely. Are the authors suggesting that gasdermin C pores are transferred to Tuft cells from neighboring cells to facilitate neurotransmitter release? The authors speculate that the cells "somehow collaborate" (line 128), but this is not a very satisfying explanation. The authors also suggest that acetylcholine might be transferred via gap junctions, but this remains unproven. Rather than using organoids from *Vil-Cre Gsdmc-flox* mice, the authors should use *Pou2f3-Cre* or other Tuft cell-specific lines to address a possible Tuft cell-intrinsic or -extrinsic role for gasdermin C.
5. In order to form pores, gasdermin proteins need to be cleaved. Does cleavage of gasdermin C occur in response to IL-4 treatment? What is the evidence that membrane pores are formed concomitant to acetylcholine release?

Step 2: Signaling of Tuft cell-derived acetylcholine to crypt enterochromaffin cells

6. Atropine antagonizes CHRM3 as well as other muscarinic acetylcholine receptors. The experiment with atropine does not rule out the involvement of other muscarinic type 3 receptors. More direct proof would be provided by genetic deletion of *Chrm3*.

7. It is hard to judge the quality of the bulk RNA-seq data of crypt and villus-specific enterochromaffin cells based on the data provided. It would be valuable to see a PCA that includes all genes as well as the expression levels of positive control genes.

Step 3: Signaling of Tuft cell-induced crypt enterochromaffin cell-derived serotonin to vagal neurons

8. In line 181, the authors state that "ACh released from tuft cells activates muscarinic receptors on crypt EC cells during type 2 inflammation", but no experiments are shown to support this statement. Is EC activation and serotonin release lost in *Pou2f3* KO mice? Ideally, this statement would require the usage of Tuft cell-specific *Chat*-deficient mice, which have been described in the literature.

9. Is gasdermin C required for acetylcholine release from Tuft cells, activation of enterochromaffin cells, serotonin release, vagal activation, and inhibition of food intake *in vivo*? The conclusions about the involvement of gasdermin C rests on an *in vitro* experiment with *Gsdmc*-deficient organoids, which is insufficient to make conclusions about its role in nocifensive behavior.

10. The role of the vagus nerve is implied by using NTS cFos as a proxy. Functional interventions like vagotomies would provide a more direct proof for an involvement of the vagus nerve.

11. In line with the comment above, how do the authors explain that other brainstem regions, such as the AP and DMV, likewise show increase in cFos signals in PFTox(-) mice but not in PFTox(+) mice? Would this imply that the communication with the brainstem is not vagal-specific?

12. IL-25 administration is used to mimic a type 2 immune response induced by helminth infection, but it is unclear whether any of the proposed pathway would be engaged by a physiological type 2 response. Does acetylcholine release occur and require gasdermin C in response to worm infection? Does this mechanism trigger serotonin release from enterochromaffin cells, and in this mechanism required for inhibition of food intake in response to worm infection? All of these conclusions are implied by the authors' model, but none of them are directly addressed experimentally.

13. Finally, throughout the manuscript, including the title, the authors suggest that the discovered mechanism mediates nocifensive behavior, but the only behavioral readout shown is a reduction in food intake in response to IL-25, which is somewhat delayed and ameliorated when tetanus toxin is expressed in enterochromaffin cells. Apart from the potential issues with this experimental design discussed above (IL-25 instead of infection, no proof for an involvement of Tuft cells, no proof for an involvement of gasdermin C etc.), this experiment does not satisfy the definition of nocifensive behavior, which according to the original term implies a protective response. Is anorexia a protective response in the context of type 2 immunity in the gut? Are any of the other behavioral manifestations the authors discuss in line 39 ("discomfort, diarrhea, nausea") affected by the proposed pathway?

Overall conclusions:

This manuscript uses a number of ingenious biosensor systems and introduces a very compelling hypothesis for a new host defense pathway in response to Tuft cell activation that involves a new and unexpected role for acetylcholine secretion via gasdermin pores and an indirect communication mechanism between Tuft cells and vagal afferents. However, in the absence of more definitive proof for each step of the proposed pathway, the overall conclusions are not supported by the experimental evidence included in the manuscript.

Referee #2

(Remarks to the Author)

In this study by Touhara et al, the authors postulate that tuft cells generate and release ACh in the setting of type 2 inflammation to activate neighboring enterochromaffin (EC) cells in the intestine, which triggers serotonin release and activation of vagal neurons that signal to brainstem regions to trigger food aversive behaviors. The authors also postulate that ACh is continuously released by tuft cells in the intestine through a calcium-independent, gasdermin C dependent non-canonical mechanism.

This is an interesting hypothesis of an intestinal epithelial loop that is triggered specifically in the setting of type 2 inflammation that leads to food aversive behaviors. This study also places this tuft-cell EC cell crosstalk at the initiating point of an intestinal-vagal-CNS axis.

The first part of the study uses several *in vitro* biosensors to demonstrate that acetylcholine (ACh) is generated by tuft cells and activates EC cells in an organoid system. The authors establish a system to measure ACh generation from intestinal

organoids to detect tuft cell Ach release. To do this, they first use HEK293T transfected with hM1R and GCAMP8 reporter to measure "bulk" Ach levels in organoids. They then use single cell biosensor measurements of Ach generation using a different system with GRABach (GPCR-activation-based Ach) sensor. To mark tuft cells in organoids, they employed TRPM5-GFP or Pou2f3Cre-tdTomato coupled with GCAMP, the latter allowing monitoring of Ca²⁺ flux in both tuft cells and biosensor cells. Using this system the authors find that the Ach induces activation of the biosensor cells independent of Ca²⁺ flux in the tuft cell, suggesting a new mechanism of Ach release. They postulate this new mechanism depends on epithelial Gasdermin C. To link tuft cell Ach to EC activation, the authors then use VilCre GRAB5HT3 mouse and demonstrate increased serotonin activity in intestinal explants after stimulation with Ach. RNAseq profiling of EC cells revealed expression of the muscarinic Chrm3 in crypt ECs, and Ach-induced EC activation was blocked by atropine.

The authors then go on to use a pharmacologic model with systemic IL-25 injections in mice to induce tuft cell hyperplasia. They employ VilCre GRAB5HT3 mice to detect serotonin activity in epithelial cells and demonstrate induced serotonin activity in EC cells in intestinal explants. This activation is reduced by atropine pre-treatment. They then show that IL25 injections are associated with cFOS induction in the nucleus of the solitary tractus (NTS) in the brainstem, which is reduced in PFTox mice that lack EC neurotransmitter release. They also found that IL25 decreased food intake and this was reduced in PFTox mice specifically in females. Overall, the authors postulate that the tuft cell-EC cell crosstalk leads to vagal sensory activation through serotonin and behavioral changes leading to food avoidance.

MAJOR CONCENS:

While organoid experiments are used, the tuft cell Ach to EC cell link is less convincingly shown in vivo. It is clear that the EC cells are involved in the activation of vagal neurons and this is nicely shown in the companion study. But in vivo data conclusively proving the contribution of tuft cells, tuft cell-derived Ach, and the link from tuft cell Ach to EC cells to vagal neurons is largely missing. Furthermore, the only in vivo physiological relevance of this pathway is with a pharmacologic model of IL-25 injections.

1. Physiologically relevant model of type 2 immunity: It would be nice to show that food intake and/or other aversive behaviors and NTS activation are affected by ECs in response to a more physiological system rather than IL25 injections. In particular, use of helminth infections (e.g. *Nippostrongylus*) as this already been shown to require tuft cells and induce tuft cell hyperplasia (PMID: 26675736, 26762460).
2. Tuft cell dependence for brainstem neuronal activation and behavioral changes. A recent study from Florsheim et al in Nature (PMID: 37437602) showed that tuft cells are dispensable for food aversive behaviors in the setting of type 2 inflammation. Thus, showing tuft cell dependence of aversive behaviors in the setting of a type 2 immune response here would be important - after IL-25 injections and/or helminth infections using Pou2f3^{-/-} mice (these mice were already used here for organoid studies).
3. Tuft cell-Ach dependence in vivo – it would be helpful to see the cFOS staining of the brainstem and behavioral studies in mice where tuft cells cannot produce Ach after IL-25 injections to secure the tuft cell-Ach part of the loop. The authors already have Pou2f3-cre mice, which they can use to breed with Chat^{fl/fl} mice (commercially available) to generate a tuft cell specific deletion and confirm that deficiency of Ach generation in tuft cells leads to altered behavior. Tuft cells generate a number of additional mediators that can potentially activate both neighboring epithelial cells and sensory neurons directly, so a role for Ach is not obvious without a conditional deletion study.
4. Consideration of immune cell (e.g. ILC2) derived Ach as activator of EC cells - although IL-25 is dominantly derived from tuft cells, this cytokine has pleiotropic effects on immune cells– most prominently on type 2 innate lymphoid cells (ILCs). Immune cell involvement here is not evaluated at all in vivo. This would be relevant as ILC2s in the setting of type 2 inflammation including after IL-25 injections have been shown to upregulate Chat (see PMID 33674322). Thus, the experiments in Fig. 5B/C showing effect on serotonin activity might be reflecting either tuft cell or ILC2-derived Ach as the atropine inhibition does not account for the source. The authors could consider immune cell ablation in the biosensor mice to account for that.
5. The possibility of a direct effect of tuft cell-derived Ach on vagal neurons is not discussed. Tuft cell-derived Ach was shown to directly activate vagal sensory neurons innervating the trachea (PMID: 21606356, 35503420). This possibility is not assessed nor discussed.
6. Non-canonical pathway of Ach release by intestinal tuft cells. Although there is little doubt that tuft cells can generate Ach, as has been shown by multiple groups in the past extensively for the airways, gallbladder, urethra (Perniss and Kummer 32294408, Deckmann and Kummer, Krasteva and Kummer, Keshavarz and Kummer 35245090), proposing a new Ca²⁺ independent mechanism needs stronger data to be conclusive: This concept is intriguing but the mechanism is unclear. The mechanism of GSDMC playing a role in tuft cell Ach release is not well defined, especially given that it is not expressed by the tuft cells themselves but rather in neighboring enterocytes or other epithelial cells. As the authors point out, IL25 is known to induce Gsdmc1-4 expression in enterocytes (Xi et al, PNAS 2021), which was shown to induce lytic cell death in worm induced immunity. The authors do not seem to actually reveal a mechanism of Ach release, only that neighboring GSDMC+ cells are somehow driving this. Is cell death of enterocytes somehow activating tuft cells to release Ach? Are tuft cells themselves undergoing cell death? Or are there DAMPs released like ATP that is driving Ach release?
7. Robust and reproducible in vivo data. The differences in food intake, which is the only major physiological readout, is only

occurring at day 6-7 in PFTox mice, and it is not very robust. It is mainly driven by females (n=3-4 mice per group). Can the authors repeat this with a second cohort of mice to show that it is reproducible and not due to one specific cohort?

8. In general, in their discussion of tuft cell Ach generation and release, the authors fail to acknowledge that 1) tuft cells in the intestine, gallbladder and airways were shown in many previous studies to express choline acetyltransferase (PMID: intestine – 26675736, airways -26033492, 32294408, 35245090, 22749863) and generate Ach (PMID: 32294408, 35245090 – shown by mass spectrometry; 31914675 – similar biosensor system to the one used in this paper); 2) tuft cell derived Ach has an established role in mediating Ca²⁺ flux in neighboring epithelial cells which drives mucociliary clearance and importantly activates neighboring epithelial cells for Ca²⁺ flux (PMID: 37531421). 3) Tuft cell-derived Ach in the airways also as mentioned above was shown to trigger sensory neuron activation. In that context, the findings here are still novel as they suggest a tuft cell to EC cell to sensory neurons communication but the previous literature should still be acknowledged.

MINOR CONCERNS:

1) ChAT expression in organoid tuft cells. Many experiments in Fig. 2 and 3, using organoids employing “single cell biosensor measurements” where as far as I understand the release of Ach by single tuft cells marked by either TPRM5 or Pou2f3 was assessed. The assumption here is that all Pou2f3 and/or Trpm5 expressing tuft cells in organoids also express choline acetyltransferase (Chat) and can generate Ach. ChAT is not ubiquitously expressed by intestinal tuft cells as shown with ChAT-GFP mice in a recent study from Billip and von Moltke (PMID: 38744291 and cited by the author as bioRxiv). The authors should assess the expression patterns of ChAT in organoid tuft cells to confirm that it is highly expressed across differentiated tuft cells, and only use the single cell biosensor measurements for cells that do express ChAT, i.e from ChAT-GFP mice. Or show that cell lysis does lead to release of Ach from the tuft cells.

2) The authors should also demonstrate that the succinate receptor is indeed expressed in the in vitro developed tuft cells in organoids since they show lack of Ach release with succinate activation. This is inconsistent with three recent studies (PMID: 38744292, 37531421, 38744291) and thus intriguing but receptor expression in all organoid tuft cells should be confirmed.

3) Liquid chromatography/mass spectrometry to show the specificity of the Ach measurements in parallel to validate the biosensor assessments would be nice. This methodology to detect Ach was shown recently to work for sorted intestinal tuft cells and intestinal lavage (PMID: 38744292), tracheal (PMID: 32294408) and gallbladder samples (PMID: 35245090). Mass spec analysis of organoids should not be prohibitively difficult.

4) Alternatively, would suggest showing the critical experiments with organoids derived from Pou2f3Cre-Chatfl/fl mice where Ach generation would be specifically deleted in tuft cells.

5) Can the authors also show the numbers of cFOS⁺ neurons in Area Postrema (AP) without and with IL25 injection, and if that is affected by PFTox ablation? The reason is that the AP plays a major role in nausea and aversive behaviors, but the authors do not focus on this even though that could be a key factor in their behavioral outcomes.

6) In supplemental Fig. 6B, the IL25 bottom right image shows very high background (red staining) for cFOS throughout the brainstem section compared to staining in the saline section above. Is background fluorescence normalized or how was imaging performed?

7) Can the authors explain why there is positive staining within the organoid in Fig. 1F (red area in the center where the organoid is marked)? Are these HEK gGRAB cells that randomly landed in the middle of the organoid?

8) Fig. S2A- can the authors show the single stain pictures and quantitate the overlap in several tissue sections.

Referee #3

(Remarks to the Author)

Touhara and colleagues investigate the impact of type 2 immunity on nocifensive behavior by focusing on the connection between the specialized intestinal epithelial cells, tuft cells and enterochromaffin cells (EC). Tuft cells have emerged as critical sensory cells in detecting intestinal parasites and initiating type 2 immunity through an array of effectors, including acetylcholine (ACh). While most research has focused on tuft-immune cell interactions, this manuscript found intestinal tuft cell derived ACh stimulates muscarinic receptor 1 (M1R) expressed by enterochromaffin cells (EC), thereby inducing the release of serotonin. In turn, serotonin from ECs stimulates the vagus nerve, activating neurons in the nTS brain stem as measured by cFOS staining. Genetic ablation of ECs reduced cFOS staining in the nTS region and the associated decline in food intake during type 2 immunity.

Recently tuft cell derived ACh was shown to induce Cl⁻ secretion from neighboring enterocytes (PMID: 38744291), but to my knowledge, this manuscript provides the first evidence of a connection between tuft cells and ECs. Furthermore, intestinal helminth infection reduces appetite, and this manuscript introduces a potential mechanism for this observation. The quality of experiments and overall interpretation of the data is excellent. The use of numerous biological sensors to measure ACh and HT-5 release and sophisticated mouse models are impressive and generate convincing data. Nevertheless, several issues diminish my enthusiasm but could be reasonably addressed to strengthen this otherwise interesting manuscript. Please see my specific concerns below.

Major points

- 1) The authors propose tuft cells release ACh as a prolonged leak through a GasderminC (Gsdmc) dependent mechanism. While this paper nicely establishes the importance of Gsdmc, the mechanism for this pore and ACh release is very vague. First, does ACh passively leak out, or do type 2 cytokines promote active and sustained release? Tuft cells express IL4Ra; therefore, adding IL4 to organoids could induce tuft cell differentiation and Gsdmc expression and stimulate tuft cells ACh release rather than as a slow passive leak. This possibility could be addressed by crossing the Pou2f3-Cre used in this paper with an IL4ra-flox mouse. Additionally, chemically blocking gap junction function could test the hypothesis proposed in the discussion (line 250) that tuft cells pass ACh to neighboring cells for release through Gsdmc. It is equally possible that Gsdmc allows release of other epithelial effectors that induce direct tuft cell release of ACh. Much of the author's argument against conventional vesicular release of ACh stems from the absence of vesicular transport genes, but tuft cells also lack Gsdmc, so this unconventional mechanism suffers from the same absence in cellular machinery.
- 2) This manuscript doesn't adequately address the competing hypothesis in the literature that tuft cell release ACh through calcium flux and Trpm5 membrane depolarization (PMID: 38744291). This could be easily tested using a Trpm5-KO mouse, which is both commercially available and widely found in the academic community. Additionally, Billipp et al. 2024 (PMID: 38744291) used a Trpm5-agonist to induce ACh release, which would strengthen this manuscript's conclusions if they found this agonist didn't induce ACh release in their system.
- 3) The experiments using thapsigargin and succinate to induce tuft cell calcium flux and disprove this mechanism for ACh release are slightly problematic. While thapsigargin induces a Ca²⁺ flux, does this activate TRPM5 in tuft cells? Additionally, succinate was applied to the basolateral side of organoids, when evidence suggests apical succinate is required to stimulate tuft cells. This may be why the calcium flux response was rather weak in succinate-treated organoids. Could these experiments be repeated with flipped organoids (or again use the Trpm5 agonist if organoids are basolateral out)?
- 4) Using recombinant IL25 to induce in vivo type 2 immunity is okay for some experiments but leaves substantial ambiguity about the importance of tuft cell derived ACh and intestinal specific effects of ECs. Could other gut intrinsic sources of ACh be stimulated with IL25 or downstream type 2 cytokines? Possibilities include cholinergic intrinsic neurons or intestinal ILC2s, which upregulate ChAT in response to IL25 (PMID: 33674322). Crossing the Vil-cre from this paper with the commercially available Chat-flox would be a nice way to address the source of ACh, but IL25 injections of Pou2f3^{-/-} mice would also get at the importance of tuft cells to cFOS staining and appetite suppression. Additionally, does Tac1-Cre;ePet-Fip;PFTox mice ablate ECs in other tissues besides the gut? If the vagus nerve innervates these tissues, then would systemic IL25 obscure the gut-specific effects of EC stimulation? Using a type 2 immune stimulus restricted to the intestine would strengthen these experiments.
- 5) As referenced above, intestinal helminth infection experiments would greatly improve this manuscript. These worms could provide a relevant intestinal type 2 stimulus. Additionally, parasites are referenced in the abstract, throughout the text, and in the model (Fig. 5I), but systemic IL25 injections do not adequately model these pathogens.

Minor points

- 1) The loss of ECs on reduced food intake during type 2 immunity is nice but could additional nocifensive metrics be included to strengthen this aspect of the story?
- 2) The authors suggest that tuft cells could avoid cell death by not expressing Gsdmc (lines 250-253). However, Gsdmc was shown to allow IL33 release by intestinal epithelial cells independent of cell death (PMID: 35385697). While Gsdmc2 expressed in HEK cell line triggered pyroptosis (PMID: 34290141), it's unclear whether these experiments are more relevant than the in vivo data from the first paper. The authors could incorporate both perspectives in their discussion.
- 3) The figure legends refer to gray traces in many biosensor experiments, but several figures have black traces. Could the figure legend be revised for consistency with the figures.
- 4) While the manuscript clearly shows that tuft cells don't directly release ACh through gasdermin C pores, the abstract (lines 22-23) could lead some casual readers to believe this was the case. Could the writing be refined in the abstract to avoid this confusion?

Version 1:

Reviewer comments:

Referee #1

(Remarks to the Author)

Touhara et al. have revised their manuscript in response to the comments on the previous version of their manuscript. They have also greatly modified the claims and conclusions they derive from the data. Specifically, they now present a different (biphasic) model for how acetylcholine release by tuft cells influences serotonin release by enterochromaffin cells and the

behavioral consequences of parasitic infection.

The upside of these changes is that the new model is better supported by the experimental data. The downside is that, somewhat disappointingly, the most exciting aspects of their previous model have now disappeared, including the intriguing idea of lateral acetylcholine signaling between tuft cells and enterochromaffin cells through gasdermin C pores, presumably because it is no longer in line with the experimental evidence. The critical question of how acetylcholine is released by tuft cells in the absence of a synaptic machinery thus remains open.

While it is of course important that the conclusions are now more solid, the advancements that the revised manuscript makes for our understanding of type II immune responses are now much less exciting. Enthusiasm is further dampened by the fact that the authors have in the meantime published their new tools for enterochromaffin cell activity monitoring in a separate paper, thus removing an additional major advancement from this manuscript. What is left is a thorough investigation of epithelial-neuronal crosstalk in type II immunity that presents several interesting insights but by itself is a less substantial advancement for the field, particularly since the exploration of infection-induced behavioral changes is still rudimentary and the new data is rather unconvincing.

Specific comments:

1. The authors now propose a distinction between acute TRPM5-dependent and prolonged TRPM5-independent modes of acetylcholine release. How are these different mechanistically? Now that the part about Gsdmc has been removed from the manuscript, it is unclear from the authors' explorations how acetylcholine leaves tuft cells and what causes the mechanisms underlying acute and prolonged release to be distinct.

2. The NTS analysis in Extended Data Figure 7 is used to make the argument that NTS neurons activated by type II immune responses are distinct from those activated by aversive stimuli, but it is unclear how this conclusion was reached. The data appear to be from GSE200003, which is an experiment using intranasal house dust mite treatment. How is this related to NTS neurons that receive input from gut-innervating vagal neurons? How have the authors defined NTS neurons activated in their study?

3. In the title of the manuscript and throughout the manuscript text, the authors refer to aversive behavior, but the behavioral experiments shown in Figure 6 (or anywhere else in the manuscript) do not address the question of aversion. Several behavioral assays are available in mice to directly study aversion and nausea, and the brain areas associated with aversive behavior have been identified. The authors should make use of all these behavioral tests and monitor the activity of these brain regions in the setting of helminth infection in order to support claims made about aversive behavior as well as the involvement of the tuft cell-enterochromaffin cell crosstalk therein.

4. The new behavioral data that is instead included in the manuscript (spontaneous behavioral monitoring), is quite unconvincing for multiple reasons:

a) In female mice, Pou2f3-deficient animals show the same overall behavior as control mice. The most straight-forward conclusion from this observation would be that tuft cells are not involved, suggesting that the observed behaviors are unrelated to the mechanisms explored in the rest of the manuscript.

b) The situation in males is even more confusing: Control mice are unaffected by IL-25 injection, but behavioral differences start to appear in PFTox(+) and Pou2f3-deficient mice. Does this mean that tuft cells and enterochromaffin cells buffer the effect of IL-25 on behavioral changes? This would be the opposite conclusion compared to the authors' final sentence which is that "that the tuft-EC axis sends signals to the NTS via vagal afferents, leading to aversive behaviors during the type 2 inflammatory response".

5. The functional involvement of vagal neurons in behavioral adaptations to type II inflammation has still not been demonstrated. Chemogenetic, optogenetic, or surgical interventions would be required to substantiate these claims, which was brought up in the comments on the initial version of the manuscript.

6. The sex differences highlighted in Figure 6a are interesting but remain unexplained. What contributes to this sex bias? Does the tuft cell-enterochromaffin cell crosstalk preferentially occur in females? Is it responsive to hormone signaling? Are other behavioral responses to IL-25 infection sex-dependent? Are behavioral responses to helminth infection sex-dependent? The authors mention that "sex-specific effects could not be adequately ascertained for these Nb infection experiments", but do not show the data based on which they make this statement.

Minor points and suggestions:

1. The authors now included additional quantifications of cFos in the NTS vs. AP and DMV and conclude that changes can only be observed in the NTS. This is in stark contrast to the image shown in Figure 5j, which actually shows the strongest differences in the AP and least pronounced changes in the NTS. It would be good to include all the images that the quantification data is based upon.

2. The authors have re-arranged the flow of the manuscript, which results in a new logical sequence that is at times hard to follow. Some examples are:

a) The first sentence of the first section: "EC cells are now recognized as polymodal integrators of noxious stimuli in the gut, prompting us to ask whether they might be a target for ACh activation". ACh comes out of nowhere – why does the function of EC cells as integrators of noxious stimuli prompt an exploration of ACh responses?

b) The first sentence of the second section: "Having identified crypt EC cells as a bona fide target of ACh, we wondered whether and how they are activated by cholinergic tuft cells". Why tuft cells rather than ACh-producing neurons?

Referee #2

(Remarks to the Author)

Summary of the key results

In this revised manuscript Touhara et al dissect a neuroepithelial sensing loop from tuft cells via EC cells to the vagus engaged in aversive behaviors in the initial stages of type 2 immune intestinal inflammation.

The authors demonstrate that tuft cells are indeed a source of acetylcholine generated in response to tuft cell ligands. ACh from tuft cells activates EC cells through ACh receptors and EC cells transmit signals to the vagus through serotonin to mediate signals leading to aversive behaviors likely due to visceral pain. The authors dissect each step using in vitro cultures and whole intestinal preps with ACh and serotonin sensing report mice. Interestingly, the authors find spatial segregation of tuft cells and EC cells (and specifically their subsets that can crosstalk) at homeostasis and a much more intimate connection in the setting of tuft cell hyperplasia after IL-25 stimulation or during helminth infections.

Originality: The authors have previously elegantly shown the important role of EC cells signaling to the vagus and brain through serotonin. Now they add another layer to this system where tuft cells are the frontline sensor and signal to EC cells through acetylcholine.

Data and methodology: The authors use multiple innovative and complementary approaches including ACh and Serotonin reporter/GRAB mice, M1R GCamp, to detect generation of each of the neuromediator in culture and in tissue preps, signaling from tuft cells to EC cells and then to sensory neurons, including reporter mice with tuft cell genetic deletion and epithelial cell-specific deletion of choline acetyltransferase and finally an integrated physiologic model in the setting of helminth infection.

Conclusions: The conclusion of this revised manuscript are robust backed by several techniques and models for each conclusion making this a comprehensive study of the gut tuft to EC cell crosstalk during helminth infection. The appropriate papers are referenced. The graphical summary and discussion nicely summarize the findings.

The authors have addressed all of my previous comments/suggestions for improvement.

Referee #3

(Remarks to the Author)

Touhara and colleagues have substantially strengthened their manuscript examining the connection between tuft cells, enterochromaffin cells, and type 2 immune induced aversive behavior. The authors have addressed all previous concerns with rigor and clarity.

The speculative GasderminC component has been removed and replaced with a well supported model in which tuft cells release acetylcholine through TRPM5 dependent and cytokine induced mechanisms. Using *Nippostrongylus brasiliensis* infection together with tuft cell deficient (Pou2f3-KO) and epithelial ChAT deficient mice, the authors show that tuft cell derived acetylcholine drives enterochromaffin cell activation, serotonin release, and feeding suppression, findings further supported by expanded behavioral and cFOS analyses in the nucleus tractus solitarius.

The revised manuscript is clear, technically sophisticated, and conceptually strong. It provides compelling evidence that epithelial crosstalk regulates behavioral responses during type 2 immune activation. I am satisfied that all concerns have been fully addressed and that this work represents a significant scientific contribution. I congratulate the authors on an interesting and important study.

Version 2:

Reviewer comments:

Referee #1

(Remarks to the Author)

The authors have appropriately responded to my remaining concerns. I do have to admit that it is somewhat dispiriting that the conclusions of the manuscript get diminished and more vague with every revision (from gasdermin C-mediated epithelial crosstalk in nocifensive behavior, to aversive behavior, and now to simply food intake), and that new results previously introduced are now deemed "inconclusive". I certainly do not want to burden the authors with additional revisions and hope the conclusions presented in the newest version of the manuscript are now solid despite the remaining oddities acknowledged by the authors.

Response to Reviewers

We thank the referees for their thoughtful and constructive feedback on our manuscript. In response to their criticisms, we have performed substantial additional experiments and restructured the manuscript to address key concerns, as summarized below:

- 1. Revised analysis of ACh release from tuft cells:** Since we cannot definitively establish a contribution of gasdermin C to ACh release, we have removed this section from the manuscript. Instead, we now focus on providing specific and critical biophysical and cell biological insights into the unconventional nature of ACh release from tuft cells. Through patch-clamp recordings, we demonstrate that intestinal tuft cells are electrically non-excitable and cannot fire action potentials (Fig. 2b-c). Analysis of single-cell RNA sequencing data confirms that these cells lack conventional synaptic release machinery (Extended Data Fig. 2a). Despite these limitations, we show that single isolated tuft cells can acutely release ACh in response to succinate or TRPM5 activation (Fig. 2d-f). We also demonstrate that during type 2 inflammation, tuft cells exhibit constitutive "leak-like" ACh release that is TRPM5-independent (Fig. 3). These findings reconcile our observations with recent publications showing that tuft-derived ACh contributes to fluid secretion (PMID: 38744291 and PMID: 38744292) while providing important constraints on possible release mechanisms. Thus, our work demonstrates that tuft cells employ a fundamentally unconventional pathway for acetylcholine (ACh) release, representing a significant advance in identifying and characterizing peripheral sensory signaling involving a non-neuronal / non-canonical mechanism of neurotransmitter release.
- 2. Validation of vagal activation by the tuft-EC pathway:** We performed a range of electrophysiological and histological experiments, including ex vivo jejunal afferent recordings and cFOS staining, to demonstrate that the tuft-EC pathway activates vagal afferents upon type 2 inflammation. We confirmed that ACh effects on vagal afferents are primarily mediated through the EC cell-serotonin pathway rather than direct cholinergic activation of sensory nerve fibers (Extended Data Fig. 6). Additionally, we showed that type 2 inflammation increases vagal afferent activity in both EC and tuft-dependent manners (Fig. 5f-h). These electrophysiological findings, combined with our robust cFOS validation in the brainstem (Extended Data Fig. 7 and 9), establish a complete neural circuit from gut epithelial sensing to brainstem processing of aversive signals during parasitic infection.
- 3. Genetic validation using tuft cell-deficient and intestinal tuft cell-specific ChAT-deficient mice:** Following the referees' suggestions, we now employ tuft cell-deficient (*Pou2f3*^{-/-}) and gut tuft cell-specific ACh-deficient (*Vil*^{Cre};*Chat*^{fllox/fllox}) mice across multiple experimental paradigms, including serotonin sensor measurements (Fig. 4c-d and 5b-e), food intake assays (Fig. 6b-c), cFOS staining (Extended Data Fig. 9), and behavioral tests (Fig. 6d and Extended Data Fig. 10). These genetic approaches further validate our conclusion that tuft-derived ACh is an essential component of the proposed tuft-EC-mediated nocifensive pathway, establishing this transmitter as the key mediator linking type 2 immune responses to aversive / protective behaviors.
- 4. Addition of a physiologically relevant helminth infection model:** As requested by the referees, we performed a substantial series of new experiments involving *Nippostrongylus brasiliensis* infection, including ex vivo serotonin biosensor assays (Fig. 5d-e), food intake measurements (Fig. 6b-c), and cFOS quantification in the brainstem (Extended Data Fig. 9) using tuft cell-deficient and gut tuft cell-specific ACh-deficient mice. Consistent with our IL-25 injection experiments, we found that *Nippostrongylus brasiliensis* infection activates the tuft-EC axis and initiates nocifensive / aversive behaviors in a tuft- and tuft-ACh-dependent manner. This bolsters the physiological relevance of our study and validates our findings in a natural infection context.
- 5. Comprehensive behavioral validation across genetic models:** We investigated a range of spontaneous behaviors during type 2 inflammation in mice using EC knockdown (PFTox), tuft cell-deficient (*Pou2f3*^{-/-}), and gut tuft-specific ACh-deficient (*Vil*^{Cre};*Chat*^{fllox/fllox}) mice (Fig. 6d and Extended Data Fig. 10). These experiments further support the tuft-EC axis as the central mediator of aversive responses during type 2 inflammation. As observed with food intake, female mice exhibited the greatest differences in locomotion, grooming, and rearing following IL-25 treatment, consistent with our previous observation that visceral pain is more pronounced in females compared to males (PMID: 36949192). This extended

behavioral analysis provides further validation that the tuft-EC axis drives aversive behaviors during type 2 immune responses.

Based on these new results, we propose a two-phase model of tuft cell ACh release that can account for the progression from asymptomatic initial infection to symptomatic established parasitic disease (Fig. 6e). We demonstrate that tuft cells employ two distinct mechanisms of ACh release: (1) acute TRPM5-dependent release in response to succinate or a selective TRPM5 agonist, representing the initial infection phase (Fig. 2), and (2) constitutive TRPM5-independent "leak-like" release during established type 2 inflammation (Fig. 3). While we have shown that both mechanisms activate crypt EC cells to evoke serotonin release (Fig. 4 and 5), only the type 2 inflammatory setting generates sufficient EC cell activation to trigger nocifensive behaviors (Fig. 5 and 6). This two-phase paradigm is consistent with the clinical observation that parasitic infections typically begin asymptotically, with progressive development of gastrointestinal symptoms only once parasites colonize the host and begin to mature and reproduce.

The revised manuscript incorporates substantial new experimental work that not only addresses the reviewers' concerns but also develops a more detailed mechanistic framework that enhances the impact, rigor, and scope of our findings. We are grateful to all of the referees for their insightful feedback, which has guided us toward a more robust and physiologically relevant study.

Referee #1 (Remarks to the Author):

Whether Tuft cells communicate with gut-innervating sensory neurons has remained unclear. It has often been the subject of speculation, particularly given the expression of many neuronal-associated genes in Tuft cells, but it has never been proven, unlike for enteroendocrine cells. Touhara et al. now propose a very clever solution to this problem, suggesting that Tuft cells communicate with enterochromaffin cells via acetylcholine and thereby indirectly signal to vagal afferents. This phenomenon involves the intriguing possibility of non-canonical release of the neurotransmitter through gasdermin C pores. The authors suggest that this cross-talk between Tuft cells, enterochromaffin cells, and vagal neurons mediates nocifensive responses to parasitic infections. If sufficiently proven, this discovery would represent a very interesting new host defense mechanism. However, in its current form, the manuscript describes this mechanism very preliminarily, and almost all elements of the proposed pathway remain insufficiently proven. Below is a list of those elements which require further proof to begin to support the claims made in the manuscript, organized by the three steps of the proposed pathway.

We appreciate the referee's comprehensive and constructive review. S/he has identified key gaps in our original manuscript that we have now addressed through extensive additional experimentation. Most importantly, we have: (1) provided genetic evidence using tuft cell-deficient and intestinal tuft cell-specific ChAT-deficient mice, (2) demonstrated the pathway's physiological relevance using helminth infection models, (3) characterized the biophysical properties of tuft cell ACh release, and (4) confirmed vagal activation through direct nerve fiber recordings and cFOS staining. We believe that these substantial additions transform the preliminary observations into a well-supported mechanistic framework.

Step 1: Tuft cell activation and acetylcholine release via gasdermin C pores

1. In IL-4 treated organoids across the various different experimental conditions used in this manuscript, what are the absolute concentrations of acetylcholine in the supernatant and how do they compare to what the authors use as their positive control in these experiments? This is important to rule out that the authors' conclusions are confounded by a threshold phenomenon in the response of the biosensor, particularly since different acetylcholine receptors are used for the different biosensor systems.

We appreciate this important technical question. We can estimate the concentration of ACh released from tuft cells by comparing responses to calibrated gGRAB_{ACh4h} signals. For example, in Figure 2e (shown below), average normalized gGRAB_{ACh4h} signals increased to ~0.5, which corresponds to ~40 nM ACh. This means that a single isolated tuft cell releases ACh at concentrations measuring ~40 nM at ~5 μm from the release site (where the gGRAB_{ACh4h} biosensor HEK cell was positioned). This concentration of ACh is not high enough to stimulate

muscarinic receptors, whose EC_{50} for ACh is ~ 500 nM (PMID: 8156638). However, this explains why multiple tuft cells are required to produce sufficient levels of ACh needed to activate crypt EC cells. Indeed, this is consistent with what we have shown, namely, that tuft cell hyperplasia and sustained leak-like ACh release during type 2 inflammation are required to activate crypt EC cells at a level that is sufficient to send signals to mucosal afferents (Fig. 5). Given that this quantity threshold appears critical for activating the vagal pathway, we have expanded our discussion of how tuft cell-derived ACh initiates the tuft-EC pathway, as well as the physiological implications of this dosage requirement (line 284).

2. In the organoid IL-4 “washout” experiment shown in Fig. 1F, have the authors verified that IL-4 is indeed removed? Alternatively, can the sustained release of acetylcholine be explained by residual stimulation of the organoids after each wash?

In IL-4 experiments now shown in Fig. 3, we washed and cultured the organoids with IL-4 free media for one full day, and then organoids were harvested from the Matrigel and replated in Ringer's solution without IL-4. We have now articulated this point in the main text (line 141) and method section. This extensive washout procedure ensures complete removal of IL-4 from the system prior to our analysis.

3. Why was IL-4 used for Tuft cell activation in vitro and IL-25 used in vivo? Does IL-4 administration in vivo not lead to serotonin release by enterochromaffin cells? Is the effect of IL-25 on serotonin release dependent on IL-4 in vivo?

We used IL-25 instead of IL-4 for biological reasons. IL-25 stimulates the type 2 immune cascade independent of tuft cells, whereas IL-4 stimulates intestinal epithelial stem cells to cause tuft cell hyperplasia and consequent release of endogenous IL-25. Therefore, we thought that IL-25 would minimize any confounding effects that IL-4 might have on this cascade and provide a more direct route to type 2 immune activation. We now include an explanation of why we used IL-25 (line 172).

4. As the authors demonstrate in Fig. S4B, gasdermin C is not expressed by Tuft cells, which makes the model of acetylcholine release through gasdermin pores unlikely. Are the authors suggesting that gasdermin C pores are transferred to Tuft cells from neighboring cells to facilitate neurotransmitter release? The authors speculate that the cells “somehow collaborate” (line 128), but this is not a very satisfying explanation. The authors also suggest that acetylcholine might be transferred via gap junctions, but this remains unproven. Rather than using organoids from Vil-Cre Gsdmc-flox mice, the authors should use Pou2f3-Cre or other Tuft cell-specific lines to address a possible Tuft cell-intrinsic or -extrinsic role for gasdermin C.

5. In order to form pores, gasdermin proteins need to be cleaved. Does cleavage of gasdermin C occur in response to IL-4 treatment? What is the evidence that membrane pores are formed concomitant to acetylcholine release?

As described above, we have now removed the gasdermin C component from our manuscript, as we cannot definitively establish a contribution of gasdermin C to ACh release. Instead, we provided mechanistic insights into how a single isolated tuft cell can release ACh without synaptic release machinery and electrical excitability, which we believe provide valuable new insights into non-canonical neurotransmitter release mechanisms (Fig. 2 and Extended Data Fig. 2). This essential information provides the impetus to explore this enigmatic question in a

future study, while we focus here on elaborating a paracrine signaling cascade between tuft and EC cells and the physiological and behavioral consequences.

Step 2: Signaling of Tuft cell-derived acetylcholine to crypt enterochromaffin cells

6. Atropine antagonizes CHRM3 as well as other muscarinic acetylcholine receptors. The experiment with atropine does not rule out the involvement of other muscarinic type 3 receptors. More direct proof would be provided by genetic deletion of *Chrm3*.

Indeed, although we identified specific expression of *Chrm3* in crypt EC cells (Fig. 1e-f), which was corroborated by our serotonin sensor experiments, where only crypt EC cells exhibit responses to ACh (Fig. 1j-k), we cannot exclude the possibility that other muscarinic receptors, especially *Chrm1*, contribute to this activation. We therefore revised the abstract and summary figure to refer to muscarinic receptors more generally instead of definitively specifying the *Chrm3* subtype, as we feel this point is not critical to the main conclusions of this study.

7. It is hard to judge the quality of the bulk RNA-seq data of crypt and villus-specific enterochromaffin cells based on the data provided. It would be valuable to see a PCA that includes all genes as well as the expression levels of positive control genes.

We have now published this dataset in our recent paper (PMID: 40327690, GEO ID: GSE291832; see the figure below). In this manuscript, we demonstrated that known crypt-biased genes (e.g. *Tac1* and *Trpa1*) and villus-biased (e.g. *Sct* and *Trpm2*) genes are indeed biased and consistent with previously published single-cell RNA seq datasets (PMID: 30038251 and PMID: 36810133). We therefore believe that our analysis of acetylcholine receptor genes accurately represents crypt vs. villus bias.

Step 3: Signaling of Tuft cell-induced crypt enterochromaffin cell-derived serotonin to vagal neurons

8. In line 181, the authors state that “ACh released from tuft cells activates muscarinic receptors on crypt EC cells during type 2 inflammation”, but no experiments are shown to support this statement. Is EC activation and serotonin release lost in *Pou2f3* KO mice? Ideally, this statement would require the usage of Tuft cell-specific *Chat*-deficient mice, which have been described in the literature.

This is an excellent suggestion. To address this point, we crossed serotonin sensor mice (*Vil^{Cre};gGRAB_{5-HT3.0}*) to tuft cell-deficient (*Pou2f3^{-/-}*) mice to remove tuft cells, or to *Chat^{flox/flox}* mice to specifically remove ACh from intestinal tuft cells. Using these lines, we now demonstrate that tuft cell-derived ACh is essential for EC cell activation and serotonin release following both acute tuft stimulation (succinate or TRPM5 agonist) and type 2 inflammation (Fig. 4c-d and 5b-e).

9. Is gasdermin C required for acetylcholine release from Tuft cells, activation of enterochromaffin cells, serotonin release, vagal activation, and inhibition of food intake in vivo? The conclusions about the involvement of gasdermin C rests on an in vitro experiment with *Gsdmc*-deficient organoids, which is insufficient to make conclusions about its role in nociceptive behavior.

As described above, we have removed the gasdermin C component and instead focused on characterizing the unconventional biophysical properties of tuft cell ACh release, which provides important new insights into non-neuronal neurotransmitter release mechanisms.

10. The role of the vagus nerve is implied by using NTS cFos as a proxy. Functional interventions like vagotomies would provide a more direct proof for an involvement of the vagus nerve.

We appreciate this suggestion. To confirm vagal activation by the tuft-EC axis, we performed comprehensive electrophysiological recordings from jejunal afferents (Fig. 5f-h and Extended Data Fig. 6e-j). We observed a significant increase in nerve fiber activity upon type 2 inflammation, which was blocked by alosetron, a 5-HT₃ receptor antagonist that inhibits communication between EC cells and nerve fibers, confirming the EC-serotonin pathway as the primary mediator (Fig. 5f-g). Additionally, this activation was diminished in tuft cell-specific ChAT-deficient mice, providing genetic evidence for the requirement of tuft cell-derived ACh (Fig. 5h). Together with our cFOS staining results demonstrating activation of nTS neurons in the brainstem (Fig. 5i-k and Extended Data Fig. 7 and 9), these complementary electrophysiological and histological approaches provide robust evidence for vagal pathway engagement by the tuft-EC axis, establishing a functional gut-to-brain circuit for aversive signaling during parasitic infection.

11. In line with the comment above, how do the authors explain that other brainstem regions, such as the AP and DMV, likewise show increase in cFos signals in PFTox(-) mice but not in PFTox(+) mice? Would this imply that the communication with the brainstem is not vagal-specific?

We thank the reviewer for this insightful comment. To address this question, we quantified cFOS signals across the nTS, AP, and DMV rather than just showing representative images (Fig. 5i-k and Extended Data Fig. 7a). Our analysis now shows that IL-25-induced type 2 inflammation significantly increased cFOS+ neurons specifically in the nTS, with no significant changes in either the AP or DMV. In any case, this supports our conclusion that vagal activation in response to type 2 inflammation is transmitted to a region of the brainstem associated with aversive behaviors. Moreover, because this signal is diminished in PFTox(+) mice, we conclude that the release of serotonin from EC cells is involved in this pathway.

To further characterize the nTS activation, we conducted a comprehensive population analysis of cFOS+ neurons (Extended Data Fig. 7b-c). We found minimal overlap between the activated cFOS+ neurons in our study and previously defined nTS neuronal populations (PMID: 27301688, 31839488, 37993711, 36778350), suggesting recruitment of a distinct neuronal subset in response to IL-25-induced inflammation.

12. IL-25 administration is used to mimic a type 2 immune response induced by helminth infection, but it is unclear whether any of the proposed pathway would be engaged by a physiological type 2 response. Does acetylcholine release occur and require gasdermin C in response to worm infection? Does this mechanism trigger serotonin release from enterochromaffin cells, and in this mechanism required for inhibition of food intake in response to worm infection? All of these conclusions are implied by the authors' model, but none of them are directly addressed experimentally.

We greatly appreciate this important concern. To address this, we initiated experiments using *Nippostrongylus brasiliensis* infection, including ex vivo serotonin biosensor assays, food intake measurements, and cFOS quantification in the brainstem using tuft cell-deficient and gut tuft cell-specific ACh-deficient mice. We found that helminth infection increased baseline serotonin levels in crypts, which was abolished in both tuft cell-deficient and gut tuft cell-specific ACh-deficient mice (Fig. 5d-e). Similarly, *N. brasiliensis* infection reduced food intake in wild-type mice at the peak of tuft cell hyperplasia (days 7-9), whereas tuft-deficient and gut tuft cell-specific ACh-deficient mice maintained normal feeding patterns (Fig. 6b-c). cFOS staining was consistent with the food intake data, showing increased signals in control animals, but less pronounced activation in knockout animals (Extended Data Fig. 9). These findings significantly increase the physiological relevance of our study and validate our IL-25 findings in a natural infection context.

13. Finally, throughout the manuscript, including the title, the authors suggest that the discovered mechanism mediates nocifensive behavior, but the only behavioral readout shown is a reduction in food intake in response to IL-25, which is somewhat delayed and ameliorated when tetanus toxin is expressed in enterochromaffin cells. Apart from the potential issues with this experimental design discussed above (IL-25 instead of infection, no proof for an involvement of Tuft cells, no proof for an involvement of gasdermin C etc.), this experiment does not satisfy the definition of nocifensive behavior, which according to the original term implies a protective response. Is anorexia a protective response in the context of type 2 immunity in the gut? Are any of the other behavioral manifestations the authors discuss in line 39 (“discomfort, diarrhea, nausea”) affected by the proposed pathway?

We appreciate this important point. We investigated a range of spontaneous behaviors during type 2 inflammation using EC knockdown (PFTox), tuft cell-deficient (*Pou2f3^{-/-}*), and gut tuft-specific ACh-deficient (*Vil^{Cre}; Chat^{flox/flox}*) mice. These experiments further support the role of the tuft-EC axis in mediating protective behavioral responses to parasitic infections.

Beyond food intake reduction, we found that female mice exhibited increased grooming behavior following IL-25 treatment, which is indicative of stress or discomfort (Fig. 6d and Extended Data Fig. 10). Additionally, we observed reduced locomotion and rearing behaviors, which are consistent with sickness behaviors. These behavioral changes were diminished in PFTox and epithelial ChAT-deficient mice, confirming their dependence on the tuft-EC axis. Interestingly, these behaviors, as well as effects on food intake, showed sex-bias with significant differences seen in females but not in males. This is consistent with our prior observations that females exhibit greater visceral sensitivity compared to males (PMID: 36949192). Together with the involvement of the nTS in the brainstem, we concluded that this tuft-EC pathway evokes aversive behaviors that likely reflect GI discomfort or diminished appetite. However, we acknowledge that the exact sensations induced by this pathway remain unclear. Behavioral assessment of visceral sensations, particularly from the small intestine, remains challenging in rodents. Even in humans, clinical evaluation of visceral discomfort is notoriously difficult, highlighting the inherent challenges in understanding nociceptive pathways for such diffuse and hard-to-describe sensations. Given the importance of this point, we have expanded our discussion of these behavioral findings and their protective significance in the Discussion section (line 301).

Overall conclusions:

This manuscript uses a number of ingenious biosensor systems and introduces a very compelling hypothesis for a new host defense pathway in response to Tuft cell activation that involves a new and unexpected role for acetylcholine secretion via gasdermin pores and an indirect communication mechanism between Tuft cells and vagal afferents. However, in the absence of more definitive proof for each step of the proposed pathway, the overall conclusions are not supported by the experimental evidence included in the manuscript.

We certainly appreciate these points, in response to which we now provide more mechanistic details and experiments with *N. brasiliensis* to investigate this pathway in a more physiological setting, combined with definitive genetic validation using tuft cell-deficient (*Pou2f3^{-/-}*) and gut tuft cell-specific ACh-deficient (*Vil^{Cre}; Chat^{flox/flox}*) mice. We believe that these substantial additions address the referee’s concerns and strengthen the evidence for our proposed mechanism.

Referee #2 (Remarks to the Author):

In this study by Touhara et al, the authors postulate that tuft cells generate and release Ach in the setting of type 2 inflammation to activate neighboring enterochromaffin (EC) cells in the intestine, which triggers serotonin release and activation of vagal neurons that signal to brainstem regions to trigger food aversive behaviors. The authors also postulate that Ach is continuously released by tuft cells in the intestine through a calcium-independent, gasdermin C dependent non-canonical mechanism.

This is an interesting hypothesis of an intestinal epithelial loop that is triggered specifically in the setting of type 2 inflammation that leads to food aversive behaviors. This study also places this tuft-cell EC cell crosstalk at the initiating point of an intestinal-vagal-CNS axis.

The first part of the study uses several in vitro biosensors to demonstrate that acetylcholine (ACh) is generated by tuft cells and activates EC cells in an organoid system. The authors establish a system to measure ACh generation from intestinal organoids to detect tuft cell ACh release. To do this, they first use HEK293T transfected with hM1R and GCAMP8 reporter to measure “bulk” ACh levels in organoids. They then use single cell biosensor measurements of ACh generation using a different system with GRABach (GPCR-activation-based ACh) sensor. To mark tuft cells in organoids, they employed TRPM5-GFP or Pou2f3Cre-tdTomato coupled with GCAMP, the latter allowing monitoring of Ca²⁺ flux in both tuft cells and biosensor cells. Using this system the authors find that the ACh induces activation of the biosensor cells independent of Ca²⁺ flux in the tuft cell, suggesting a new mechanism of ACh release. They postulate this new mechanism depends on epithelial Gasdermin C. To link tuft cell ACh to EC activation, the authors then use VilCre GRAB5HT3 mouse and demonstrate increased serotonin activity in intestinal explants after stimulation with ACh. RNAseq profiling of EC cells revealed expression of the muscarinic Chrm3 in crypt ECs, and ACh-induced EC activation was blocked by atropine.

The authors then go on to use a pharmacologic model with systemic IL-25 injections in mice to induce tuft cell hyperplasia. They employ VilCre GRAB5HT3 mice to detect serotonin activity in epithelial cells and demonstrate induced serotonin activity in EC cells in intestinal explants. This activation is reduced by atropine pre-treatment. They then show that IL25 injections are associated with cFOS induction in the nucleus of the solitary tractus (NTS) in the brainstem, which is reduced in PFTox mice that lack EC neurotransmitter release. They also found that IL25 decreased food intake and this was reduced in PFTox mice specifically in females. Overall, the authors postulate that the tuft cell-EC cell crosstalk leads to vagal sensory activation through serotonin and behavioral changes leading to food avoidance.

MAJOR CONCENS:

While organoid experiments are used, the tuft cell ACh to EC cell link is less convincingly shown in vivo. It is clear that the EC cells are involved in the activation of vagal neurons and this is nicely shown in the companion study. But in vivo data conclusively proving the contribution of tuft cells, tuft cell-derived ACh, and the link from tuft cell ACh to EC cells to vagal neurons is largely missing. Furthermore, the only in vivo physiological relevance of this pathway is with a pharmacologic model of IL-25 injections.

We thank the referee for recognizing the novelty and potential importance of our findings while providing constructive criticism. Indeed, the original manuscript lacked direct evidence for involvement of tuft cell-derived ACh and its physiological relevance. We now employed tuft cell-deficient and intestinal tuft cell-specific ACh-deficient mice to pinpoint the contributions of tuft-derived ACh (Fig. 4c-d, 5b-e, and 6b-c). We also performed nerve fiber recordings to reinforce the link between the tuft-EC axis and vagal afferents (Fig. 5f-h and Extended Data Fig. 6). Lastly, we employed *N. brasiliensis*, an actual helminth, to perform serotonin sensor imaging (Fig. 5d-e), food intake measurements (Fig. 6b-c), and cFOS staining (Extended Data Fig. 9), demonstrating the relevance of our findings to a *bona fide* pathological condition. We believe these experiments fulfill the missing links in the original manuscript.

1. Physiologically relevant model of type 2 immunity: It would be nice to show that food intake and/or other aversive behaviors and NTS activation are affected by ECs in response to a more physiological system rather than IL25 injections. In particular, use of helminth infections (e.g. *Nippostrongylus*) as this already been shown to require tuft cells and induce tuft cell hyperplasia (PMID: 26675736, 26762460).

We greatly appreciate this suggestion. We have now performed substantial experiments using *Nippostrongylus brasiliensis* infection, including ex vivo serotonin biosensor assays, food intake measurements, and cFOS quantification in the brainstem using tuft cell-deficient and gut tuft cell-specific ACh-deficient mice. We found that helminth infection increased baseline serotonin levels in crypts, which was abolished in both tuft cell-deficient and gut tuft-specific ACh-deficient mice (Fig. 5d-e). Similarly, *N. brasiliensis* infection reduced food intake in wild-type mice at the peak of tuft cell hyperplasia (days 7-9), whereas tuft-deficient and gut tuft cell-specific ACh-deficient mice maintained normal feeding patterns (Fig. 6b-c). FOS staining was consistent with the food intake data (Extended Data Fig. 9), showing increased signals in control animals but less pronounced activation in knockout animals. This significantly increases the physiological relevance of our study and validates

our IL-25 findings in a natural infection context.

2. Tuft cell dependence for brainstem neuronal activation and behavioral changes. A recent study from Florsheim et al in Nature (PMID: 37437602) showed that tuft cells are dispensable for food aversive behaviors in the setting of type 2 inflammation. Thus, showing tuft cell dependence of aversive behaviors in the setting of a type 2 immune response here would be important - after IL-25 injections and/or helminth infections using *Pou2f3*^{-/-} mice (these mice were already used here for organoid studies).

3. Tuft cell-Ach dependence in vivo – it would be helpful to see the cFOS staining of the brainstem and behavioral studies in mice where tuft cells cannot produce Ach after IL-25 injections to secure the tuft cell-Ach part of the loop. The authors already have *Pou2f3*-cre mice, which they can use to breed with *Chat*^{fl/fl} mice (commercially available) to generate a tuft cell specific deletion and confirm that deficiency of Ach generation in tuft cells leads to altered behavior. Tuft cells generate a number of additional mediators that can potentially activate both neighboring epithelial cells and sensory neurons directly, so a role for Ach is not obvious without a conditional deletion study.

To address points 2 and 3, we employed tuft cell-deficient (*Pou2f3*^{-/-}) mice and gut tuft cell-specific ChAT-deficient mice (*Vil*^{Cre};*Chat*^{fl/fl}) to perform comprehensive analyses, including food intake measurements, behavioral assessments, and cFOS staining following helminth infections (Fig. 6b-d and Extended Data Fig. 9-10). We observed that both food intake reduction and aversive behaviors are dependent on tuft cell-derived ACh in these settings. Additionally, cFOS signals in the brainstem were significantly decreased in both knockout mouse lines, confirming the requirement for tuft cells and tuft-derived ACh in this nocifensive pathway.

Regarding the Florsheim et al. study, their findings demonstrate that allergens can directly stimulate immune cells in the gut to evoke avoidance behaviors through a mast cell-dependent pathway. However, our work identifies a distinct tuft cell-ACh-dependent mechanism that mediates food aversion during parasitic infections. These represent parallel pathways that utilize different molecular mechanisms to achieve similar protective outcomes. While both pathways ultimately induce type 2 inflammation, they employ distinct cellular intermediates and signaling molecules. Our tuft-EC-vagal pathway appears particularly relevant for parasitic infections where tuft cells serve as the primary sensors for parasite-derived metabolites. We believe that future studies investigating how these complementary pathways might synergize during complex parasitic infections would provide valuable insights into the redundancy and integration of gut protective mechanisms.

4. Consideration of immune cell (e.g. ILC2) derived Ach as activator of EC cells - although IL-25 is dominantly derived from tuft cells, this cytokine has pleiotropic effects on immune cells– most prominently on type 2 innate lymphoid cells (ILCs). Immune cell involvement here is not evaluated at all in vivo. This would be relevant as ILC2s in the setting of type 2 inflammation including after IL-25 injections have been shown to upregulate *Chat* (see PMID 33674322). Thus, the experiments in Fig. 5B/C showing effect on serotonin activity might be reflecting either tuft cell or ILC2-derived Ach as the atropine inhibition does not account for the source. The authors could consider immune cell ablation in the biosensor mice to account for that.

This is an excellent point. To address this concern, we generated serotonin sensor mice that 1) lack tuft cells (*Vil*^{Cre};*gGRAB*_{5-HT3.0};*Pou2f3*^{-/-}); 2) lack ACh specifically in intestinal tuft cells (*Vil*^{Cre};*gGRAB*_{5-HT3.0};*Chat*^{fl/fl}), and performed serotonin biosensor imaging. We indeed found that tuft cells and tuft-derived ACh are indispensable for increased serotonin release in both acute stimulation (by succinate or TRPM5 agonist) or upon type 2 inflammation (Fig. 4c-d and 5b-e). We believe that these experiments pinpoint tuft cell-derived ACh and minimize contributions from other possible sources of ACh.

5. The possibility of a direct effect of tuft cell-derived Ach on vagal neurons is not discussed. Tuft cell-derived Ach was shown to directly activate vagal sensory neurons innervating the trachea (PMID: 21606356, 35503420). This possibility is not assessed nor discussed.

This is an intriguing point. To address this, we traced mucosal afferents from the small intestine and performed Ca^{2+} imaging and RT-PCR to assess expression of acetylcholine receptors. We indeed found that a subset (28%) of mucosal vagal afferents are directly sensitive to ACh (Extended Data Fig. 6b-d). However, when we performed mesenteric afferent recordings to measure the contribution of direct ACh activation vs. ACh-stimulated EC-

derived serotonin activation, we found that ACh mostly stimulates mucosal vagal afferents in an alosteron-sensitive manner and thus via EC cells, which supports our mechanism of the tuft-EC axis being a major driver of the nociceptive pathway (Extended Data Fig. 6e-h).

6. Non-canonical pathway of Ach release by intestinal tuft cells. Although there is little doubt that tuft cells can generate Ach, as has been shown by multiple groups in the past extensively for the airways, gallbladder, urethra (Perniss and Kumer 32294408, Deckmann and Kummer, Krasteva and Kummer, Keshavarz and Kummer 35245090), proposing a new Ca²⁺ independent mechanism needs stronger data to be conclusive: This concept is intriguing but the mechanism is unclear. The mechanism of GSDMC playing a role in tuft cell Ach release is not well defined, especially given that it is not expressed by the tuft cells themselves but rather in neighboring enterocytes or other epithelial cells. As the authors point out, IL25 is known to induce Gsdmc1-4 expression in enterocytes (Xi et al, PNAS 2021), which was shown to induce lytic cell death in worm induced immunity. The authors do not seem to actually reveal a mechanism of Ach release, only that neighboring GSDMC⁺ cells are somehow driving this. Is cell death of enterocytes somehow activating tuft cells to release Ach? Are tuft cells themselves undergoing cell death? Or are there DAMPs released like ATP that is driving Ach release?

As described above, we have removed the gasdermin C component and instead focused on characterizing the unconventional biophysical properties of tuft cell ACh release, which provides important new insights into non-neuronal neurotransmitter release mechanisms. This new information now provides the impetus to explore this enigmatic question in a future biophysical study.

7. Robust and reproducible in vivo data. The differences in food intake, which is the only major physiological readout, is only occurring at day 6-7 in PFTox mice, and it is not very robust. It is mainly driven by females (n=3-4 mice per group). Can the authors repeat this with a second cohort of mice to show that it is reproducible and not due to one specific cohort?

We performed additional experiments to increase the sample size and enhance the robustness of our findings. We now have N = 5-7 and 6-8 mice per group for females and males, respectively, with the conclusion still holding that reduction in food intake is at least partially mediated by EC cells (Fig. 6a). The effect remains statistically significant and reproducible across multiple cohorts.

Although we also observed sex differences in spontaneous mouse behaviors (Fig. 6d and Extended Data Fig. 10), we were unable to perform sufficient food intake measurements with helminth infection to adequately power sex-specific analyses due to experimental constraints and resource limitations. We therefore combined males and females for the *Nippostrongylus brasiliensis* infection paradigm (Fig. 6b-c).

8. In general, in their discussion of tuft cell Ach generation and release, the authors fail to acknowledge that 1) tuft cells in the intestine, gallbladder and airways were shown in many previous studies to express choline acetyltransferase (PMID: intestine – 26675736, airways -26033492, 32294408, 35245090, 22749863) and generate Ach (PMID: 32294408, 35245090 – shown by mass spectrometry; 31914675 – similar biosensor system to the one used in this paper); 2) tuft cell derived Ach has an established role in mediating Ca²⁺ flux in neighboring epithelial cells which drives mucociliary clearance and importantly activates neighboring epithelial cells for Ca²⁺ flux (PMID: 37531421). 3) Tuft cell-derived Ach in the airways also as mentioned above was shown to trigger sensory neuron activation. In that context, the findings here are still novel as they suggest a tuft cell to EC cell to sensory neurons communication but the previous literature should still be acknowledged.

We appreciate this feedback. We now discuss tuft cell-derived ACh in other organs in the last paragraph of the discussion (line 314). Indeed, in light of our observations, it will be interesting to determine whether this unconventional mechanism of ACh release is conserved across different tuft cells in visceral organs.

MINOR CONCERNS:

1) ChAT expression in organoid tuft cells. Many experiments in Fig. 2 and 3, using organoids employing “single cell biosensor measurements” where as far as I understand the release of Ach by single tuft cells marked by either TPRM5 or Pou2f3 was assessed. The assumption here is that all Pou2f3 and/or Trpm5 expressing tuft cells in organoids also express choline acetyltransferase (Chat) and can generate Ach. ChAT is not ubiquitously expressed

by intestinal tuft cells as shown with ChAT-GFP mice in a recent study from Billip and von Moltke (PMID: 38744291 and cited by the author as bioRxiv). The authors should assess the expression patterns of ChAT in organoid tuft cells to confirm that it is highly expressed across differentiated tuft cells, and only use the single cell biosensor measurements for cells that do express ChAT, i.e from ChAT-GFP mice. Or show that cell lysis does lead to release of Ach from the tuft cells.

2) The authors should also demonstrate that the succinate receptor is indeed expressed in the in vitro developed tuft cells in organoids since they show lack of Ach release with succinate activation. This is inconsistent with three recent studies (PMID: 38744292, 37531421, 38744291) and thus intriguing but receptor expression in all organoid tuft cells should be confirmed.

To answer these concerns, we now show that tuft cells release ACh in response to the TRPM5 agonist in both acutely dissociated settings and in organoids (Fig. 2). However, succinate stimulates ACh release only from tuft cells dissociated from intact intestinal tissue, not from tuft cells in organoids, which represent crypts. We then found that succinate receptors are only expressed in villus tuft cells (Extended Data Fig. 3a-b), consistent with recent single-cell RNA sequencing analysis demonstrating spatial heterogeneity of tuft cell populations (PMID: 40695798). We believe these data now reconcile our findings with published papers from the von Moltke lab and Jay labs showing that succinate stimulates ACh-mediated fluid secretion (PMID: 38744291 and PMID: 38744292), while also revealing an interesting new facet pertaining to the spatial specification of succinate signaling.

3) Liquid chromatography/mass spectrometry to show the specificity of the Ach measurements in parallel to validate the biosensor assessments would be nice. This methodology to detect Ach was shown recently to work for sorted intestinal tuft cells and intestinal lavage (PMID: 38744292), tracheal (PMID: 32294408) and gallbladder samples (PMID: 35245090). Mass spec analysis of organoids should not be prohibitively difficult.

We appreciate this suggestion, but we believe the strength of our study is the use of gGRAB_{ACh4h}, which was designed on the basis of native ACh receptors and thus provides excellent spatial and temporal dynamics for measuring ACh release that goes beyond prior measurements of bulk transmitter release. Moreover, we believe that our new experiments using tuft cell-specific ChAT-deficient mice convincingly validate the specificity of the gGRAB_{ACh4h} reporter.

4) Alternatively, would suggest showing the critical experiments with organoids derived from Pou2f3Cre-Chatfl/fl mice where Ach generation would be specifically deleted in tuft cells.

We have now performed ACh biosensor experiments with single isolated tuft cells, which excludes the contribution from other cells (Fig. 2d-e). Furthermore, we performed ex vivo serotonin sensor experiments to show that tuft cell-derived ACh is essential for tuft-EC axis activation (Fig. 4c-d and 5b-e). In conjunction with recent publications where ChAT is only present in tuft cells in the intestinal epithelium (PMID: 38744291 and PMID: 38744292), we believe that our study supports the source of ACh as being from tuft cells.

5) Can the authors also show the numbers of cFOS⁺ neurons in Area Postrema (AP) without and with IL25 injection, and if that is affected by PFTox ablation? The reason is that the AP plays a major role in nausea and aversive behaviors, but the authors do not focus on this even though that could be a key factor in their behavioral outcomes.

We thank the reviewer for this important point. As noted in our response to Referee 1, Question 11, we quantified cFOS⁺ neurons in the AP (and DMV) under all experimental conditions (Extended Data Fig. 7a). IL-25 injection did not significantly change AP cFOS⁺ counts in either PFTox(-) or PFTox(+) mice, with greater variability observed across individuals than seen in the nTS. These results suggest that under our experimental conditions, AP activity is not strongly engaged and is unaffected by PFTox ablation. Thus, we cannot claim that AP-mediated nausea or aversion per se contributes substantially to the behavioral outcomes.

6) In supplemental Fig. 6B, the IL25 bottom right image shows very high background (red staining) for cFOS

throughout the brainstem section compared to staining in the saline section above. Is background fluorescence normalized or how was imaging performed?

We thank the reviewer for this technical question. In the revised manuscript, we provide a more comprehensive colocalization analysis of cFOS⁺ neurons in the nTS (Extended Data Fig. 7b–c), rather than showing only cFOS and Tac1 colocalization. We also updated the related figure panel (Fig. 5j). All representative images were prepared in the same batch and imaged under identical acquisition settings to ensure comparability across groups, and the same parameters were used when generating image outputs. We note that IL-25–treated mice showed higher background fluorescence in the AP region compared to other groups, perhaps reflecting their inflammatory state; however, this background did not correspond to specific cFOS⁺ staining, as confirmed by lack of colocalization with DAPI. Importantly, all quantitative analyses were performed on background-subtracted data, ensuring that this variation did not affect the results.

7) Can the authors explain why there is positive staining within the organoid in Fig. 1F (red area in the center where the organoid is marked)? Are these HEK gGRAB cells that randomly landed in the middle of the organoid?

Indeed, we plated biosensor HEK cells on top of organoids, therefore, some HEK cells are positioned on top of organoids, which we now clearly describe in the methods section.

8) Fig. S2A- can the authors show the single stain pictures and quantitate the overlap in several tissue sections.

We now solely rely on TRPM5-GFP organoids, as we found that *Pou2f3*^{CreER} organoids lose their specificity when passaged. We hope this addresses the referee's concern about specificity.

Referee #3 (Remarks to the Author):

Touhara and colleagues investigate the impact of type 2 immunity on nocifensive behavior by focusing on the connection between the specialized intestinal epithelial cells, tuft cells and enterochromaffin cells (EC). Tuft cells have emerged as critical sensory cells in detecting intestinal parasites and initiating type 2 immunity through an array of effectors, including acetylcholine (ACh). While most research has focused on tuft-immune cell interactions, this manuscript found intestinal tuft cell derived ACh stimulates muscarinic receptor 1 (M1R) expressed by enterochromaffin cells (EC), thereby inducing the release of serotonin. In turn, serotonin from ECs stimulates the vagus nerve, activating neurons in the nTS brain stem as measured by cFOS staining. Genetic ablation of ECs reduced cFOS staining in the nTS region and the associated decline in food intake during type 2 immunity.

Recently tuft cell derived ACh was shown to induce Cl⁻ secretion from neighboring enterocytes (PMID: 38744291), but to my knowledge, this manuscript provides the first evidence of a connection between tuft cells and ECs. Furthermore, intestinal helminth infection reduces appetite, and this manuscript introduces a potential mechanism for this observation. The quality of experiments and overall interpretation of the data is excellent. The use of numerous biological sensors to measure ACh and HT-5 release and sophisticated mouse models are impressive and generate convincing data. Nevertheless, several issues diminish my enthusiasm but could be reasonably addressed to strengthen this otherwise interesting manuscript. Please see my specific concerns below.

We appreciate the referee's positive assessment of our experimental approaches and data quality. His/her concerns about the ACh release mechanism and the need for physiological validation were well-founded and have guided our revisions. We have removed the speculative gasdermin C component and instead focused on providing robust mechanistic insights into tuft cell ACh release properties (Fig. 2). The addition of helminth infection studies and genetic validation experiments has substantially strengthened the physiological relevance of our findings (Fig. 4c-d, 5b-e, and 6).

Major points

1) The authors propose tuft cells release ACh as a prolonged leak through a GasderminC (Gsdmc) dependent

mechanism. While this paper nicely establishes the importance of Gsdmc, the mechanism for this pore and ACh release is very vague. First, does ACh passively leak out, or do type 2 cytokines promote active and sustained release? Tuft cells express IL4Ra; therefore, adding IL4 to organoids could induce tuft cell differentiation and Gsdmc expression and stimulate tuft cells ACh release rather than as a slow passive leak. This possibility could be addressed by crossing the Pou2f3-Cre used in this paper with an IL4ra-flox mouse. Additionally, chemically blocking gap junction function could test the hypothesis proposed in the discussion (line 250) that tuft cells pass ACh to neighboring cells for release through Gsdmc. It is equally possible that Gsdmc allows release of other epithelial effectors that induce direct tuft cell release of ACh. Much of the author's argument against conventional vesicular release of ACh stems from the absence of vesicular transport genes, but tuft cells also lack Gsdmc, so this unconventional mechanism suffers from the same absence in cellular machinery.

As described above, we removed the gasdermin C component from our manuscript as we cannot definitively establish its contribution to ACh release. Instead, we now provide mechanistic insights into how a single isolated tuft cell can release ACh without synaptic release machinery and electrical excitability, which we believe provides valuable new insights into non-canonical neurotransmitter release (Fig. 2 and Extended Data Fig. 2). These observations now provide a rigorous foundation for exploring this enigmatic question in a future biophysical study.

2) This manuscript doesn't adequately address the competing hypothesis in the literature that tuft cell release ACh through calcium flux and Trpm5 membrane depolarization (PMID: 38744291). This could be easily tested using a Trpm5-KO mouse, which is both commercially available and widely found in the academic community. Additionally, Billipp et al. 2024 (PMID: 38744291) used a Trpm5-agonist to induce ACh release, which would strengthen this manuscript's conclusions if they found this agonist didn't induce ACh release in their system.

We appreciate this important point. Unfortunately, we could not obtain TRPM5 knockout animals, however, a selective TRPM5 agonist (agonist 39) and antagonist (NDNA) became commercially available after our initial submission. Using these new pharmacological tools, we found that single isolated tuft cells can release ACh in response to succinate or agonist 39, which was blocked by NDNA (Fig. 2d-f). We believe that these data now reconcile our observations with the recent publication by Billipp et al., (PMID: 38744291) while providing additional mechanistic insight into the biophysical properties of tuft cells and their unconventional ACh release mechanism.

3) The experiments using thapsigargin and succinate to induce tuft cell calcium flux and disprove this mechanism for ACh release are slightly problematic. While thapsigargin induces a Ca²⁺ flux, does this activate TRPM5 in tuft cells? Additionally, succinate was applied to the basolateral side of organoids, when evidence suggests apical succinate is required to stimulate tuft cells. This may be why the calcium flux response was rather weak in succinate-treated organoids. Could these experiments be repeated with flipped organoids (or again use the Trpm5 agonist if organoids are basolateral out)?

We appreciate these insightful comments. As described above, we now found that either succinate or the TRPM5 agonist can evoke ACh release from single-isolated tuft cells, despite their lack of an excitable membrane (Fig 2d-f). We believe that these data now reconcile our results with those of Billipp et al., while providing additional mechanistic details on the biophysical properties of tuft cells and their unconventional ACh release mechanisms. Regarding the polarity issue, we found that succinate receptors (*Sucnr1*) are only expressed by villus tuft cells (Extended Data Fig. 3a-b), which explains why succinate fails to stimulate ACh release from tuft cells in crypt-like organoids. Our findings indeed support the paradigm that succinate receptor activation increases intracellular calcium, which then activates TRPM5 channels to trigger ACh release from tuft cells through an unconventional, non-vesicular mechanism.

4) Using recombinant IL25 to induce in vivo type 2 immunity is okay for some experiments but leaves substantial ambiguity about the importance of tuft cell derived ACh and intestinal specific effects of ECs. Could other gut intrinsic sources of ACh be stimulated with IL25 or downstream type 2 cytokines? Possibilities include cholinergic intrinsic neurons or intestinal ILC2s, which upregulate ChAT in response to IL25 (PMID: 33674322). Crossing the Vil-cre from this paper with the commercially available Chat-flox would be a nice way to address the source of ACh, but IL25 injections of Pou2f3^{-/-} mice would also get at the importance of tuft cells to cFOS

staining and appetite suppression. Additionally, does Tac1-Cre;ePet-Flp;PFTox mice ablate ECs in other tissues besides the gut? If the vagus nerve innervates these tissues, then would systemic IL25 obscure the gut-specific effects of EC stimulation? Using a type 2 immune stimulus restricted to the intestine would strengthen these experiments.

We appreciate this concern, which we now address through the use of tuft cell-deficient (*Pou2f3*^{-/-}) and gut tuft cell-specific ACh-deficient (*Vil*^{Cre}; *Chat*^{fllox/fllox}) mice across multiple experimental paradigms, including serotonin sensor measurements (Fig. 4c-d and 5b-e), food intake assays (Fig. 6b-c), cFOS staining (Extended Data Fig. 9), and behavioral tests (Fig. 6d and Extended Data Fig. 10). We found that both IL-25 and helminth infection increased baseline serotonin levels in crypts, which were abolished in both tuft cell-deficient and gut tuft cell-specific ACh-deficient mice (Fig. 5b-e). Similarly, *N. brasiliensis* infection reduced food intake in wild-type mice at the peak of tuft cell hyperplasia (days 7-9), whereas tuft-deficient and gut tuft cell-specific ACh-deficient mice maintained normal feeding patterns (Fig. 6b-c). cFOS staining was consistent with the food intake data, showing increased signals in control animals but less pronounced activation in knockout animals (Extended Data Fig. 9). These experiments establish that tuft cell-derived ACh is the key mediator of our proposed pathway. As for the specificity of PFTox mice, we have previously performed a systematic analysis of the expression of TeNT in other tissues (PMID: 36949192) and confirmed the specificity. Together with the other genetically modified lines that we used (tuft-deficient and gut epithelial ChAT-deficient mice), we believe that we provide substantial evidence for the specificity of our findings and proposed signaling pathway.

5) As referenced above, intestinal helminth infection experiments would greatly improve this manuscript. These worms could provide a relevant intestinal type 2 stimulus. Additionally, parasites are referenced in the abstract, throughout the text, and in the model (Fig. 5I), but systemic IL25 injections do not adequately model these pathogens.

We appreciate this suggestion. As noted above, we have now performed substantial experiments using *Nippostrongylus brasiliensis* infection, including *ex vivo* serotonin biosensor assays (Fig. 5d-e), food intake measurements (Fig. 6b-c), and cFOS quantification (Extended Data Fig. 9) in the brainstem using tuft cell-deficient and gut tuft cell-specific ACh-deficient mice. This significantly increases the physiological relevance of our study and validates our IL-25 findings in a natural infection context.

Minor points

1) The loss of ECs on reduced food intake during type 2 immunity is nice but could additional nocifensive metrics be included to strengthen this aspect of the story?

We agree that additional nocifensive metrics strengthen this aspect of our study. Beyond food intake reduction, we systematically measured spontaneous behaviors during IL-25-induced type 2 inflammation, including locomotion, grooming, orienting, and rearing behaviors (Fig. 6d and Extended Data Fig. 10). We found that wild-type female mice exhibited increased grooming behavior, which is an established indicator of stress or discomfort in rodents, along with reduced locomotion and rearing. These behavioral changes were diminished in PFTox mice (EC knockdown) and epithelial ChAT-deficient mice, confirming their dependence on the tuft-EC axis.

Additionally, we demonstrated activation of the nucleus tractus solitarius (nTS) in the brainstem following type 2 inflammation, which was absent in PFTox mice (Fig. 5i-k). The nTS is a key brain region involved in processing visceral sensory information and mediating aversive responses. We also validated our findings using physiologically relevant *Nippostrongylus brasiliensis* infection, showing that both food intake reduction and nTS activation depend on tuft cells and tuft-derived acetylcholine (Fig. 6b-c, Extended Data Fig. 9a-b). Together, these multiple behavioral and neuronal readouts provide robust evidence that the tuft-EC axis mediates a coordinated nocifensive response during type 2 immune activation.

2) The authors suggest that tuft cells could avoid cell death by not expressing *Gsdmc* (lines 250-253). However, *GsdmC* was shown to allow IL33 release by intestinal epithelial cells independent of cell death (PMID: 35385697). While *Gsdmc2* expressed in HEK cell line triggered pyroptosis (PMID: 34290141), it's unclear

whether these experiments are more relevant than the in vivo data from the first paper. The authors could incorporate both perspectives in their discussion.

As described above, we removed the gasdermin C component from our manuscript as we cannot definitively establish its contribution to ACh release. Instead, we provided mechanistic insights into how a single isolated tuft cell can release ACh without synaptic release machinery and electrical excitability.

3) The figure legends refer to gray traces in many biosensor experiments, but several figures have black traces. Could the figure legend be revised for consistency with the figures.

Thank you for catching this. We revised the figures for consistency.

4) While the manuscript clearly shows that tuft cells don't directly release ACh through gasdermin C pores, the abstract (lines 22-23) could lead some casual readers to believe this was the case. Could the writing be refined in the abstract to avoid this confusion?

We removed the gasdermin C component from our manuscript, so this has been removed from the abstract as well.

We thank the reviewers for their thoughtful evaluation of our revised manuscript. We are gratified that Referees 2 and 3 found our revisions compelling and that they recognize the significance of our findings on epithelial crosstalk during type 2 immune responses. We also appreciate Referee 1's careful reading and constructive criticism, which has helped us further refine our manuscript and clarify our conclusions.

In response to Referee 1's comments, we have made several important changes to strengthen the manuscript (major changes to the text are indicated in **red type**):

1. We have refocused the behavioral component and claims within the manuscript to focus primarily on food intake, which is strongly supported by all aspects of our data. Assessment of other behaviors is now de-emphasized and discussed with relevant limitations and caveats in mind, reflecting the fact that assessing such behaviors in the context of type 2 inflammation and dysregulation of the small intestine remains challenging.
2. We have performed additional behavioral experiments and consolidated all behavioral data, including previous measurements on spontaneous behaviors, in Extended Data Fig. 10, focusing the main figures on the well-supported food intake phenotype and its underlying mechanisms.
3. We have added caveats in the Discussion section acknowledging the potential contribution of other pathways beyond the tuft-EC-vagal axis (**line 309**).
4. We have replaced the representative cFOS image in Fig. 5 with data from an independent experimental cohort to address concerns about background fluorescence in the area postrema.

While we acknowledge Referee 1's concern that removing the gasdermin C hypothesis reduces some of the initial excitement, we believe the revised manuscript actually presents a more robust and biologically important story that reveals (i) how biphasic ACh release from tuft cells relates to phases of parasite infection, (ii) how spatial dynamics of tuft-EC interactions change during infection to amplify this detection/effector circuit, and (iii) how a neuroimmune signaling pathway from the gut epithelium to the brainstem alters food intake. We believe that this represents a novel and significant advance in understanding how parasitic infections influence host behavior. Moreover, our characterization of non-canonical ACh release from tuft cells provides an important new insight for the field and a rigorous foundation for future studies aimed at deciphering the precise mechanism underlying this novel and intriguing finding.

Detailed responses to Referee 1's specific comments are provided below.

Referee #1 (Remarks to the Author):

Touhara et al. have revised their manuscript in response to the comments on the previous version of their manuscript. They have also greatly modified the claims and conclusions they derive from the data. Specifically, they now present a different (biphasic) model for how acetylcholine release by tuft cells influences serotonin release by enterochromaffin cells and the behavioral consequences of parasitic infection.

The upside of these changes is that the new model is better supported by the experimental data. The downside is that, somewhat disappointingly, the most exciting aspects of their previous model have now disappeared, including the intriguing idea of lateral acetylcholine signaling between tuft cells and enterochromaffin cells through gasdermin C pores, presumably because it is no longer in line with the experimental evidence. The critical question of how acetylcholine is released by tuft cells in the absence of a synaptic machinery thus remains open.

While it is of course important that the conclusions are now more solid, the advancements that the revised manuscript makes for our understanding of type II immune responses are now much less exciting. Enthusiasm is further dampened by the fact that the authors have in the meantime published their new tools for enterochromaffin cell activity monitoring in a separate paper, thus removing an additional major advancement from this manuscript. What is left is a thorough investigation of epithelial-neuronal crosstalk in type II immunity that presents several interesting insights but by itself is a less substantial advancement for the field, particularly since the exploration of infection-induced behavioral changes is still rudimentary and the new data is rather unconvincing.

We thank Referee 1 for thoughtful and critical evaluation of our revised manuscript. We appreciate his/her acknowledgment that our revised model is now better supported by the experimental data and that our conclusions are more solid. We address below the concerns raised about the overall impact and novelty of the work.

While we removed the gasdermin C model based on recent results that call this hypothesis into question, we respectfully disagree that this makes the manuscript substantially less exciting or impactful. The revised manuscript presents several major advances that represent significant contributions to our understanding of how parasitic infection of the gut modulates neuro-immune responses:

First, we now provide direct evidence that isolated tuft cells release ACh through a non-canonical mechanism, operating via both acute TRPM5-dependent and prolonged cytokine-induced pathways. Through rigorous patch-clamp analysis and biosensor imaging, we demonstrate that isolated tuft cells lack conventional synaptic machinery yet robustly release ACh. This biphasic release model explains how parasitic infections elicit graded physiological responses over time, with initial acute ACh release driving immediate EC cell activation, followed by sustained ACh production that stimulates the gut-brain axis during chronic infection. Determining the precise molecular mechanism of this non-canonical release will require extensive biophysical and biochemical studies that are beyond the scope of this work. However, we believe that establishing both the unconventional nature of ACh release and its biphasic kinetics represents a significant advance that provides novel insight into how intra-epithelium communication orchestrates physiological responses to infection.

Second, while we have described the serotonin sensor and its basic applications in a companion *Nature* paper, the current manuscript demonstrates the power of integrating such tools with electrophysiological, anatomical, metabolic, and behavioral measurements to map a functional interoceptive pathway that elicits physiological and behavioral responses to a clinically relevant maladaptive condition.

Regarding the behavioral studies, we acknowledge the limitations raised by Referee 1 and have revised the manuscript accordingly. We have refocused our results and conclusions on food intake—a well-supported phenotype with clear demonstration of tuft cell, EC cell, ACh, serotonin, and nTS involvement—rather than making broader claims about aversive behaviors. We believe this represents responsible scientific reporting that accurately reflects what our data can and cannot support, while also noting the limitations of assessing visceral discomfort associated with dysregulation of the small intestine (line 307).

Specific comments:

1. The authors now propose a distinction between acute TRPM5-dependent and prolonged TRPM5-independent modes of acetylcholine release. How are these different mechanistically? Now that the part about *Gsdmc* has been removed from the manuscript, it is unclear from the authors' explorations how acetylcholine leaves tuft cells and what causes the mechanisms underlying acute and prolonged release to be distinct.

As noted by the referee, further experimentation ruled out an essential role for Gasdermin C in mediating ACh release from tuft cells, and thus, we pulled back from this hypothesis and decided to carry out a novel series of patch-clamp and biosensor analyses to rigorously confirm the non-canonical nature of ACh release. Irrespective of a role for Gasdermin C, our overall conclusion that ACh is released by a unique mechanism still stands and represents an interesting and novel aspect of our study, as does the fundamental conclusion that ACh is released directly from tuft cells to initiate sensory signaling.

While we greatly share the referee's interest in this process, rigorously determining the precise release mechanism will involve a separate project requiring detailed biophysical and biochemical approaches, which we believe falls outside the scope of the current study. The overall focus of this study centers on determining how parasitic infection promotes the activation of sensory nerve fibers to suppress food intake. In fact, in our revised manuscript, we have expanded our model to explain how bi-phasic release of ACh leads to temporal changes in symptomology with infection. We believe that this represents an important, novel, and rigorous contribution, as noted by Referees 2 and 3.

2. The NTS analysis in Extended Data Figure 7 is used to make the argument that NTS neurons activated by type II immune responses are distinct from those activated by aversive stimuli, but it is unclear how this conclusion was reached. The data appear to be from GSE200003, which is an experiment using intranasal house dust mite treatment. How is this related to NTS neurons that receive input from gut-innervating vagal neurons? How have the authors defined NTS neurons activated in their study?

We thank the reviewer for this comment and believe there is a misunderstanding. We did not use the PBS- or HDM-treated datasets from GSE200003. Instead, we analyzed the nTS reference atlas generated from naïve wild-type mice (samples GSM6001919, GSM6001920, GSM8091867, and GSM8091868) reported in the same study, solely to identify marker genes for specific nTS neuronal populations (Extended Data Fig. 7b). We then assessed the activation of these populations in our experiments using cFos quantification via RNAscope (*Dbh*, *Nr4a2*, *Npy*), immunofluorescence (*Cck*), or reporter analysis (*Tac1^{Cre};Ai14* mouse line). The summarized activation data are shown in Extended Data Fig. 7c. We have now clarified this issue in the figure legend.

3. In the title of the manuscript and throughout the manuscript text, the authors refer to aversive behavior, but the behavioral experiments shown in Figure 6 (or anywhere else in the manuscript) do not address the question of aversion. Several behavioral assays are available in mice to directly study aversion and nausea, and the brain areas associated with aversive behavior have been identified. The authors should make use of all these behavioral tests and monitor the activity of these brain regions in the setting of

helminth infection in order to support claims made about aversive behavior as well as the involvement of the tuft cell-enterochromaffin cell crosstalk therein.

We thank the referee for raising this important point about our behavioral assays. To address his/her concern, we investigated abdominal mechanical sensitivity (measured by the von Frey test) and nesting behavior (to assess general well-being and overall activity) during type 2 inflammation (Extended Data Fig. 10c-d). However, we did not observe significant changes in either measure, which is not surprising for a chronic infection scenario that typically does not produce localized acute pain. Other studies have used retching behaviors and conditioned place preference assays following acute interventions, such as injection of bacterial toxins (PMID: 36323317). However, IL-25 injection and helminth infection induce chronic illness rather than acute responses. Unlike bacterial toxin injections that provide a discrete, time-locked aversive stimulus, the gradual onset and sustained nature of type 2 inflammation make it difficult to establish the temporal associations required for conditioned place preference and similar behavioral paradigms. Moreover, as shown in Figure 6b-c, helminth infection causes a dramatic drop in food intake 1-2 days post-infection, corresponding to helminth migration from blood vessels to the trachea. This inflammatory process further complicates the interpretation of behavioral experiments related to aversion and nausea.

In light of the referee's comments and our additional behavioral experiments, we agree that our data do not definitively demonstrate that type 2 immune responses induce behaviors that can be generally described as aversive or nocifensive. Therefore, we have revised the title, abstract, and manuscript text to remove references to aversive and nocifensive behaviors, replacing them with the more specific and accurate terms such as "reduced food intake." We have moved all other behavioral analyses to Extended Data Fig. 10 and now focus solely on food intake measurements in Fig. 6. We have also explicitly noted in both the Results and Discussion sections (lines 276 and 307) that the exact sensations experienced during type 2 inflammation remain to be determined. In any case, we have clearly demonstrated that helminth infection reduces food intake through activation of the tuft-EC axis and nTS neurons, which we believe represents an important mechanistic advance in understanding how parasitic infections modulate host behavior.

4. The new behavioral data that is instead included in the manuscript (spontaneous behavioral monitoring), is quite unconvincing for multiple reasons:

Thank you for raising this point. As described above, we agree that our behavioral data do not definitively demonstrate that type 2 immune responses induce aversive behaviors, and we have acknowledged in the manuscript that the exact sensations experienced during type 2 inflammation remain unclear (lines 276 and 307). Therefore, we have refocused our study on food intake. Nonetheless, we address below each specific concern that the referee has raised about the behavioral data.

a) In female mice, *Pou2f3*-deficient animals show the same overall behavior as control mice. The most straight-forward conclusion from this observation would be that tuft cells are not involved, suggesting that the observed behaviors are unrelated to the mechanisms explored in the rest of the manuscript.

We would like to note that in female mice, grooming behaviors are observed only in control mice but not in genetically modified mice, including *Pou2f3*-deficient animals, which we believe reflects disruption of the tuft-EC-NTS pathway (Extended Data Fig. 10b). However, as the referee mentioned, locomotion and rearing behaviors persist in *Pou2f3*-deficient animals. We attribute this to the fact that tuft cells are present not only in the intestine but also in other organs, including the trachea, nasal cavity, and gallbladder, and their global absence could affect differentiation of other epithelial cell types throughout these tissues. We consider the intestine-specific knockout of ACh in tuft cells (*Vil^{Cre};ChAT^{fl/fl}*) to be a much cleaner intervention, and indeed, all behavioral phenotypes are abolished in these mice. Furthermore, IL-25 initiates the type 2 immune cascade, which could affect other pathways that influence spontaneous behaviors, including enteric neurons (via communication with ILC2 cells; PMID: 40403128) and direct effects of IL-25 on the central nervous system (PMID: 40199322). Given these confounding factors and the inconsistent effects on locomotion and rearing behaviors across genotypes, we now consider the spontaneous behavior data to be inconclusive regarding the nature of aversive sensations during type 2 inflammation. We have explicitly noted that the exact sensations experienced remain unknown (lines 276 and 307), and added discussion of potential contributions by other IL-25-induced pathways (line 309).

b) The situation in males is even more confusing: Control mice are unaffected by IL-25 injection, but behavioral differences start to appear in PFTox(+) and *Pou2f3*-deficient mice. Does this mean that tuft cells and enterochromaffin cells buffer the effect of IL-25 on behavioral changes? This would be the opposite conclusion compared to the authors' final sentence which is that "the tuft-EC axis sends signals to the nTS via vagal afferents, leading to aversive behaviors during the type 2 inflammatory response".

We agree that the situation in male animals is more complex. We have noted in the main text that type 2 inflammation could contribute to behavioral changes through other IL-25-induced pathways beyond the tuft-EC axis (line 309), and that lower ILC2 numbers in males due to androgen suppression (PMID: 28484078) may also contribute to the different response patterns compared to females (line 271). In light of these confounding factors, we have de-emphasized the sex differences in spontaneous behaviors and explicitly

acknowledged in the manuscript that our behavioral assays cannot definitively demonstrate that type 2 inflammation induces aversive behaviors beyond reduced food intake (lines 276 and 307).

5. The functional involvement of vagal neurons in behavioral adaptations to type II inflammation has still not been demonstrated. Chemogenetic, optogenetic, or surgical interventions would be required to substantiate these claims, which was brought up in the comments on the initial version of the manuscript.

We respectfully disagree with this assessment. We have demonstrated that type 2 inflammation induces serotonin release from EC cells, which then activates mucosal afferents in the small intestine. We have previously shown that nearly all (> 95%) mucosal afferents innervating the small intestine are of vagal origin (data shown below; PMID: 39939779) and that 92% of these neurons express 5-HT₃ receptors. Furthermore, we have now demonstrated using afferent fiber recordings from the mesenteric nerve of the small intestine that these afferents are activated during type 2 inflammation in a serotonin-dependent manner, as responses were attenuated by application of a 5-HT₃ receptor antagonist (Fig. 5f-h and Extended Data Fig. 6). We then confirmed that this pathway activates the nTS region of the brainstem, which is the primary projection target of vagal neurons (Fig. 5i-k and Extended Data Fig. 7).

We believe that chemogenetic or optogenetic manipulations of vagal neurons would be difficult to interpret in our model. These approaches require abdominal surgery and local injections of retrograde AAVs, which could themselves substantially affect food intake behavior—a highly sensitive readout that can be influenced by surgical stress and general malaise. Similarly, vagotomy would affect not only the small intestine but also multiple other organs, including the stomach, large intestine, liver, and cardiovascular system, making it difficult to attribute effects specifically to intestinal vagal signaling. Nonetheless, we acknowledge that we cannot exclude contributions from other pathways, including enteric neurons activated by ILC2 cells (PMID: 40403128) and direct effects of IL-25 on central pathways outside the nTS (PMID: 40199322). We have therefore added this caveat to our Discussion (line 309).

Extended Data Fig. 4b from PMID: 39939779: Mucosal afferents innervating the small intestine (which communicate with EC cells) were retrogradely labeled, and both vagal and spinal ganglia were examined. We observed robust labeling in vagal ganglia but minimal labeling in spinal ganglia, demonstrating that most small intestinal mucosal afferents are of vagal origin.

6. The sex differences highlighted in Figure 6a are interesting but remain unexplained. What contributes to this sex bias? Does the tuft cell-enterochromaffin cell crosstalk preferentially occur in females? Is it responsive to hormone signaling? Are other behavioral responses to IL-25 infection sex-dependent? Are behavioral responses to helminth infection sex-dependent? The authors mention that “sex-specific effects could not be adequately ascertained for these Nb infection experiments”, but do not show the data based on which they make this statement.

Thank you for raising this important point. We agree that the sex differences in spontaneous behaviors are interesting and could have several potential explanations. We have demonstrated in separate studies that the EC cell-mucosal afferent circuit is sensitized in females compared to males in an estrogen-dependent manner (PMID: 36949192; PMID: 40501885). Furthermore, it has been shown that androgens suppress ILC2 number and activity in males (PMID: 28484078), and females are more susceptible in various ILC2-dependent asthma models (PMID: 29186686; PMID: 30913260). These mechanisms could contribute to the observed sex differences.

However, in light of the referee’s thoughtful comments about the interpretation of these behavioral data, we have now combined sexes for the food intake measurements and refocused the manuscript on food intake (Fig. 6a-c). We have moved all other behavioral data to Extended Data Fig. 10 and explicitly acknowledged that the nature of aversive sensations during type 2 inflammation remains unclear (lines 276 and 307). Given this uncertainty about what these behaviors represent, we believe that attempting to explain sex-specific differences would be premature and potentially confusing. We have therefore de-emphasized these sex differences in the revised

manuscript.

Minor points and suggestions:

1. The authors now included additional quantifications of cFos in the NTS vs. AP and DMV and conclude that changes can only be observed in the NTS. This is in stark contrast to the image shown in Figure 5j, which actually shows the strongest differences in the AP and least pronounced changes in the NTS. It would be good to include all the images that the quantification data is based upon.

We thank the reviewer for this comment. As acknowledged in our previous responses, we had noted a relatively higher background signal in the area postrema (AP) of IL-25–treated PFTox(-) animals, which we have carefully monitored throughout our analyses. Importantly, this background was determined to be nonspecific (based on the absence of colocalization between the fluorescent signal and nuclear staining) and more prevalent if sections are examined when not freshly prepared.

To address this concern more rigorously, we generated an independent cohort of IL-25-treated animals, from which fresh tissue sections and immunofluorescence staining for cFOS was performed. The panel in question has now been replaced with the newly acquired data. Notably, quantification of cFOS⁺ neurons from these animals yielded results consistent with our previous findings. Correspondingly, both the quantification (now presented in Fig.5k and Extended Data Fig. 7a) and the underlying raw data have been updated.

With respect to the reviewer's observation that "*the authors now include additional quantifications of cFOS in the NTS vs. AP and DMV and conclude that changes can only be observed in the NTS, which appears in contrast to Fig. 5j,*" we now ensure that the representative images precisely correspond to the quantified datasets. The apparent discrepancy in the earlier version likely reflected non-specific background fluorescence in the AP region, which has been resolved through examination of new, freshly prepared samples.

2. The authors have re-arranged the flow of the manuscript, which results in a new logical sequence that is at times hard to follow. Some examples are:

a) The first sentence of the first section: "EC cells are now recognized as polymodal integrators of noxious stimuli in the gut, prompting us to ask whether they might be a target for ACh activation". ACh comes out of nowhere – why does the function of EC cells as integrators of noxious stimuli prompt an exploration of ACh responses?

b) The first sentence of the second section: "Having identified crypt EC cells as a bona fide target of ACh, we wondered whether and how they are activated by cholinergic tuft cells". Why tuft cells rather than ACh-producing neurons?

We agree that the logical flow needs improvement following various revisions to the text, which has now been achieved through smoother transitions that clarify the logical progression of the manuscript (lines 66 and 103). We thank the referee for encouraging us to step back and read the manuscript with fresh eyes.

Referee #2 (Remarks to the Author):

Summary of the key results

In this revised manuscript Touhara et al dissect a neuroepithelial sensing loop from tuft cells via EC cells to the vagus engaged in aversive behaviors in the initial stages of type 2 immune intestinal inflammation.

The authors demonstrate that tuft cells are indeed a source of acetylcholine generated in response to tuft cell ligands. Ach from tuft cells activates EC cells through Ach receptors and EC cells transmit signals to the vagus through serotonin to mediate signals leading to aversive behaviors likely due to visceral pain. The authors dissect each step using in vitro cultures and whole intestinal preps with Ach and serotonin sensing report mice. Interestingly, the authors find spatial segregation of tuft cells and EC cells (and specifically their subsets that can crosstalk) at homeostasis and a much more intimate connection in the setting of tuft cell hyperplasia after IL-25 stimulation or during helminth infections.

Originality: The authors have previously elegantly shown the important role of EC cells signaling to the vagus and brain through serotonin. Now they add another layer to this system where tuft cells are the frontline sensor and signal to EC cells through acetylcholine.

Data and methodology: The authors use multiple innovative and complementary approaches including Ach and Serotonin

reporter/GRAB mice, M1R GCamp, to detect generation of each of the neuromediator in culture and in tissue preps, signaling from tuft cells to EC cells and then to sensory neurons, including reporter mice with tuft cell genetic deletion and epithelial cell-specific deletion of choline acetyltransferase and finally an integrated physiologic model in the setting of helminth infection.

Conclusions: The conclusion of this revised manuscript are robust backed by several techniques and models for each conclusion making this a comprehensive study of the gut tuft to EC cell crosstalk during helminth infection. The appropriate papers are referenced. The graphical summary and discussion nicely summarize the findings.

The authors have addressed all of my previous comments/suggestions for improvement.

Referee #3 (Remarks to the Author):

Touhara and colleagues have substantially strengthened their manuscript examining the connection between tuft cells, enterochromaffin cells, and type 2 immune induced aversive behavior. The authors have addressed all previous concerns with rigor and clarity.

The speculative GasderminC component has been removed and replaced with a well supported model in which tuft cells release acetylcholine through TRPM5 dependent and cytokine induced mechanisms. Using *Nippostrongylus brasiliensis* infection together with tuft cell deficient (*Pou2f3*-KO) and epithelial ChAT deficient mice, the authors show that tuft cell derived acetylcholine drives enterochromaffin cell activation, serotonin release, and feeding suppression, findings further supported by expanded behavioral and cFOS analyses in the nucleus tractus solitarius.

The revised manuscript is clear, technically sophisticated, and conceptually strong. It provides compelling evidence that epithelial crosstalk regulates behavioral responses during type 2 immune activation. I am satisfied that all concerns have been fully addressed and that this work represents a significant scientific contribution. I congratulate the authors on an interesting and important study.

Referee #1 (Remarks to the Author):

The authors have appropriately responded to my remaining concerns. I do have to admit that it is somewhat dispiriting that the conclusions of the manuscript get diminished and more vague with every revision (from gasdermin C-mediated epithelial crosstalk in nocifensive behavior, to aversive behavior, and now to simply food intake), and that new results previously introduced are now deemed “inconclusive”. I certainly do not want to burden the authors with additional revisions and hope the conclusions presented in the newest version of the manuscript are now solid despite the remaining oddities acknowledged by the authors.

Response:

We sincerely thank Referee 1 for his/her thoughtful and rigorous evaluation of our manuscript throughout the review process.

We appreciate the referee's candid reflection on the evolution of our conclusions. We understand that the refinement of our claims from the initial gasdermin C hypothesis to the current focus on food intake may appear as a narrowing of scope. However, we are confident that the conclusions presented in this final version are robustly supported by our data. The manuscript now provides a multifaceted yet focused demonstration of how parasitic infection activates a tuft cell–enterochromaffin cell–vagal afferent pathway to modulate host feeding behavior. We believe this represents a significant and well-substantiated contribution to our understanding of gut-brain communication during type 2 inflammation.

We thank all the referees for their constructive criticism during this review process, which has ultimately helped us refine our thinking and expand our experiments to connect cellular insights with in vivo physiology, thereby strengthening the rigor, reliability, and impact of our work.